# Epigenetic modulators link mitochondrial redox homeostasis to cardiac function in a sex-dependent manner

Zaher ElBeck [1,2] ✉, Mohammad Bakhtiar Hossain [3], Humam Siga [1], Nikolay Oskolkov [4], Fredrik Karlsson [5], Julia Lindgren [6], Anna Walentinsson [7], Dominique Koppenhöfer [1], Rebecca Jarvis [8], Roland Bürli[8], Tanguy Jamier [8], Elske Franssen[8], Mike Firth[5], Andrea Degasperi [5,9], Claus Bendtsen [5], Robert I. Menzies[3], Katrin Streckfuss-Bömeke[10,11,12], Michael Kohlhaas[12], Alexander G. Nickel[12], Lars H. Lund[13], Christoph Maack [12], Ákos Végvári [14] & Christer Betsholtz [1,2]

While excessive production of reactive oxygen species (ROS) is a characteristic hallmark of numerous diseases, clinical approaches that ameliorate oxidative stress have been unsuccessful. Here, utilizing multi-omics, we demonstrate that in cardiomyocytes, mitochondrial isocitrate dehydrogenase (IDH2) constitutes a major antioxidative defense mechanism. Paradoxically reduced expression of IDH2 associated with ventricular eccentric hypertrophy is counterbalanced by an increase in the enzyme activity. We unveil redox-dependent sex dimorphism, and extensive mutual regulation of the antioxidative activities of IDH2 and NRF2 by a feedforward network that involves 2-oxoglutarate and L-2-hydroxyglutarate and mediated in part through unconventional hydroxy-methylation of cytosine residues present in introns. Consequently, conditional targeting of ROS in a murine model of heart failure improves cardiac function in sex- and phenotype-dependent manners. Together, these insights may explain why previous attempts to treat heart failure with antioxidants have been unsuccessful and open new approaches to personalizing and, thereby, improving such treatment.

The incidence of heart failure (HF) continues to increase worldwide, and despite improvements in medical and device-based treatments, HF morbidity and mortality remain unacceptably high. In this context, metabolic dysfunction and oxidative stress play crucial roles[1], with mitochondria being the major source of reactive oxygen species (ROS) in cardiac tissue. Indeed, formation of ROS through leakage of electrons from complex I of the respiratory chain is elevated in failing hearts[2,3].

More recently, we revealed that disturbance of excitation-contraction coupling in failing cardiac myocytes attenuates activation of the Krebs cycle by $Ca^{2+}$, causing oxidation of NADH, $FADH_2$, and

NADPH and thereby disrupts enhancement of energy production in response to increased demand[4]. While NADH and $FADH_2$ donate electrons to the mitochondrial respiratory chain for the production of ATP, NADPH is a necessary cofactor for antioxidative processes that eliminate hydrogen peroxide ($H_2O_2$), i.e., the enzyme glutathione peroxidase and the peroxiredoxin/thioredoxin system[5]. Nonetheless, clinical trials designed to treat HF by ameliorating oxidative stress have, to date, been unsuccessful[6,7] and, in some patients with chronic kidney disease, have even been reported to promote the development of HF[8].

One potential explanation for this failure is that signaling by physiological levels of ROS has protective effects on the heart, and

consequently, that excessive quenching of these species may be undesirable[9,10]. Accordingly, in the attempt to develop more effective treatments for HF, a more detailed molecular understanding of cardiac redox regulation is necessary. Furthermore, since HF is associated with a variety of etiologies and phenotypes[11], an individualized approach to interventions that alter mitochondrial metabolism and redox status in cardiac tissue may be required.

Alterations of mitochondrial biogenesis and function in response to various stimuli and stressors are tightly regulated through control of the expression of key nuclear genes. In addition to producing ATP, it is becoming increasingly clear that mitochondria generate metabolites that participate in cellular signal transduction and serve as cofactors for various biochemical processes, including several reactions involved in gene expression and its regulation[12]. For instance, 2-oxoglutarate (2OG, also known as α-ketoglutarate, α-KG) produced via the Krebs cycle is an essential cofactor for certain dioxygenases (OGDD) that catalyze hydroxylation of nucleic acids, chromatin, proteins, lipids, and metabolites[13]. A major route for synthesis of 2OG involves oxidative decarboxylation of isocitrate catalyzed by mitochondrial isocitrate dehydrogenase (IDH2).

Importantly, IDH2 plays an essential role in antioxidant systems, is present at high levels in cardiac tissue[14], and is mainly responsible for direct regeneration of mitochondrial NADPH in cardiac myocytes[15]. In addition, IDH2 contributes indirectly to NADPH production by providing the substrate for 2-oxoglutarate dehydrogenase (OGDH), which generates NADH that is utilized preferentially by nicotinamide nucleotide transhydrogenase (NNT) to regenerate NADPH[16]. The activity of IDH2 is altered through changes in $NAD^+$ levels or oxidative stimuli, which are modulated by SIRT3 and SIRT5[17,18], as well as stimulated when the Krebs cycle is activated by ADP and/or $Ca^{2+}$[19]. NRF2, a master regulator for antioxidative response, is not an upstream regulator of IDH2, whose gene lacks the antioxidant response element (ARE) in its promoter. Both $Idh2^{-/-}$ and $Nrf2^{-/-}$ strains of mice are viable, but with reduced defenses against oxidative stress and enhanced susceptibility to heart failure[20,21].

Moreover, several members of the OGDD family catalyze epigenetic modifications, e.g., demethylation of histones and gDNA[22]. OGDD enzymes referred to as TETs can convert 5-methylcytosine to 5-hydroxymethylcytosine (5hmC), thereby mediating DNA ten-eleven translocation[23]. Indeed, hydroxymethylation of cytosine in DNA has unique roles in normal development and physiology, as well as in pathological processes, especially in organs containing terminally differentiated cells[24]. In connection with the maladaptive remodeling of the heart, the state of 5-methylcytosine hydroxylation, particularly in genes related to energy production and mitochondrial function, undergoes pronounced alteration[25]. The activities of various TET enzymes are largely dependent on access to their co-factor 2OG and its oxidized enantiomers L/D 2-hydroxyglutarate (2HG). L2HG is produced during adaptation to hypoxic and oxidative stress, and this metabolite and the changes in 5hmC it causes may play an important role in maladaptive cardiac remodeling and HF development.

The current investigation was designed to test our hypothesis that IDH2 is a major regulator of antioxidant defenses in the heart. We elucidate the molecular and regulatory epigenetic functions of IDH2, 2OG, and L2HG and provide unique insights into the impact of antioxidative capacity on HF. These findings deepen our molecular understanding of the transcriptional and epigenetic control of redox homeostasis in cardiac tissue and may aid in the development of new approaches to treat HF.

## Results

### Expression of IDH2 is downregulated in association with eccentric hypertrophy

Previously, we performed RNA sequencing on left ventricular tissue (LV) from end-stage patients with dilated cardiomyopathy harboring mutations in lamin A/C (*LMNA*), RNA binding motif protein 20 (*RBM20*), and titin (*TTN*)[26]. Here, to examine the involvement of mitochondria and related molecular pathways in HF, we performed an Ingenuity Pathway Analysis (IPA) on the same data. This approach revealed significant enrichment in the levels of RNAs encoding proteins involved in pathways related to altered metabolic and redox homeostasis and dysfunction in mitochondria (Fig. 1a), supporting previous observations[6]. We next confirmed these findings by performing IPA on published transcriptomic data from 65 samples of cardiac tissue from patients with idiopathic dilated cardiomyopathy[27], and from another cohort of 13 patients with ischemic cardiomyopathy (ICM), some of which also had a diabetic history[28] (Supplementary Fig. 1a).

To unravel the underlying cause of this mitochondrial dysfunction, we subsequently analyzed left ventricle (LV) samples from patients with dilated cardiomyopathy (DCM), as well as from muscle LIM protein (*Mlp*)-deficient (*Mlp*$^{-/-}$ or *Csrp3*$^{-/-}$) mice, which develop eccentric hypertrophy comparable to human DCM due to alterations in the cytoarchitecture of cardiomyocytes[29]. As in the human material, RNAs encoding proteins involved in pathways related to mitochondrial dysfunction were enriched in the *Mlp*$^{-/-}$ mice (Fig. 1b and Supplementary Fig. 1b).

Transmission electron microscopy of the murine samples revealed no relevant alterations in mitochondrial surface area or structure (Supplementary Fig. 1c); nor were substantial reductions in the levels of components of complexes of the mitochondrial electron transport chain observed by Western blotting of either the human (Fig. 1c and Supplementary Fig. 1d) or murine samples (Supplementary Fig. 1e, f). In agreement with these findings, respiration by mitochondria isolated from the transgenic animals was comparable to the corresponding findings with control mice (Supplementary Fig. 1g, h). Furthermore, no statistically significant general common dysregulation in the expression of genes involved in the regulation of mitochondrial biogenesis was observed in the human and murine samples (Supplementary Fig. 1i). Thus, although elevated pre- and afterload increase the energy demand in HF[30], mitochondrial morphology and general function appear largely unchanged.

Since the energetic deficit associated with HF is, at least in part, related to alterations in substrate utilization[30], we next identified genes encoding mitochondrial metabolic enzymes by intersecting the Mammalian Metabolic Enzyme and MitoCarta2 databases[14,31]. This revealed 447 genes, of which 45 were significantly dysregulated in both patients with DCM and *Mlp*$^{-/-}$ mice (Fig. 1d, e). Intriguingly, *IDH2* was the most downregulated gene in this intersection (Fig. 1e, f). Moreover, *IDH2* was also significantly downregulated in all other transcriptomic data sets from failing genetic DCM, idiopathic DCM and ICM hearts, as well as from hearts of rodent models of genetic DCM, e.g., *Mlp*$^{-/-}$ and mutated phospholamban[32] *Pln-R14*$^{\Delta/\Delta}$; ischemic DCM, e.g., myocardial infarction[32]; diabetic DCM, e.g., rats treated with streptozotocin (STZ)[33] (Fig. 1f, g and Supplementary Fig. 1j). The level of IDH2 protein in failing ischemic human hearts has been reported to be reduced[34], an observation which we confirmed in human DCM (Fig. 1c and Supplementary Fig. 1d) and *Mlp*$^{-/-}$ cardiac materials (Fig. 1h and Supplementary Fig. 1e), although this reduction was less pronounced than the decrease in the corresponding level of mRNA.

In contrast to above-described rodent models, which all display eccentric hypertrophy, the expression of *Idh2* was not dysregulated in both genetic *Mybpc3*$^{-/-}$ and pressure overload induced by transaortic banding (TAB) mice (Fig. 1i and Supplementary Fig. 1l), which are models of concentric hypertrophy[35,36]. Thus, downregulation of *IDH2* appears to occur specifically in cardiomyopathies involving eccentric hypertrophy.

### Oxidative stress downregulates *Idh2*

IDH2 is the primary source for the regeneration of mitochondrial NADPH required for enzymatic elimination of $H_2O_2$ in

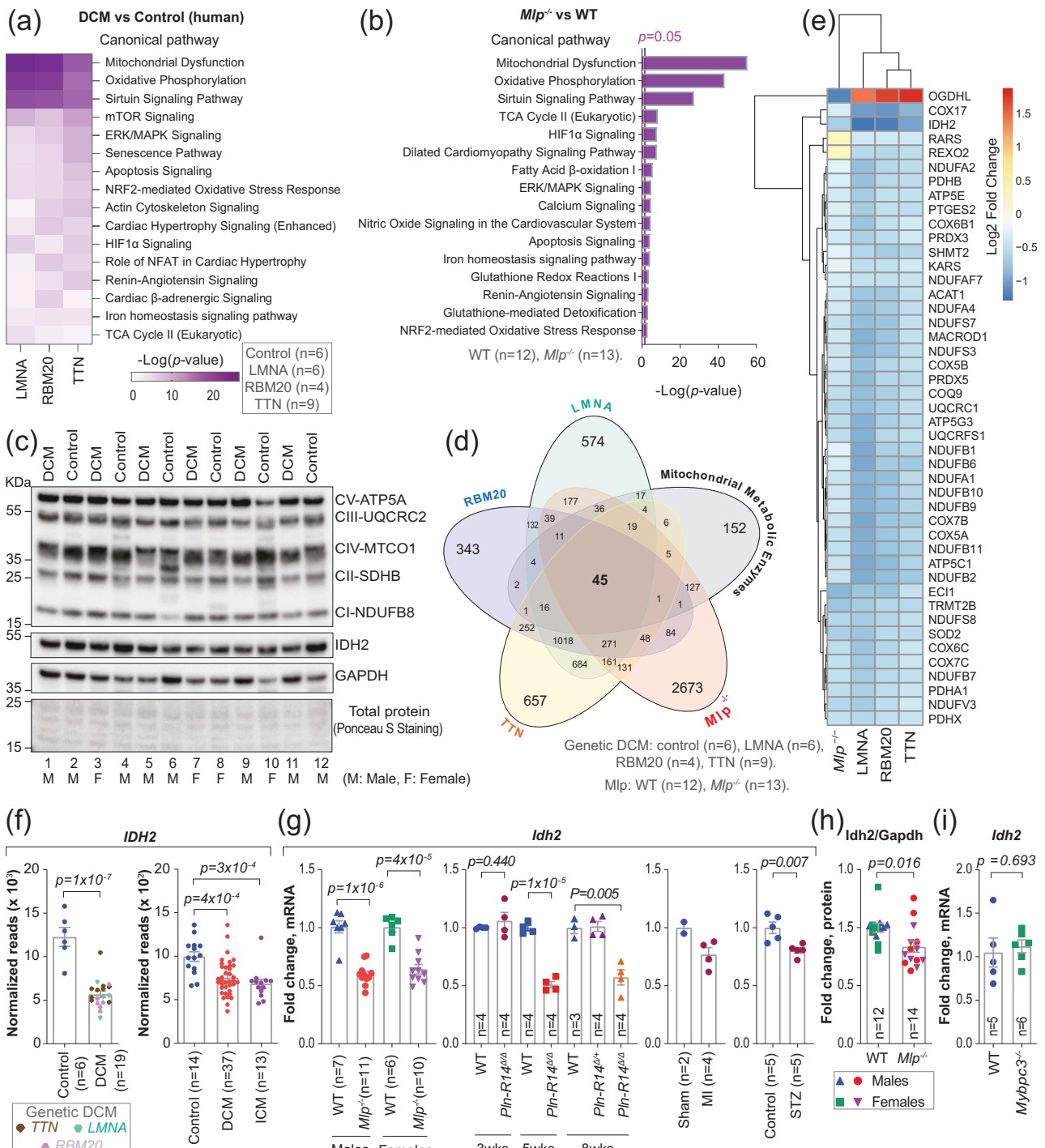

**Fig. 1 | Mitochondrial dysfunction and downregulation of IDH2 expression in association with eccentric hypertrophy. a** Selected pathways from IPA enrichment analysis on published transcriptomic data from left ventricle (LV) myocardium of patients with dilated cardiomyopathy (DCM)[26]. **b** Selected pathways from IPA enrichment analysis on transcriptomic data from LV of $Mlp^{-/-}$ mice (males + females). **c** Western blotting of OXPHOS complexes and IDH2 in the LV of patients with eccentric genetic DCM (the bands are quantified in Supplementary Fig. 1d, and the samples' metadata are described in Supplementary Table 1). **d** Venn diagram of genes encoding mitochondrial metabolic enzymes, and differentially expressed genes (DEGs) in the LV of patients with DCM (described in Fig. 1a) or $Mlp^{-/-}$ mice. Only DEGs with $p < 0.05$ and adjusted $P$ value $< 0.01$ were included in this analysis. **e** List of genes encoding for mitochondrial metabolic enzymes that were commonly dysregulated in all data sets analyzed in panel (**d**). **f** $IDH2$ expression in LV of patients with genetic DCM described in Fig. (1a), and in LV of patients with idiopathic DCM and ischemic cardiomyopathy (ICM)[28]. **g** $Idh2$ expression in LV of $Mlp^{-/-}$ mice analyzed by qPCR; or inferred from published transcriptomic data from LV of $Pln-R14^{\Delta/\Delta}$ mice[32]; mice with myocardial infarction (MI) surgery, 8 weeks after the surgery[32]; and rat model of diabetic cardiomyopathy induced by streptozotocin (STZ)[33]. **h** Quantification of Western blotting of Idh2 normalized to Gapdh in LV of $Mlp^{-/-}$ mice. Blots are shown in Supplementary Fig. 1k. **i** qPCR analysis of $Idh2$ expression in the LV of male $Mybpc3^{-/-}$ mice. Bars represent mean ± SEM, analyzed with unpaired two-tailed $t$-test. The sample size is indicated in all figures. Source data and uncropped blots are available in the Source Data file.

cardiomyocytes[15,16]. As oxidative stress is elevated in failing hearts[2,3], we next examined how oxidative stress modulates IDH2 expression and activity.

When neonatal rat cardiomyocytes (NRCMs) were subjected to oxidative stress through exposure to increasing concentrations (0-300 $\mu$M) of $H_2O_2$ for 6 h (Fig. 2a), concentration-dependent down-regulation of *Idh2* occurred (Fig. 2a). Moreover, as the level of $H_2O_2$ gradually declined, *Idh2* returned almost to normal, particularly in the case of cells exposed to low $H_2O_2$ concentrations. At the same time, further incubation of these cells with fresh medium free from $H_2O_2$ for an additional 24 h did not raise the level of *Idh2* expression any further, indicating the potential presence of long-lasting effects (Fig. 2a).

Utilizing RT-qPCR, we confirmed that exposure of NRCMs to oxidative stress induces expression of *Hmox1*, *Nqo1,* and *Osgin1*, which are downstream target genes for Nrf2 (Fig. 2a). This effect was also associated with a loss of cardiomyocytes, as indicated by a decrease in the total amount of RNA recovered (Fig. 2a). This loss of RNA exerted no impact on the downregulation of *Idh2* observed, since 5 ng total RNA was employed in all qPCR reactions, and it is also evident from the concurrent increase in the expression of *Hmox1, Nqo1* and *Osgin1*.

## Downregulation of *Idh2* is associated with enhancement of its protein activity

Post-translational modifications modulate the stability and activity of IDH2 in response to oxidative stimuli, such as desuccinylation that elevates the activity of this enzyme[17]. Our analysis of publicly available mass spectroscopic data on the LV of patients with ICM[34] revealed desuccinylation of multiple lysine residues in IDH2 (Fig. 2b). Furthermore, the overall level of protein succinylation, as quantified by Western blotting, was lowered in *Mlp*[-/-] hearts (Supplementary Fig. 2a), reminiscent of what has been observed in human HF[34].

Therefore, we measured the activity of Idh2 protein in sub-sarcolemmal mitochondria isolated from *Mlp*[-/-] LV. Intriguingly, despite the lower levels of mRNA and protein (Fig. 1g, h), this activity was higher in *Mlp*[-/-] mice than in WT littermates (Fig. 2c). Moreover,

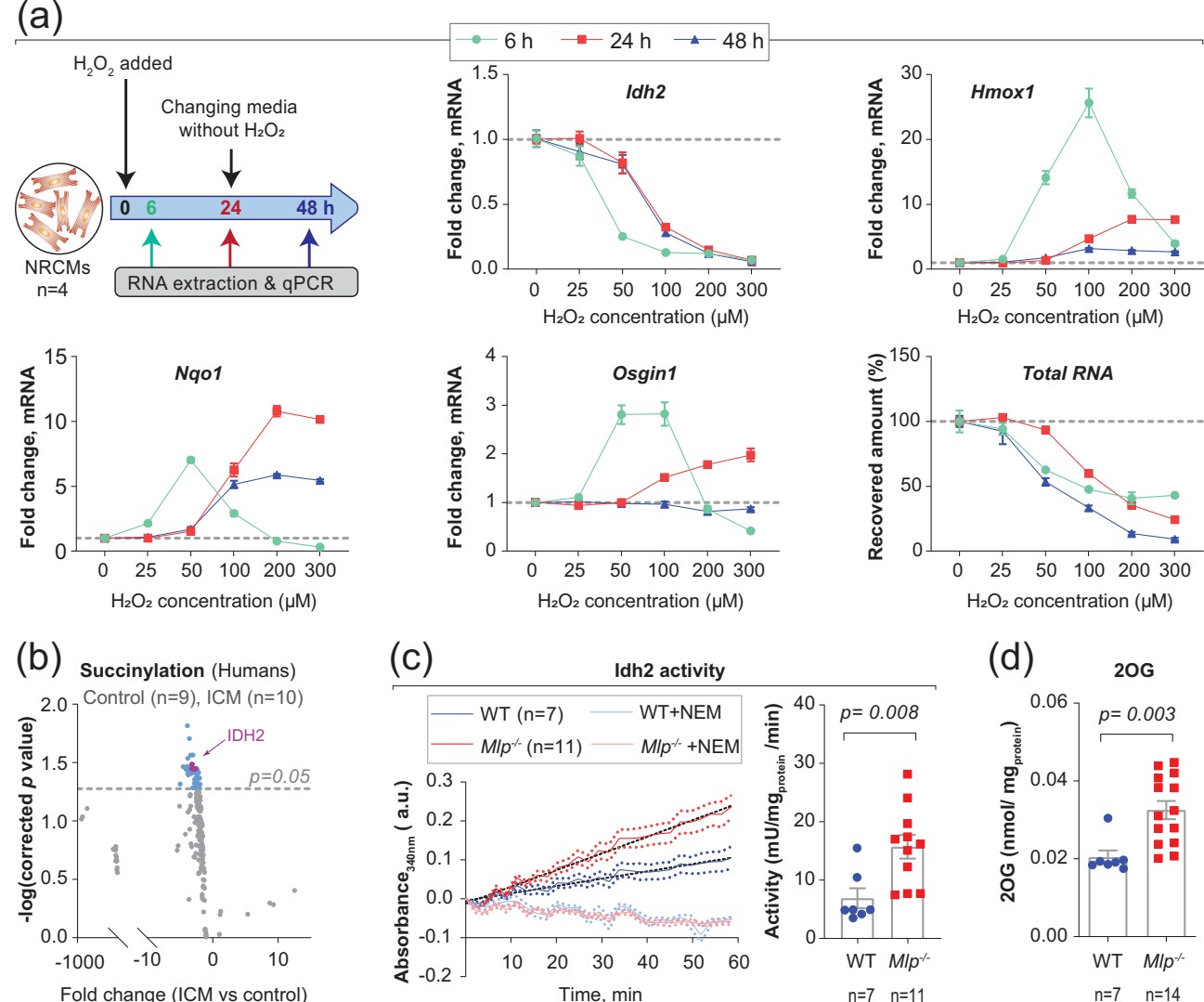

**Fig. 2 | Redox status modulates the expression and the activity of IDH2.**
**a** Experimental scheme depicting the treatment of NRCMs with increasing concentrations of $H_2O_2$, and qPCR analysis of *Idh2*, *Hmox1*, *Nqo1* and *Osgin1* expression, normalized to untreated control. Equal amounts (5 ng) of total RNA were utilized for all qPCR reactions. Total RNA depicts the recovered amount of extracted RNA from the wells of $H_2O_2$-treated NRCMs, compared to untreated control. Horizontal dashed lines mark the mean levels in control samples. Values represent the average of 4 individual wells ±SEM. **b** Volcano plot for differentially succinylated peptides of mitochondrial metabolic enzymes in the LV of patients with ICM, obtained from publicly available mass spectroscopic data[34]. **c** Idh2 activity in cardiac mitochondria isolated from frozen LV myocardium of male *Mlp*[-/-] with or without the Idh2-activity-inhibitor, N-ethylmaleimide (NEM). **d** 2OG levels in the LV of male *Mlp*[-/-]. Bars represent mean ± SEM, analyzed with unpaired two-tailed *t*-test. The sample size is indicated on all figures. Source data are available in the Source Data file.

this elevation was associated with higher levels of 2OG (Fig. 2d). Observed discrepancies between substantially lowered *IDH2* mRNA, slightly lowered protein, and its increased activity in DCM indicate the presence of opposing mechanisms that maintain a certain level of IDH2 activity.

### L-2-hydroxyglutarate upregulates Idh2 and deactivates Nrf2

Under hypoxic conditions, cellular levels of 2OG and lactate rise while the level of L-malate falls, promoting lactate (LDH) and malate dehydrogenases (MDH) to reduce 2OG to L-2-hydroxyglutarate (L2HG)[37,38]. As we observed increased levels of 2OG in DCM, we applied chemical derivatization followed by targeted mass spectrometry to characterize the metabolic remodeling associated with HF.

*Mlp*[−/−] hearts displayed reduced levels of L-malate, while lactate was elevated (Fig. 3a and Supplementary Fig. 3a). A more detailed analysis through chemical derivatization revealed that the level of L2HG, but not D2HG, was increased (Fig. 3b). This was not associated with upregulation of *Slc2a1*, a target gene for hypoxia inducible factor 1 (Hif1α) (Supplementary Fig. 3b).

L2HG participates in a redox couple (2OG/L2HG) that regenerates NADH[39]. To examine the impact of L2HG and D2HG on the redox status of cardiomyocytes, the levels of these compounds in NRCMs were

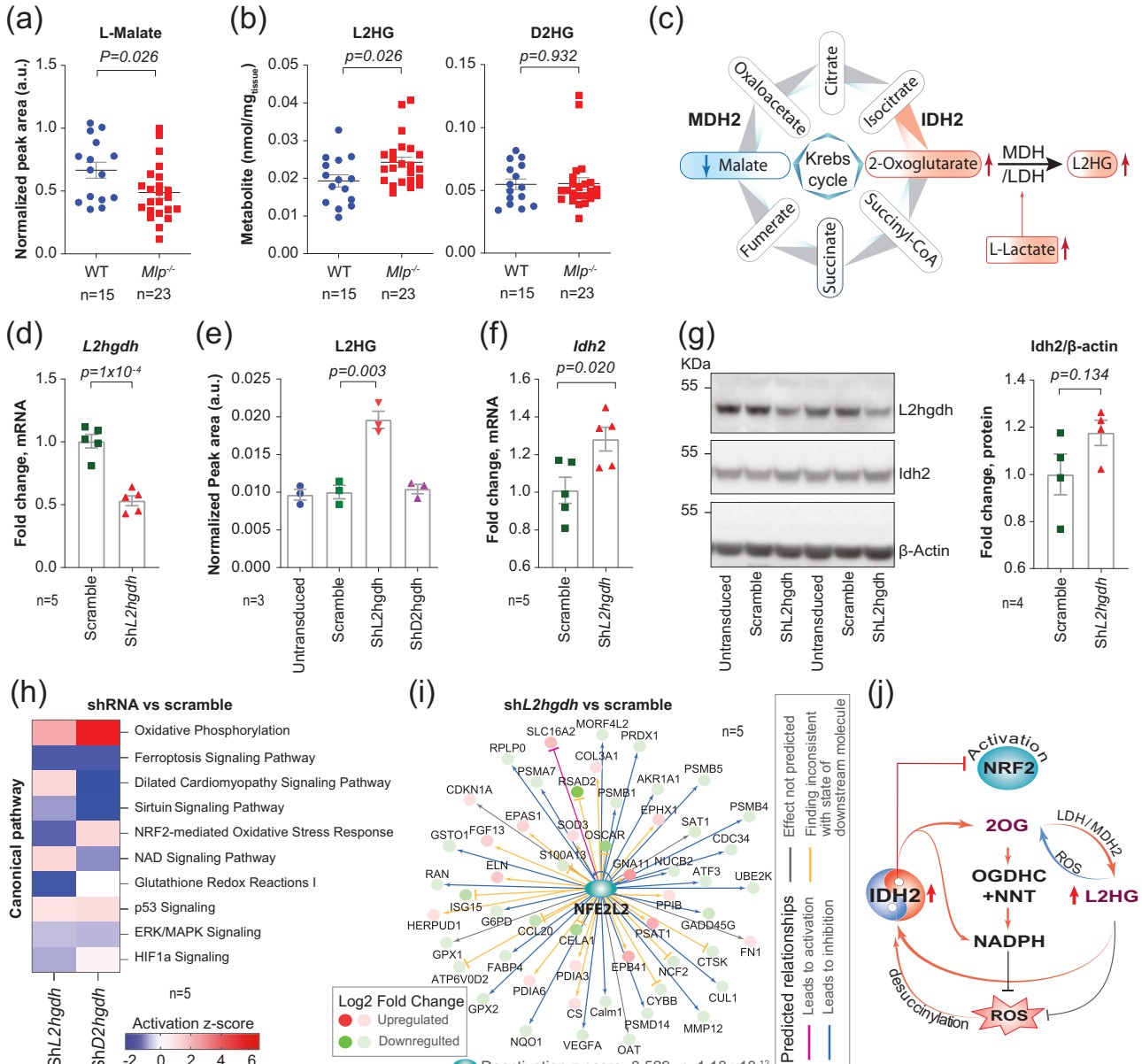

**Fig. 3 | Antioxidative role of L2-hydroxyglutarate (L2HG) in the heart.**
**a**, **b** Targeted LC-MS/MS analysis of L-malate, L2HG, and D2HG in the left ventricle (LV) of male and female *Mlp*[−/−], normalized to tissue weight. **c** Scheme depicting metabolic remodeling of Krebs cycle and LDH leading to L2HG production. **d**–**g** NRCMs transduced with ShRNA targeting L2hgdh were analyzed for *L2hgdh* expression by qPCR in panel (**d**), L2HG levels by a targeted LC-MS in panel (**e**), Idh2 mRNA expression by qPCR in panel (**f**), and protein level by western blotting in panel (**g**). **h** Selected pathways from IPA comparison analysis on transcriptomic data from NRCMs transduced with ShRNA targeting *L2hgdh* or *D2hgdh*. **i** IPA upstream regulator analysis of Nrf2 in NRCMs transduced with ShRNA targeting *L2hgdh* with Right-Tailed Fisher's Exact Test. **j** Proposed molecular mechanism for the antioxidative role of L2HG in the heart. Values in figures d-i are the average of n (numbers indicated on the figures) individual wells. Bars represent mean ± SEM with unpaired two-tailed *t*-test. Source data and uncropped blots are available in Source Data file.

increased by shRNA-mediated knockdown of *L2hgdh* and *D2hgdh*, i.e., the genes that encode mitochondrial dehydrogenases that oxidize L2HG and D2HG to 2OG, respectively. Approximately 50% knockdown of *L2hgdh* resulted in a two-fold increase in the level of L2HG (Fig. 3d, e), while approximately 70% knockdown of *D2hgdh* elevated the level of D2HG 1.5-fold (Supplementary Fig. 3c, d).

Although elevated levels of D2HG have been reported to downregulate IDH2 in a variety of cell types[40], we observed that elevated levels of L2HG in NRCMs, but not D2HG, were associated with upregulation of both Idh2 mRNA and protein (Fig. 3f, g and Supplementary Fig. 3e), suggesting a role for L2HG in regulating cellular redox homeostasis. Transcriptomic profiling followed by IPA analysis revealed that L2HG, but not D2HG, attenuates Nrf2-mediated responses to oxidative stress, as well as glutathione redox reaction I (Fig. 3h, Supplementary Fig. 3f, g). We confirmed deactivation of certain components of the Nrf2 pathway by L2HG through IPA upstream regulator analysis of Nrf2 and through qPCR analysis of the expression of the downstream genes *Hmox1*, *Nqo1,* and *Osgin1* (Fig. 3i and Supplementary Fig. 3h).

These data suggest the presence of a vicious cycle, by which the increases in IDH2 activity and ensuing elevations in 2OG and L2HG levels enhance the antioxidative capacity of cardiomyocytes (Fig. 3j). However, the actual mechanism for downregulation of *IDH2* in response to oxidative stress remains to be elucidated.

### Mutual redox regulation downregulates *IDH2*

To resolve the redox paradox described above, we treated NRCMs with 5 μM sulforaphane (SF) and/or 3 mM N-acetyl cysteine (NAC) and then monitored the expression of *Idh2* and three downstream target genes for Nrf2 (Fig. 4a). SF enhances production of ROS by mitochondria in cardiomyocytes[41], while NAC donates a thiol to produce sulfane sulfur species, which are potent scavengers of ROS[42,43].

SF induced the expression of Nrf2 target genes while attenuating *Idh2* expression (Fig. 4a), effects reminiscent of those produced by $H_2O_2$ (Fig. 2a). NAC mitigated these effects of SF, while NAC alone upregulated *Idh2* and reduced the levels of downstream Nrf2 targets (Fig. 4a), changes comparable to those that occur upon induction of L2HG (Fig. 3f). These observations indicate the existence of an antioxidative control mechanism in which Idh2 and Nrf2 regulate each other's activity (Fig. 4b).

### Antioxidative response to low concentrations of oxidants triggers a reductive milieu

When NRCMs were exposed to different concentrations of SF, the lowest concentration, i.e., 1 μM, induced only a slight downregulation of *Idh2* and a mild induction of *Hmox1*, while stronger induction of *Nqo1* and *Osgin1* (Fig. 4c and Supplementary Fig. 4a), with no observable cardiomyocyte death inferred from the amount of recovered total RNA (Supplementary Fig. 4b). Targeted mass spectrometry quantification of reduced and oxidized glutathione (GSH and GSSG, respectively) in NRCMs revealed that the exposure to low concentrations of SF, i.e., 1 and 2 μM, and especially the 1 μM concentration, substantially increased intracellular concentrations of GSH. The pronounced increase in GSH was accompanied by only a minor increase in GSSG concentrations, indicating a significant increase in the net-reducing equivalents in these cells. For instance, in cells treated with 1 μM SF, we observed an increase of 28.9 nmol/mg$_{protein}$ in GSH level compared to only 0.119 nmol/mg$_{protein}$ increase in GSSG level (Fig. 4d, e and Supplementary Fig. 4c). Whereas higher concentrations of SF (>5 μM), which induced substantial cell death (Supplementary Fig. 4b), did not appear to increase the level of GSH, but rather gradually lowered it (Fig. 4d, e and Supplementary Fig. 4c).

Furthermore, consistent patterns of GSH and GSSG modulation upon exposure to SF were similarly evident in human cardiomyocytes, like induced pluripotent stem cell-derived cardiomyocytes (hiPSC-CMs), albeit with NRCMs exhibiting lower GSH and higher GSSG basal concentrations (Fig. 4d, e and Supplementary Fig. 4d). These differences between the species could be attributed to the metabolic nature of cardiomyocytes. In addition, we also confirmed that 1 μM of SF was well-tolerated in a different cell type - a cell line of human foreskin fibroblasts (hFF1) - and elicited antioxidative response without inducing observable cleavage of caspase 3, a marker for apoptosis (Supplementary Fig. 4e, f, g). hFF1 cells also exhibited a substantial increase in the intracellular concentrations of GSH when exposed to low concentrations of SF,i.e., 1 and 2 μM, and a minor increase in GSSG concentration (Fig. 4d, e and Supplementary Fig. 4h), consistent with our observations in cardiomyocytes. Thus, as GSH represents a major reductant in almost all cells, the response to low concentrations of oxidants appears to trigger a reductive milieu in cells. This concept is further supported by observed normal levels of malondialdehyde (MDA) in cells treated with SF ≤ 5 μM in both NRCMs and hFF1 cells (Fig. 4f and Supplementary Fig. 4i). MDA is a marker for lipid peroxidation and oxidative damage[44].

### Sex associated differences in responses to oxidative stress

When we explored potential dysregulation of the expression of proteins involved in antioxidation in *Mlp*$^{-/-}$ hearts, we observed upregulation of Hmox1 and Nqo1 in both male and female mice (Fig. 4g). However, the increase in Hmox1 was more noticeable in male mice compared to female ones, a difference that was even more pronounced on the mRNA levels, where *Hmox1* mRNA was upregulated only in males (Fig. 4h). As Hmox1 is a marker for acute response to oxidants, these data indicate that male *Mlp*$^{-/-}$ hearts exhibit higher oxidative stress than females.

Subsequent quantification of GSH and GSSG in *Mlp*$^{-/-}$ hearts revealed a small increase in the level of GSH in *Mlp*$^{-/-}$, especially in female mice, which also exhibited an increase in GSSG levels (Fig. 4i). The increase in both GSH and GSSG indicates a reductive milieu in female *Mlp*$^{-/-}$ hearts, and reminiscent of what we observed in cells treated with low concentrations of SF (Fig. 4d, e and Supplementary Fig. 4c, d, h) or NAC (Supplementary Fig. 4j). Moreover, calculating the GSH/GSSG ratio, which is considered a marker for oxidative stress, revealed that male *Mlp*$^{-/-}$ hearts had substantially higher ratio in comparison to female ones (Fig. 4j). This was reflected on female *Mlp*$^{-/-}$ hearts having substantially lower levels of MDA (Fig. 4k).

To further characterize oxidative stress in male and female *Mlp*$^{-/-}$ hearts in greater detail on the transcriptome level, we carried out transcriptomic profiling at 12 weeks of age, followed by comparative IPA analysis. This showed that the female animals exhibited less extensive changes in Nrf2-mediated responses to oxidative stress and in glutathione redox reaction I when compared to males (Supplementary Fig. 4k), reflecting their lower level of oxidative stress, as suggested from measuring redox metabolites described above (Fig. 4i-k).

### Redox associated sex dimorphism in cardiac phenotype

As we observed sex-associated differences in responses to oxidative stress in *Mlp*$^{-/-}$ mice, we aimed next to investigate whether these differences are reflected on cardiac sex dimorphism. Echocardiography revealed strong negative correlation between left ventricle ejection fraction (LVEF) and the expression of the primary oxidative stress response gene *Hmox1* in mice exhibiting a severe phenotype (LVEF < 40%) (Fig. 5a). Moreover, the walls of the left ventricle of 14-week-old male mice were less thick than those of females, but with no significant difference at 10 weeks of age (Fig. 5b and Supplementary Fig. 4a). Furthermore, at the younger age females exhibited a greater fraction of LVEF (Fig. 5c), a difference that became more pronounced with age as LVEF improved in female mice, but not in males (Fig. 5d). This improvement in the LVEF in female animals was associated with

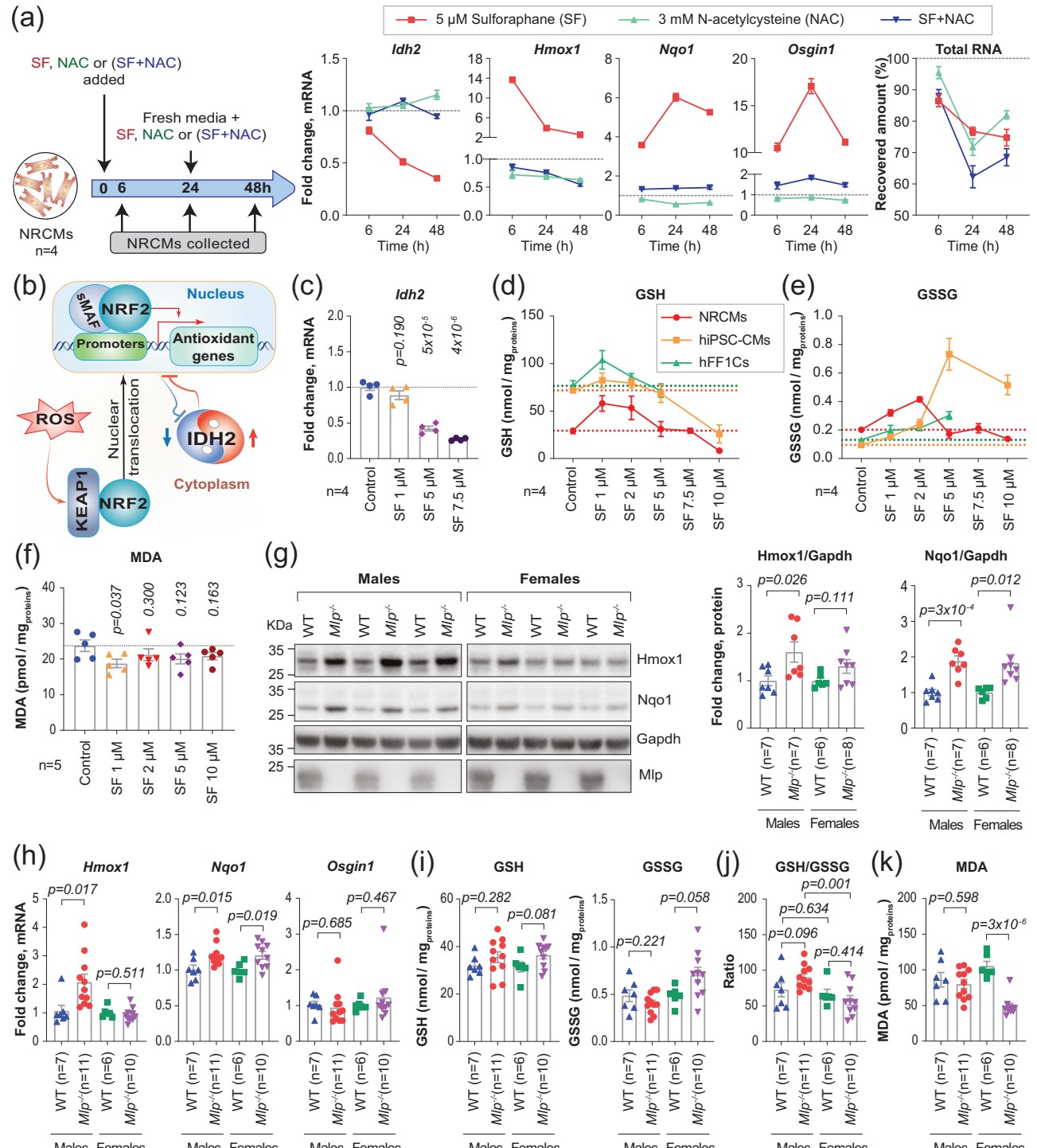

**Fig. 4 | Sex associated differences in response to oxidative stress. a** Scheme depicting the experimental approach for treating NRCMs with sulforaphane (SF), N-acetyl cysteine (NAC) or in combinations, qPCR analysis of *Idh2*, *Hmox1*, *Nqo1* and *Osgin1* expression in treated NRCMs normalized to untreated control, and recovered amount of total RNA extracted from the treated-NRCMs, compared to untreated controls. **b** Proposed molecular mechanism for the IDH2-NRF2 mutual regulation. **c** qPCR analysis of *Idh2* expression in treated NRCMs with increasing concentrations of SF, normalized to untreated control. **d**, **e** Targeted LC-MS/MS analysis of GSH and GSSG levels in NRCMs, hiPSC-CMs, and hFF1 cells treated with increasing concentrations of SF for 48 h, *p*-values and individual data points are displayed in Supplementary fig. (4c, d, and h). **f** Targeted LC-MS/MS analysis of

MDA levels in NRCMs treated with increasing concentrations of SF for 48 h, compared to untreated control. **g** Western blotting of Hmox1 and Nqo1 in the LV of *Mlp*⁻/⁻, and their subsequent quantification, normalized to Gapdh. **h** qPCR analysis of *Hmox1, Nqo1* and *Osgin1* expression in LV myocardium of *Mlp*⁻/⁻. **i**–**k** Targeted LC-MS/MS analysis of GSH, GSSG and MDA levels in the LV of male and female *Mlp*⁻/⁻, normalized to soluble proteins. Values in figures **a**–**f** are the average values of 4 or 5 individual wells/plates ±SEM. Depicted *p*-values on all figures represent two-tailed unpaired *t*-tests. Horizontal dashed lines in figures (**a**) and (**c**–**f**) mark the mean value in the control samples. Bars in figures (**g**–**k**) represent the mean ± SEM. Source data and uncropped blots are available in Source Data file.

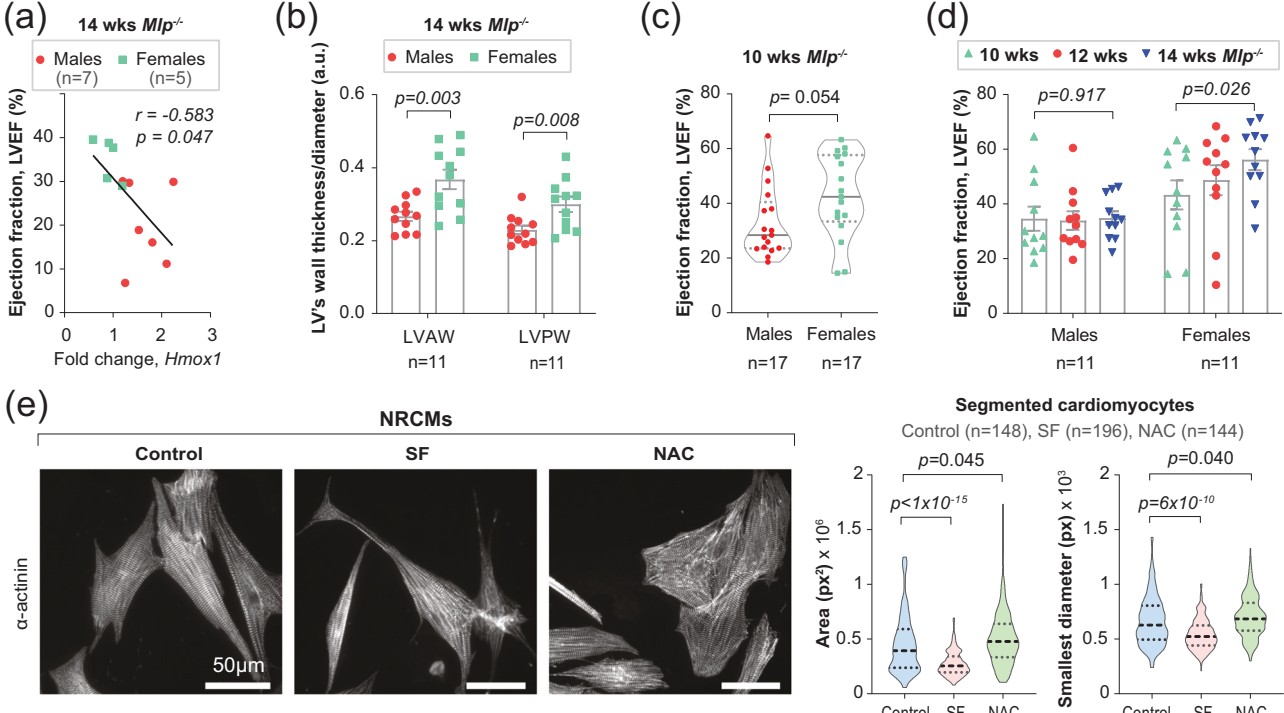

**Fig. 5 | Redox-associated sex dimorphism in cardiac phenotype. a** Pearson correlation between left ventricle ejection fraction (LVEF) of *Mlp*⁻ᐟ⁻ mice with reduced LVEF (EF < 40%) and cardiac expression of *Hmox1* estimated by qPCR. The *p*-value was derived from a two-tailed Pearson correlation analysis. **b** Thickness of the left ventricle anterior wall (LVAW) or posterior wall (LVPW), normalized to LV diameter, in male and female *Mlp*⁻ᐟ⁻ mice. The values are the average of systolic and diastolic states. Bars represent mean ± SEM with unpaired two-tailed *t*-test. **c** LVEF in male and female *Mlp*⁻ᐟ⁻ mice at the age of 10 weeks. The median is shown on the plot, with the interquartile range and unpaired two-tailed *t*-test. **d** Changes in LVEF

of male and female *Mlp*⁻ᐟ⁻ mice over time. Bars represent mean ± SEM with paired two-tailed *t*-test. **e** Neonatal rat cardiomyocytes (NRCMs) stained for α-actinin and imaged with a spinning disk confocal microscope to visualize the effect of SF and NAC treatments on cell shape and structure, and subsequent ImageJ analysis of cell surface area and the minimum caliper diameter of the SF or NAC-treated NRCMs, measured in pixel (px). The median is shown on the plot, with the interquartile range and unpaired one-tailed *t*-test. Source data are available in the Source Data file.

an increase in the thickness of the wall of the LV, with no such change in the case of the males (Supplementary Fig. 5b).

Thus, the DCM phenotype of male *Mlp*⁻ᐟ⁻ mice is more severe than in females. In contrast, no such differences between male and female WT animals were observed (Supplementary Fig. 5c). Moreover, two-way ANOVA analysis of certain cardiac parameters in mice of both genotypes indicated that the effect of genotype on the thickness of the LV wall was dependent on sex (Supplementary Fig. 5c).

Results described above suggest that both the level of oxidative stress and the severity of the cardiac phenotype contribute to the sexual dimorphism in *Mlp*⁻ᐟ⁻. However, the direct relation between redox hemostasis and cardiac morphology remains to be investigated next.

### Intracellular redox homeostasis modulates the morphology of cardiomyocytes

As reflected in the decrease in the total amount of RNA recovered (Fig. 4a), treatment of NRCMs with SF 5 μM and/or NAC 3 mM for 24 h led to significant loss of cells. However, upon treatment with SF for an additional 24 h, there was little further loss, while with NAC there was even a slight increase in the amount of RNA recovered (Fig. 4a). Moreover, treatment of NRCMs for 48 h with SF 1 μM or NAC 1 mM also resulted in a slight increase in the amount of RNA recovered (Supplementary Fig. 4b).

Cardiomyocytes do not divide and the potential proliferation of contaminating cardiac fibroblasts was inhibited by the presence of horse serum in combination with cytarabine, an inhibitor of mitosis, in the culture medium. Therefore, we hypothesized that the blunting of cell death and the increase in RNA content observed upon

prolonged treatment with NAC is the result not only of an altered antioxidant response, but also of remodeling of cardiomyocyte morphology. Indeed, 5 μM SF was found to lead to sarcomeric disarray and cardiomyocyte atrophy, while NAC 3 mM promoted cellular hypertrophy (Fig. 5e and Supplementary Fig. 5d). These findings indicate that intracellular redox homeostasis is involved in regulating the morphology of cardiomyocytes and supports the observed direct relation between redox hemostasis and cardiac morphology in *Mlp*⁻ᐟ⁻.

### Activation of Nrf2 improves cardiac function in male, but not female *Mlp*⁻ᐟ⁻ mice

In light of the considerable defensive antioxidative capacity of the heart, we hypothesized that patients with HF involving impairment of this capacity would benefit more from redox treatments. To test this hypothesis in mice, we exploited the higher oxidative stress observed in male *Mlp*⁻ᐟ⁻ hearts than in females.

To this end, we activated Nrf2-mediated antioxidant responses by destabilizing the Keap1-Nrf2-Cul3 complex, which ubiquitinylates Nrf2, using AZ925[45], a novel small molecule drug that blocks the BTB domain of Keap1. Such destabilization releases free Nrf2 into the cytoplasm, which then translocates into the nucleus and activates the transcription of antioxidant genes such as *Hmox1*, *Noq1*, and *Osgin1*.

Consistent with our hypothesis, daily gavage with AZ925 for 30 days and monitoring with echocardiography on days 0, 14 and 30 (Fig. 6a) revealed significant improvement of LVEF in male but not female *Mlp*⁻ᐟ⁻ mice (Fig. 6b–e). The improvement in males was associated with an increase in the thickness of the left anterior wall during

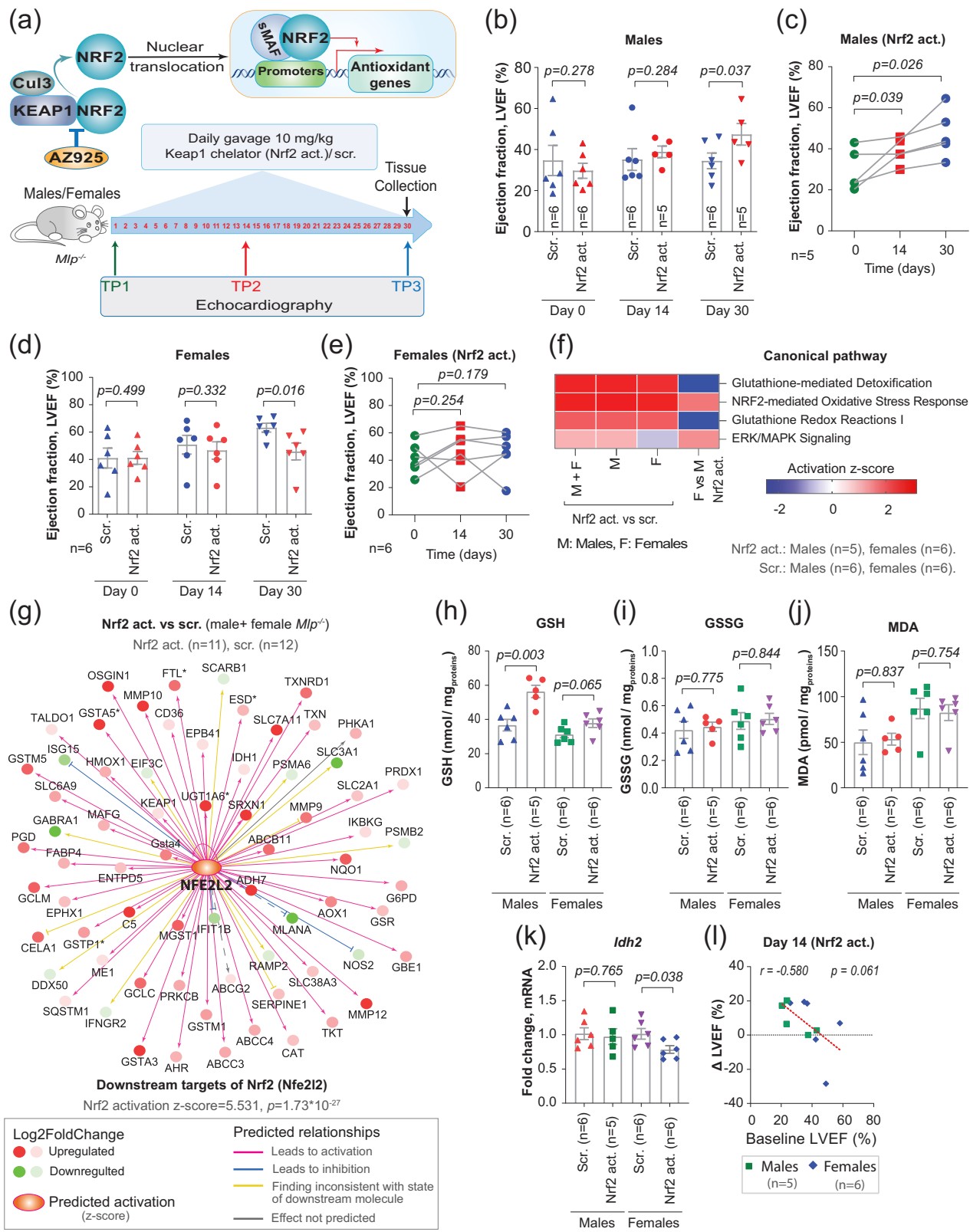

systole, whereas no such change occurred in females (Supplementary Fig. 6a).

RNA sequencing and IPA analysis suggest that AZ925 potently activated pathways involved in antioxidant responses, including glutathione-mediated detoxification, Nrf2-mediated response, and glutathione redox reactions in mice of both sexes (Fig. 6f). In addition, induction of genes encoding downstream targets of Nrf2 and its

activation in the treated animals was confirmed (Fig. 6g and Supplementary Fig. 6b).

Interestingly, AZ925-treated female $Mlp^{-/-}$ mice exhibited more potent activation of the Nrf2-mediated responses to oxidative stress, but weaker activation of glutathione-mediated pathways. The latter is further attested by a substantial increase in the levels of GSH in treated male $Mlp^{-/-}$ hearts, but with no observable changes in GSSG or MDA

**Fig. 6 | Activation of Nrf2 improves cardiac function in male, but not in female** *Mlp*[-/-] **mice. a** Experimental settings of the in vivo Keap1 chelator (Nrf2 activator) study and the proposed underlying molecular mechanism. **b–e** Left ventricle ejection fraction (LVEF) of male (**b**) and female (**d**) *Mlp*[-/-] mice treated with Nrf2 activator or with scrambled (scr.) control. The changes over time in LVEF for individual animals treated with Nrf2 activator are shown for males in (**c**) and for females in (**e**). Bars represent mean ± SEM with unpaired one-tailed *t*-test (in **b** and **d**) or paired one-tailed *t*-test (in **c** and **e**). **f, g** Selected pathways from IPA comparison analysis (in **f**), and IPA upstream regulator analysis of the downstream targets of Nrf2 with Right-Tailed Fisher's Exact Test (in **g**) on transcriptomic data from the LV of *Mlp*[-/-] mice treated with Nrf2 activator or scrambled control. **h–j** Targeted LC-MS/MS analysis of GSH, GSSG, and MDA levels in the LV of *Mlp*[-/-] treated with Nrf2 activator or scrambled control normalized to soluble proteins. **k** qPCR analysis of *Idh2* expression in the LV of *Mlp*[-/-] treated with Nrf2 activator, normalized to scrambled controls. Values in **h–k** are the mean fold change ± SEM with unpaired two-tailed *t*-test. **l** Pearson's correlation between baseline LVEF of male and female *Mlp*[-/-] mice and Δ LVEF after 14 days of treatment with Nrf2 activator. The *p*-value was derived from a two-tailed Pearson correlation analysis. Sample size is provided on all figures. Source data are available in Source Data file.

levels (Fig. 6h–j). These differences are consistent with our observation that untreated *Mlp*[-/-] females demonstrate less oxidative stress than males and an increase in the reductive equivalents (Fig. 4i–k), and therefore additional exogenous induction of antioxidative response resulted in further pronounced downregulation of *Idh2* in treated female mice (Fig. 6k).

Altogether, these findings indicate that patients with HF and insufficient cardiac antioxidant capacity would, indeed, benefit more from antioxidant treatment to a degree dependent on LV systolic function and the level of oxidative stress. However, this does not imply that female patients with heart failure would not benefit from such treatment. For example, such treatment initially improved the cardiac phenotype of female *Mlp*[-/-] mice with a low ejection fraction (EF < 40%) (Fig. 6e). Moreover, there was a general negative correlation ($r = -0.580$) between baseline LVEF and improvement in this parameter following 14 days of treatment for mice of both sexes (Fig. 6l).

## Unique epigenetic control of IDH2 expression and redox response

Experiments described above reveal that the expression of *IDH2* is regulated through antioxidative mechanisms, but the linking mechanism remains to be elucidated. Since mitochondrial 2OG/L2HG ratio is a potent modulator of the activities of TET1-3 enzymes[46], and there is a publicly available whole-genome oxidative bisulfite (OxBS) and bisulfite (BS) raw sequencing data with single-base pair resolution on *Mlp*[-/-] hearts (SRA accession ID: PRJNA327790), we examined potential involvement of epigenetic changes on the redox-modulated expression of *Idh2*. Although D2HG appears to be involved in remodeling of the cardiac epigenome and consequent mitochondrial dysfunction[47,48], no such role for cardiac L2HG has yet been reported.

Accordingly, we assessed the level and distribution of 5-methylcytosine (5mC) and 5-hydroxymethylcytosine (5hmC) in the DNA of *Mlp*[-/-] hearts by developing a tool to analyze the whole-genome OxBS and BS sequencing data. Our analysis indicated that the average methylation percentages of the total CpGs in the murine genome were 58.7% for 5mC and 4.4% for 5hmC. In *Mlp*[-/-] hearts, there were trends, albeit statistically insignificant, toward increased levels of 5mC and fewer 5hmC (Fig. 7a). Moreover, principal component analysis demonstrated that 5mC and 5hmC are distributed differently in the cardiac genomes of the transgenic and wild-type animals (Fig. 7b). At the same time, there was no global significant difference in either the level of mRNA encoding Tet1-3 or its enzymatic activity (Supplementary Fig. 7a, b). These results indicate that in cardiomyocytes cofactors of Tet1-3 enzymes, including L2HG, may modulate their activities, thereby altering the genomic profiles of 5mC and 5hmC in a manner that contributes to the development of heart failure.

Functional analysis of the distributions of 5mC and 5hmC in the cardiac genomes of *Mlp*[-/-] and WT mice revealed that *Mlp*[-/-] hearts contained lower levels of 5hmC in all functional regions, with no differences in 5mC levels (Supplementary Fig. 7c). Subsequent, Spearman's rank correlation between the levels of expression of individual genes and the proportion of methylation in each functional region showed a noteworthy positive correlation between gene expression and intronic levels of 5hmC (Fig, 7c). Moreover, computing

differentially methylated regions (DMRs) indicated that introns, followed thereafter by promoters harbored the highest number of the identified DMRs (Supplementary Fig. 7d). In addition, the genes that contained DMRs in their introns encoded a particularly high number of proteins involved in hypertrophic remodeling and regulation of redox homeostasis (Supplementary Fig. 7e).

The genes in *Mlp*[-/-] hearts that contained lower levels of 5hmC in an intron included *Idh2* (Fig. 7d), a difference that was associated with a lower level of the expression of this gene (Fig. 1g). Further, genomic *Idh2* locus was embedded in a highly 5hmC enriched region (Supplementary Fig. 7f). To examine whether the levels of Tet1-3 enzymes influence *Idh2* expression, we knocked down *Tet1-3* in NRCMs with ShRNA, but this only reduced the levels of *Tet1* and *Tet3* mRNA (Supplementary Fig. 7g). At the same time, simultaneous knockdown of Tet1/3 and L2hgdh attenuated the induction of *Idh2* caused by elevated levels of L2HG (Fig. 7e), indicating that 5hmC plays a role in mediating this induction.

Transcriptomic profiling revealed that knockdown of Tet1/3 influenced a number of pathways involved in regulating oxidative stress and mitochondrial function (Supplementary Fig. 7h). Moreover, simultaneous knockdown of Tet1/3 and L2hgdh or D2hgdh affected the enrichment in many metabolic pathways mediated by L2HG and D2HG that we observed previously (Fig. 7f). These findings suggest that Tet1/3 and the 5hmC that it generates are involved in regulating energy metabolism and redox-modulated expression of *Idh2* in cardiomyocytes (Fig. 7g).

## Discussion

The present comprehensive approach - involving a combination of metabolic, transcriptional, and epigenetic data - reveals unique features of cardiac antioxidant defenses. In essence, we show that IDH2 governs robust antioxidative defenses in mitochondria and coordinates these with other defense elements (Fig. 7g). Therapeutically, activating Nrf2 with a small therapeutic molecule boosted endogenous antioxidative capacity and thereby improved LV function in a sex-specific manner in a murine model of HF (*Mlp*[-/-]). These findings provide important insights that will help to design novel and more specific approaches to target antioxidant defense mechanisms in patients with HF.

Here, we observed that exposure to low concentrations of oxidants induced substantial antioxidative response, resulting in pronounced increase in cellular reductive equivalents. Higher concentrations of oxidants weakened this increase, until a stage where oxidants overrode the cellular antioxidative capacity and resulted in an oxidative stress-induced cell death. Even at this stage, surviving cells maintained a slightly reduced level of GSH, yet normal levels of lipid peroxidation, and it required even higher oxidative conditions to induce observable lipid peroxidation and thus oxidative damage. The latter supports previous observations that inducing cardiac oxidative damage requires extreme oxidative conditions to occur[44]. Accordingly, our findings indicated that the chronic DCM in *Mlp*[-/-] mice was associated with a mild level of oxidative stress, especially in females, which triggered an antioxidative response and consequently induced a reductive milieu,

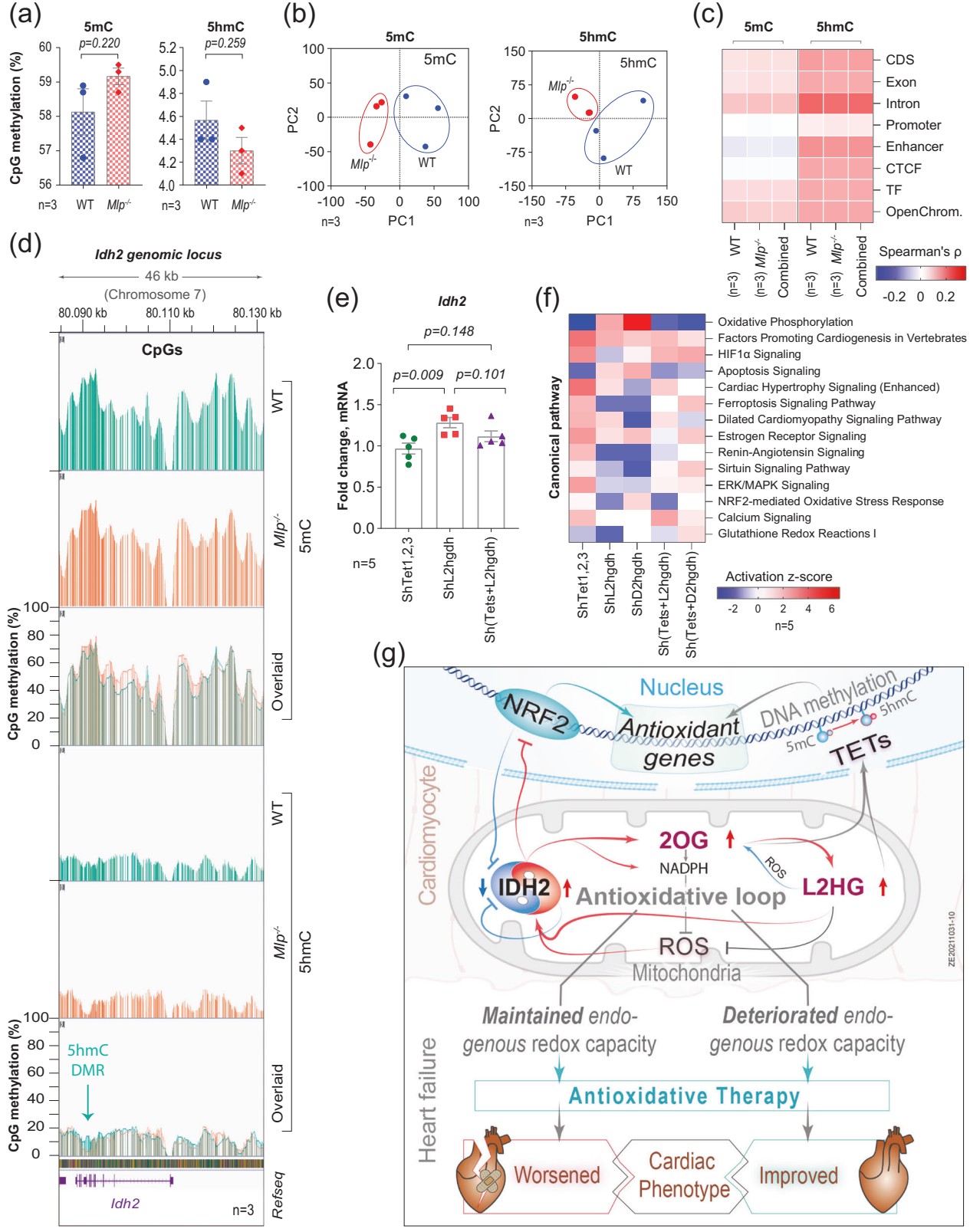

similar to the effect observed in cells exposed to low concentrations of oxidants.

In this context, both acute and chronic exposure to oxidants led to concentration- and time-dependent downregulation of *IDH2*, which initially seemed counter-productive. However, we then found that downregulation of *IDH2* was associated with an increase in total IDH2 activity. Moreover, when reductive equivalents were further increased by exogenous activation of Nrf2 with AZ925, additional downregulation of *Idh2* in female *Mlp*−/− occurred. These observations indicate that the downregulation of *IDH2* in response to mild oxidative stress is regulated through antioxidative mechanisms, which may reflect the presence of an antithetical regulation through a negative feedback loop that limits overwhelming cells with reductive equivalents. Further evidence for this proposal was provided by our findings

**Fig. 7 | Unique epigenetic control of Idh2 expression and redox response.**
**a** Percentages of whole genome 5mC or 5hmC-methylated CpGs in the myocardium of *Mlp*⁻/⁻ mice. Bars represent mean ± SEM with unpaired two-tailed *t*-test. **b** Principal component (PC) analysis of the distribution of whole genome 5mC and 5hmC (n = 3 in each group). **c** Spearman correlation between the expression of individual genes and the proportion of methylation in each adjacent functional region (CDS, coding regions; CTCF, transcriptional repressor CTCF binding sites; TF, transcription factors binding sites.). Values presented are an average of three biological replicates in WT and *Mlp*⁻/⁻, or 6 replicates in combined WT and *Mlp*⁻/⁻. **d** Screenshot from the Integrative Genomic Viewer for actual methylation levels in *Idh2* locus in chromosome 7 of murine genome. **e** qPCR analysis of *Idh2* expression in NRCMs transduced with ShRNA targeting *Tet1-3* and/or *L2hgdh*. The values are normalized to their respective scrambled control. Bars represent mean ± SEM with unpaired two-tailed *t*-test. **f** Selected pathways from IPA comparison analysis on transcriptomic data of NRCMs transduced with ShRNA targeting *Tet1-3*, *L2hgdh*, *D2hgdh*, or *Tet1-3* combined with *L2hgdh* or *D2hgdh*. **g** A summary of the proposed molecular mechanism. Source data are available in Source Data file.

that in vitro activation of the NRF2 pathway with oxidants in cardiomyocytes downregulated *IDH2*, whereas deactivation of this same pathway by NAC or L2HG upregulated *IDH2*.

The increase in IDH2 activity elevated intracellular levels of 2OG and L2HG, which also mitigate oxidative stress[39]. In addition, knockdown of Tet1/3 provides direct evidence that activation of antioxidative pathways and the marked induction of IDH2 expression by L2HG are mediated, at least in part, through epigenetic modifications. These findings suggest that 5hmC, 5mC and L2HG play critical roles in epigenetic processes that link oxidative stress to mitochondrial function, with intronic 5hmC as a fundamental regulator of gene expression in the heart. Altogether, these observations indicate that IDH2, 2OG and L2HG participate in an antioxidative feedforward loop that not only prevents oxidative damage but also senses and regulates the antioxidative status of mitochondria (Fig. 7g). Consequently, the pronounced downregulation of *IDH2* in eccentric hypertrophic hearts may reflect the extensive activity of this cycle. Moreover, this unique antioxidative defense mechanism is not restricted to the heart but also expected to play an important role in other metabolically active organs that express IDH2 and NNT, such as in liver, kidney, and spleen[15].

Male *Mlp*⁻/⁻ hearts appeared to exhibit more oxidative stress than females, but not to the extent that overrides the capacity of the antioxidative system, which may explain the normal levels of lipid peroxidation observed in male hearts. However, scavenging oxidative species is metabolically costly, and adds to the increased energy demand in HF[30]. Therefore, alleviating oxidative stress by exogenous activation of NRF2 with AZ925 might have underlain improved cardiac functions in males.

Moreover, the role of intracellular redox homeostasis in regulating the morphology of cardiomyocytes, which we reported here, might also represent another contributor to the observed improved cardiac function in males. AZ925-treated males exhibited an increase in left ventricle wall thickness, which alleviated diastolic dysfunction in these mice. These findings support previous indirect observations that antioxidants, such as NAC, activate the ERK pathway[49,50], which in turn is expected to promote concentric hypertrophy[51].

The impact of endogenous antioxidative capacity is not only the result of a robust antioxidative system or of potential sex dimorphism in redox modulation, but also of variations of other genetic traits. For example, promoter polymorphisms resulting in low HMOX1 expression and activity in certain individuals are associated with incident arterial hypertension and with increased mortality[52]. Therefore, our study urges considerations of individual and sex-dependent variations in endogenous antioxidative capacity when designing personalized redox treatments. However, it does not challenge the general notion of a crucial role for oxidative stress in the development of HF. Excess or inadequate exogenous antioxidants may perturb the delicate balance of redox homeostasis as an off-target effect[53], such as the herein observed in females. Assessment of this capacity may help predict the chances of success when subjecting cardiac oxidative stress or, perhaps, certain cancers as well to antioxidant treatment. In this context, application of such individualized treatment in the clinic would require noninvasive biomarkers. As described here, L2HG and 2OG may represent promising novel candidates for such biomarkers and, in fact,

elevated levels of 2OG have been detected in the plasma of patients with DCM[54].

Our data regarding 5hmC remains correlative due to current technical challenges in available technologies; this represents a limitation to our approach. We demonstrate here that during cardiac remodeling, 5hmC in specific regions of genes is a potent regulator of gene expression. Unfortunately, because of its extremely low levels in the genome, elucidating this regulatory role of 5hmC in greater detail would require high sequencing coverage ( > 200X) with relatively large sample size, which is technically challenging at present. Hopefully, emerging third-generation sequencing and other future advances will facilitate such investigations in the future.

## Methods
All described experiments were carried out according to good scientific practice, and no data points were excluded from analyzes. All experiments performed on human materials or involved animals complied with all relevant ethical regulations as detailed below.

### Human samples
Samples from explanted human hearts with familial DCM and the matched controls were obtained from Sydney Heart Bank. The donors of the control hearts had normal ECG and ventricular function with no previous history of cardiac diseases. All samples were frozen in liquid nitrogen upon collection. General functional characteristics of these samples are provided in Supplementary Table 1 together with references to previous studies that also used these samples. The use of human samples was approved by the local ethics committee in Stockholm (2015/559-31/2) and Human Research Ethics Committees at the University of Sydney (2016/7326), and St Vincent's Hospital (H03/118). All other data from patients utilized in this work and their corresponding characteristics were publicly available[26–28]. The source of the data was cited wherever this data was used.

### Animal experiments
Animal experiments were performed in accordance with European ethical regulation (Directive 2010/63/EU) and approved by the local animal ethics committee of Linköping (permit numbers S43-15, 1369, and 2713) and Göteborg (permit 1852-2018) in Sweden, and the responsible government agency of Unterfranken (RUF-55.2.2-2532-2-659) in Germany. Animals were housed on a 12 h light/12 h dark cycle with free access to chow and water, in 45–65% humidity and ambient temperature at 20–24 °C.

The background of the muscle lim protein-deficient mice (*Mlp*⁻/⁻), also known as cysteine and glycine-rich protein 3 (*Csrp3*⁻/⁻), is a hybrid cross of original 129/Sv background[29] and C57BL/6 N strain. The mice were bred in-house for multiple generations. Heterozygous males and females were bred to obtain both WT and *Mlp*⁻/⁻ animals. Both male and female mice were utilized for experiments at the age of 10-14 weeks. If animals of other ages were used, a description was mentioned in the figure legend or in the Source Data file.

Rats from Sprague Dawley strain were purchased from Charles River Laboratories and bred in-house to obtain neonatal pups, which were dissected at the age of 3 days to harvest their hearts.

## In vivo KEAP1 inhibitor study

Novel Keap1 inhibitor (AZ925)[45] and the vehicle control were formulated and provided by AstraZeneca- MedImmune. The study was designed and performed in accordance with the PREPARE guidelines[55]. Utilized dose (10 mg/kg) was adapted from previous unpublished studies performed during the development of AZ925. We validated this dose through a pilot study, by administrating 10 mg/kg of AZ925 by a single oral gavage in two mice. We terminated the study 9 h post-administration and assessed the induction of Nrf2- downstream target genes (*Hmox1*, *Nqo1* and *Osgin1*) by qPCR analysis in the heart, liver, brain, and kidneys. We observed a robust induction of these genes (1.5-7-fold increase) in all examined organs (Supplementary Fig. 8a). We also observed a robust increase in the protein level of Nrf2 and Hmox1 in the myocardium of these animals (Supplementary Fig. 8b).

*Mlp*[−/−] animals at the age of 10 weeks were treated by a daily gavage with 10 mg/kg of AZ925 or vehicle control for 30 days. Both male and female mice were included in the study, with 6 animals in each of the 4 groups (Keap1 inhibitor/scrambled control-Males/females). The mice were randomized into different treatment groups based on their body weight. Echocardiography was performed at days 0, 14 and 30 (see Fig. 6a for experimental scheme). One male in the Keap1 inhibitor group died immediately after first gavage, but all other animals survived to the end of the study, with no observable changes in their behaviors or body weight. The investigators were blinded to the different groups during experimental procedures.

## Echocardiography

Echocardiography was performed with the ultrasound Vevo 3100 System, with the MX550D probe (Visualsonics, Canada). Animals were initially anaesthetized with 3% isoflurane (Forene®) mixed to oxygen (Oc-P50, Sysmed) and then maintained at 1.5% isoflurane during the examination. The animals were kept on a Physio Plate (Visualsonics, Canada) during whole procedure to maintain normal body temperature, and to monitor ECG and respiration. Body temperature was monitored by an inserted rectal probe. An external heat lamp was used when it was needed. Left ventricle parasternal long and short axis projections were captured in both B-Mode and motion-mode (M-MODE). Short axis midventricular level, with a clear projection of the papillary muscles, was captured after rotating the probe ~90° clockwise from the long-axis view. By using the tool *LV Analysis* in the Vevolab software (version 5.5.0), several cardiac parameters were evaluated from the M-mode's short-axis view, in at least three beats and averaged. Among the main obtained parameters were left ventricular internal diameter (LVID) in systole (s) and diastole (d), LV's ejection fraction, and the left ventricular posterior and anterior wall thickness (LVPW, LVAW) in systole and diastole. The thickness of anterior or posterior LV wall was normalized by dividing the inferred wall thickness in diastole or systole to LV diameter in diastole or systole, respectively. The investigators were blinded to the groups during experimental procedures.

## Tissue collection

Mice hearts were excised immediately after cervical dislocation, washed with cold PBS to remove blood, and then weighed. The left ventricles were quickly separated and flash-frozen in liquid nitrogen. Mice included in the Keap1 inhibitor AZ925 study were anaesthetized (10 ± 3 h post-administration of the last dose) with 3% isoflurane (Forene®) and then the hearts were excised and processed as described above.

## Tissue grinding and homogenization

Frozen human and murine left ventricle cardiac tissues were grinded and homogenized in Cellcrusher (Cellcrusher, Irland) in liquid nitrogen.

## Experiment on isolated neonatal rat cardiomyocytes (NRCMs)

**NRCMs isolation.** Hearts from 3 days old Sprague-Dawley rats were excised after decapitation and placed immediately in ice-cold preservative buffer (concentrations in mM: NaCl 13, KCl 0.54, HEPES 2.5, MgCl$_2$ 0.05, NaH$_2$PO$_4$ 0.04, Glucose 22.20, pH was adjusted to 7.4 with NaOH). Ventricles were separated from atria and minced into small pieces. Tissue pieces were dissociated into single cells at 37 °C by utilizing the gentleMACS Octo Dissociator with Heaters (Miltenyi Biotec, Bergisch Gladbach, Germany) using Neonatal Heart Dissociation Kit (Miltenyi Biotec) according to the manufacturer's instructions. The single cell suspension was diluted with 7.5 ml of cardiomyocytes plating medium (CM plating medium: DMEM (high glucose): M199 (4:1), supplemented with 5% horse serum, 2.5% fetal bovine serum, 20 mM HEPES and 100 U/ml penicillin/streptomycin (Thermo Fisher Scientific, Waltham, Massachusetts, USA)) at room temperature (RT) to inactivate digestive enzymatic mix and to gradually bring the temperature of the cell suspension down, and then passed through MACS Smart Strainers (70 μM, Miltenyi Biotec) to remove tissue residues. Then the red blood cells were lysed with Red Blood Cell Lysis Solution (Miltenyi Biotec). The lysis step was performed at RT, but subsequent enzymatic treatment was done in ice-cold PBS buffer. After lysing red blood cells, cardiomyocytes were enriched in the cellular suspension by allowing non-cardiomyocytes to adhere to a surface of 10 cm plates (Corning) by pre-plating them for 75 min at 37°C in CM plating medium. Enriched cardiomyocytes were counted with TC20 automated cell counter (BioRad, US) and plated in either 6-well or 24-well plates (Corning). 1.25 ×10$^6$ or 3.33 ×10$^5$ live cells were plated in each well of the 6-well or 24-well plates, respectively. Before plating the cells, the plates were gelatinized with 0.15% gelatine solution for 1 h at 37°C. Plated cells were maintained in incubators at 37°C with 5% CO$_2$.

**Treating NRCMs with redox active compounds.** Enriched NRCMs were plated in CM plating medium in either 6-well plate for microscopic florescence imaging, or in 24-well plate for RNA extraction. After 24 h of incubation, the cells were washed once with PBS to removed dead cells and incubated for another 48 h with CM maintenance medium (similar composition of the CM plating medium but supplemented with 2.5 μM of the mitotic inhibitor cytarabine, (Sigma-Aldrich, Germany)). Cells were then treated with hydrogen peroxide, Sulforaphane or N-Acetyl-L-cysteine (Sigma-Aldrich, Germany).

Hydrogen peroxide (H$_2$O$_2$): A stock solution (50 mM) was prepared in water and diluted with NRCM maintenance medium to concentrations of 25, 50, 100, 200 or 300 μM. NRCMs were treated with H$_2$O$_2$ for 24 h, thereafter, the culturing medium was replaced with a fresh culturing medium without H$_2$O$_2$ for another 24 h. Cells were collected at different time-points at 6, 24 or 48 h from the start of the treatment (see Fig. 2a for the experimental scheme).

DL-Sulforaphane (SF) and N-Acetyl-L-cysteine (NAC): SF powder was dissolved in DMSO to obtain a 0.1 mg/μl solution, which was aliquoted and stored at −20 °C. This solution was diluted with water to get 500 μM stock solution, which was further diluted with NRCMs maintenance medium to a final concentration of 5 μM. SF solutions were freshly prepared for every experiment from the frozen stock. Fresh stock solution of 100 mM NAC was prepared from NAC powder and diluted with NRCM maintenance medium to a final concentration of 3 mM.

Used concentrations for SF (5 μM) and NAC (3 mM) were selected after testing multiple concentrations in NRCM for 6, 24 or 48 h, and assessing cell death through monitoring recovered amount of total RNA. For SF we tested 1, 5 and 7.5 μM. Whereas for NAC, we tested 1, 3 and 6 mM. Doses higher than 5 μM for SF and 3 mM for NAC induced substantial loss in NRCMs (Supplementary fig. 4b). Therefore, we used 5 μM for SF and 3 mM for NAC in most experiments, which are also commonly used in redox studies.

NRCMs were treated with SF or NAC for 48 h or 72 h. The maintenance medium was replaced every 24 h with fresh supplements of SF or NAC. For RNA expression analysis, cells were collected after 6, 24 or 48 h of treatment (see Fig. 4A for the experimental scheme). For GSH/GSSG measurement, cells treated with 1, 2, 5, 7.5, and 10 μM of SF in 24-well plates were harvested. Meanwhile, to measure MDA, NRCMs were treated with 1, 2, 5, and 10 μM of SF in 1.5 ml of maintenance media per well in 6-well plates. For immunofluorescence, cells were treated for 72 h and fixed with 4% formalin for 10 min at 37°C, then washed twice with PBS.

**Transducing NRCMs with ShRNA targeting L2hgdh, D2hgdh or Tet1-3.** All constructs were AAV based vectors packaged into AAV9 viruses (VectorBuilder) where small-RNA expression was driven by the Pol III promoter. Enhanced green fluorescent protein (EGFP) transcript was added to the construct under the cytomegalovirus (CMV) promoter.

Two experimentally validated ShRNA sequences targeting each of the rat's *Tet1-3* were obtained from published literature (Supplementary Table 2). To achieve simultaneous knockdown of all *Tet1-3* in individual cardiomyocytes, three ShRNA sequences, each targeting one of the three *Tet1-3* mRNA were constructed together in one vector, resulting in two vectors as it is shown in the vector map available with the provided link in Supplementary Table 2. For targeting rat's *L2hgdh* and *D2hgdh* mRNA, we tested multiple predicted ShRNA sequences. Each designed vector contained two or three sequences targeting either *L2hgdh* or *D2hgdh*, to increase the efficiency of finding a positive hit (Supplementary Table 2). The vector that achieved more than 50% knockdown of the target mRNA at a dose of 10000 multiplicity of infection (MOI) was selected for downstream experiments and their ShRNA sequences are shown in Supplementary Table 2, and their full map can be obtained through the provided link. Several viral titers were tested (1000-20000 MOI), where we found that 10000 MOI gave the optimum transduction efficacy, assessed through examination under florescence microscope on day 4 of transduction.

Enriched cardiomyocytes were plated in CM plating medium in either 6-well plate for protein extraction, or in 24-well plate for RNA extraction. After 24 h of plating, cells were washed twice with PBS, and transduced with AAV9 in CM maintenance medium. To increase the transduction efficiency, cells were kept in the same culturing medium containing viruses for 3 days. After 3 days, one volume of fresh maintenance medium was added to each well. Then, on days 4 and 5 the medium was replaced with a fresh maintenance medium. Cells were collected after 6 days of transduction.

**Experiments in human ventricular induced pluripotent stem cell-derived cardiomyocytes (hiPS-CM)**
iPSC-lines from two healthy donors[56] were cultured on Geltrex-coated cell culture dishes without feeder cells in the chemically defined medium E8 (Life Technologies).

The differentiation of iPSC into iPSC-derived cardiomyocytes (iPSC-CM) was achieved through manipulation of the Wnt signaling pathway. iPSCs were cultured on 12-well plates until they reached 70 to 90% confluency. The culture medium was then changed to cardiac differentiation media, composed of RPMI1640 GlutaMAX (Thermo Fisher Scientific), human recombinant albumin, and L-ascorbic acid 2-phosphate, supplemented with the Wnt signaling activator CHIR99021 (4 μmol/L, Millipore). After two days, the medium was switched to cardiac differentiation media containing the Wnt inhibitor IWP2 (5 μmol/L, Millipore) for 48 h. Starting from day 8, the cells were maintained in a cardio culture medium (RPMI1640 GlutaMAX with 1x B27 with insulin, Thermo Fisher Scientific), with medium changes every 2 to 3 days. iPSC-CM were subjected to metabolic selection using 4 mmol/L lactate as a source of carbon for 4 to 5 days. Experiments were performed after long-term culturing of the cells

for 90 days. One week prior to measurements, the cells were re-plated onto Geltrex-coated 24-well plates (Sarstedt, 200k/well) through trypsinization for 5 min at 37 °C. The purity of the differentiated cells was assessed by flow cytometry analysis (> 90% cardiac TNT + ). Four differentiation experiments into ventricular iPSC-CMs were used. Plated hiPSC-CM in 24-well plates were treated with SF (1, 2, 5 and 10 μM) in 0.5 ml of RPMI. After 24 h, the medium was changed with 0.5 ml of fresh RPMI medium containing same concentrations of SF. After an additional 24 h, the cells were harvested for GSH/GSSG measurement.

All procedures conducted in this study adhered to the principles outlined in the Declaration of Helsinki and received approval from the local ethics committee of the University Medicine of Göttingen (Az-10/9/15). Informed consent was signed by all tissue donors.

**Experiments in human foreskin fibroblast (hFF1) cell line**
HFF1 cells (ATCC, SCRC-1041) were cultured in DMEM high glucose (#41966), supplemented with 15% fetal bovine serum and 100 U/ml penicillin/streptomycin (Thermo Fisher Scientific, Waltham, Massachusetts, USA). The cells were cultured in 24-well plates or 10 cm dishes and treated with SF (1, 2, 5 and 10 μM) or NAC (3 mM) for 24 h after reaching 90-100% confluency. The culturing medium (0.5 ml per well for the 24-well plate and 10 ml for the 10-cm dish) was replaced with fresh supplements of SF or NAC after 24 h of treatment. After an additional 24 h, the cells were harvested for GSH/GSSG measurement or MDA measurement.

**Metabolites analyzes**
**L-malate, succinate, lactate, and L/D2HG.** L-malate, succinate, lactate and L/D2HG were analyzed by a targeted mass spectrometric approach. We developed an analytical method that enables extraction and separation of the two 2HG enantiomers in heart tissues and NRCM cells, based on a previously published method for analyzing L/D2HG in cancer tissues[57]. The method utilizes chemical derivatization of L/D2HG by N-(p-toluenesulfonyl)-L-phenylalanyl chloride (TSPC). TSPC was reported to enable higher sensitivity for L/D2HG detection and quantification than DATAN based method[57].

Metabolites extraction from heart tissues: 10 mg of homogenized and grinded tissues were weighed in a 2 ml Eppendorf tubes (Sarstedt). Then 100 μl of deionized water containing 0.5 nmol of heavy isotope-labeled L/D2HG internal standard (IS) (Deuterated RS2HG, Cambridge Isotope Laboratories) was added to each tube, with a 0.5 mm metallic ball (Qiagen). The tubes were immediately homogenized by the TissueLyser LT (Qiagen, Germany) at 4 °C for 2 min at 50 Hz and then spun quickly before placing them back on dry ice. After the tubes' content got completely frozen, the tubes were allowed to thaw partially, and then homogenized again for 2 min at 50 Hz. This cycle of freezing, thawing, homogenization, and brief spinning was repeated for the total of 4 times to break down all cellular and mitochondrial walls. The temperature was always kept below 4 °C during the whole procedure to minimize any enzymatic reaction of the metabolites. After last homogenization cycle, 400 μl of cold pure methanol (−80 °C) were added to each tube (to obtain 80% MeOH aqueous solution) to precipitate proteins, and the tubes were homogenized again for another 2 min at 50 Hz, and then incubated on dry ice for 5 min. As negative controls, 3 tubes containing 100 μl of deionized water with 0.5 nmol of deuterated-RS2HG, L2HG or D2HG standards underwent the same described procedure with the samples.

Metabolites extraction from NRCMs: Transduced NRCMs with ShRNA targeting *L2hgdh*, *D2hgdh* or scramble control were trypsinized with TripleE (Invitrogen), centrifuged at 6000 g for 10 min at 4 °C to remove the supernatant, and then stored at −80 °C. Then 40 μl of water containing 0.5 nmol of the internal standard (deuterated-RS2HG) was added to each pellet of cells. The pellets were thawed on ice and then frozen again for a total of 10 cycles, with vortexing in between, to

completely lyse the cells. Then 160 µl of −80 °C methanol was added. The mixture was vortexed and incubated on dry ice for 5 min.

Derivatization with TSPC: TSPC was dissolved in acetonitrile (ACN) for a stock concentration of 62.5 mM, and then aliquoted and stored at −80 °C.

The 80% MeOH aqueous extracts from tissue and NRCMs were centrifuged at 4 °C at the maximum speed of 20800 g for 10 min. The supernatant was transferred to a new tube and the solvent was completely evaporated under vacuum at 45 °C (SpeedVac Savant SPD2010). The dried metabolites' residues were dissolved in 100 µl of acetonitrile (ACN) by vigorous vortexing. Then another 60 µl of ACN containing 2 µl of pyridine was added. The derivatization reaction was initiated by adding 3.2 µl of 62.5 mM TSPC and incubating the tubes at 25 °C for 10 min. The solvent was then evaporated completely under vacuum at 45 °C (SpeedVac Savant SPD2010). The resulting precipitants were dissolved with 15 µl 50% ACN/water and centrifuged at a maximum speed of 20800 g for 10 min. ~14 µl were transferred to a 0.2 mL PCR strips and stored at −80 °C for the mass spectrometric analysis.

**GSH and GSSG.** We developed a targeted MS/MS approach to simultaneously analyze reduced and oxidized glutathione in cardiac tissue samples and in cultured cells.

Extraction solution for glutathione was composed of pure water containing 5 mM of N-Ethylmaleimide (NEM) to derivatize GSH, and two internal standards, i.e., 0.05 mM of GSH-d5 (Toronto Research Chemical, G597953) and 4 nM of GSSG-$(^{[13]}C_4{}^{[15]}N_2)$ (Toronto Research Chemical, G597972). We used two different isotopic labeled internal standards to correct for auto-oxidation of endogenous GSH and auto-reduction of endogenous GSSG.

Grinded frozen cardiac tissues (5 mg) were mixed with 100 µl of the extraction solution in 2 ml-round bottom Eppendorf tubes. A 0.5 mm metallic ball (Qiagen) was added, and the tubes were frozen on dried ice and then homogenized by TissueLyser LT (Qiagen, Germany) at 4 °C for 2 min at 50 Hz. The freezing and homogenizing cycle was repeated for a total of 3 cycles. After the last homogenization step, tubes were incubated at room temperature for 20 min to increase the efficiency of GSH derivatization by NEM. After centrifugating at maximum speed for 5 min, 2.5 µl of the tissue lysate were taken for protein quantification with BCA assay, and then 400 µl of pure methanol was added to precipitate proteins. Precipitated proteins were removed by centrifugating twice at 4 °C at the maximum speed of 20800 g for 10 min. The supernatant was transferred to a new tube and dried under vacuum at 45 °C (SpeedVac Savant SPD2010), and the resulting precipitants were then dissolved in 10 µl of solvent A of the running buffer for chromatographic separation prior to MS/MS analysis, as described below.

NRCMs, hiPSC-CM or HFF1 cells, cultured in 24-well plates and treated two times with SF or NAC for 2 days, were washed twice with PBS and then lysed with 100 µl of glutathione extracting solution by freezing the whole plate and then thawing it with quick sonication in an ultrasonic water-path at room temperature. The freezing and thawing cycle was repeated for a total of 3 rounds. Cellular lysates were then transferred to 1.5 ml Eppendorf tubes and the well was washed with two portions of 200 µl of pure methanol, combined cellular lysates, and then processed as described above. Residual NRCMs and hiPSC-CMs were removed from the plate surface by a cell-scraper (Sarstedt, 83.1832) after adding the first portion of ethanol. Precipitated proteins were dissolved in 10 µl of 1 M NaOH at 60 °C for 30 min and used for quantifying total proteins by BCA assay. The total protein amounts quantified from four wells for each treatment were averaged and subsequently used for normalization.

**Malondialdehyde (MDA).** Total MDA was quantified in frozen cardiac tissues by a targeted MS/MS approach adapted from Mendonça et. al., 2017[58], which was originally developed to analyze MDA in plasma samples. For the internal standard of MDA, we could not find a commercially available isotopic labeled MDA. Therefore, we synthetized D2-MDA from 1,1,3,3-Tetraethoxypropane-1,3-D2 (D2-TEP) (Toronto Research Chemicals, T292955) by acidic hydrolysis. For this purpose, 30 µM of D2-TEP in 0.1 M of HCl was incubated at 40 °C for 40 min[58].

Grinded frozen cardiac tissues (1–2 mg) were mixed with 50 µl of 1 M NaOH in 250 µl-PCR tubes. The tubes were centrifuged at maximum speed for 1 min to ensure that all grinded tissue particles were submerged in NaOH to prevent sample oxidation. Then the tubes were heated to 60 °C for 30 min in a thermocycler, and then cooled to 4 °C, followed by a quick sonication in an ultrasonic water-path, which resulted in a clear yellowish lysate. 2.5 µl of the lysate was taken for protein quantification, and then 10 µl of 30 µM D2-MDA and 150 µl of 20% (W/V) trichloroacetic acid (TCA) solation were added to each tube to precipitate proteins. After centrifugation at the maximum speed of 20800 g for 5 min, 190 µl of the supernatant was moved to a new 1.5-ml Eppendorf tube containing 19 µl of 5 mM 2,4-Dinitrophenylhydrazine (DNPH) dissolved in 20% TCA. Tubes were then incubated for 10 min at room temperature to derivatize MDA before stopping the reaction by adding 22 µl of 10 M NaOH. The resulting MDA-DNPH was then extracted twice with 250 µl of a mixture of cyclohexane: toluene (1:1 v/ v). The solvent was then dried under vacuum at 45 °C (SpeedVac Savant SPD2010), and the resulting precipitants were dissolved in 10 µl of solvent A of the running buffer for chromatographic separation, prior to MS/MS analysis, as described below.

It is important to note here that the quantities of reagents described above were optimized for 1–2 mg of tissue. Observing a turbidity in the NaOH tissue lysate indicates excess amounts of tissues, which results in sample oxidation and biased results.

NRCMs cultured in 6-well plates, or HFF1 cells cultured in 10 cm dishes, which were treated with SF or NAC, were trypsinized (TrypLE™ Express, ThermoFisher Scientific, 12604013) and centrifugated at 650 g. 5 mM of Butylated hydroxytoluene (BHT, stock solution 500 MM dissolved in acetonitrile) was added to the trypsinizing solution to prevent sample oxidation. The resulting pellet was processed as described above to derivatize and extract MDA.

**Mass spectrometric analysis.** Malate, succinate, lactate and L/D2HG: Two µl of samples were injected in an Ultimate™ 3000 UPLC coupled with a heated electrospray ion source to an Orbitrap™ Fusion Lumos™ tribrid mass spectrometer (ThermoFisher Scientific). The chromatographic separation was achieved using a 25 cm long (2.1 mm i.d., 5 µm particle size) Inertsil™ ODS-3 column (GL Sciences, Tokyo, Japan) at 35 °C. Formic acid in water (0.1%, as solvent A) and a 50:50 mixture of acetonitrile and methanol (as solvent B) were employed as the mobile phase. A gradient of 3 min 30% B, 7 min 30–70% B, 15 min 70% B, 1 min 70–30% B, and 14 min 30% B was used at a flow rate of 200 µl/min.

The mass spectrometer was set to acquire tandem mass spectra in small molecule mode with negative ion detection. The precursors defined in an inclusion mass list were quadrupole isolated aiming at minimum six points across the peak with $m/z$ 0.7 isolation width in the ranging from $m/z$ 50 to 500 at a resolution of R = 30,000 (at $m/z$ 200) targeting $5 \times 10^4$ ions for maximum injection time of 54 ms, using higher energy collision dissociation (HCD) fragmentations at 27% normalized collision energy in 2 s cycle time.

GSH, GSSG and MDA: The GSH and MDA samples were dissolved in 10 µL of solvent A. One µl of samples was injected in an Ultimate™ 3000 UPLC coupled with a heated electrospray ion source to an Orbitrap™ Fusion Lumos™ tribrid mass spectrometer (Thermo Fisher Scientific). The chromatographic separation was achieved using a 25 cm long (2.1 mm i.d., 5 µm particle size) Inertsil™ ODS-3 column (GL Sciences, Tokyo, Japan) at 200 µL/min flow rate at 35 °C for GSH/GSSG analysis, and a 15 cm long (1 mm i.d., 1.7 µm particle size) Kintex EVO C18 core-shell column (Phenomenex, USA) at 75 µl/min flow rate at

60 °C for analyzing MDA. Formic acid in water (0.1%, as solvent A) and in methanol (as solvent B) were employed as the mobile phase. A gradient of 4 min 50% B, 2 min 50–99% B, 1 min 99% B, 2 min 0% B was used.

The mass spectrometer was set to acquire tandem mass spectra in small molecule mode with positive ion detection. The precursors defined in an inclusion mass list were quadrupole isolated aiming at minimum nine points across the peak with $m/z$ 0.7 isolation width in the ranging from $m/z$ 150 to 700 at a resolution of $R = 30{,}000$ (at $m/z$ 200) targeting $5 \times 10^4$ ions for maximum injection time of 54 ms, using higher energy collision dissociation (HCD) fragmentations at 30% normalized collision energy in 2 s cycle time.

Skyline v20.1.0.155 (MacCoss Lab, Dept. of Genome Sciences, University of Washington[59]) was used for data analysis by importing raw data files with target masses ($\Delta_{mass} = 3$ Da). The ion match tolerance was set to 0.05 $m/z$, allowing automatic selection of all matching transitions. The imported data was manually controlled for peak boundaries. The peak areas were extracted and used for quantitative comparisons. When several fragments were detected from the precursor ion, the highest intensity ion was used for the quantitative comparisons (Supplementary Table 3).

### Mass spectrometric data analysis and calculations
**Malate, succinate, lactate and L/D2HG.** The concentration of L2HG or D2HG in the LV myocardium of WT and $Mlp^{-/-}$ mice was calculated by dividing the peak area of the 155 $m/z$ fragments of L/D 2HG-TSPC by the corresponding peak area of the spiked-in L/D2HG-TSPC_heavy. The resulting values were then normalized to tissue weight. The levels of L/D2HG in transduced NRCMs with ShL2hgdh or ShD2hgdh were compared to the scrambled control-transduced cells by dividing the peak area of the 318 $m/z$ fragment of L/D 2HG-TSPC by the corresponding peak area of spiked-in L/D2HG-TSPC_heavy. The levels of L-malate (MA), succinate (SA), and lactate in cardiac tissues $Mlp^{-/-}$ mice were compared to WT by dividing the peak area of the 155 $m/z$ fragment of MA-TSPC, 73 $m/z$ fragment of succinate or 71 $m/z$ fragment of MA-TSPC to the peak area of the 155 $m/z$ fragment of spiked-in D2HG-TSPC_heavy, and then normalized to the tissue weight.

**GSH, GSSG and MDA.** The concentration of GSH, GSSG or MDA in the LV myocardium of WT and $Mlp^{-/-}$ mice and in the treated cells were calculated by dividing the peak area of the 201, 355, 235 $m/z$ fragment or precursor ion of GSH-NEM, GSSG or MDA-DNPH, respectively, by the corresponding peak area of the spiked-in isotopic labeled internal standard, i.e., GSH_D5-NEM (201 $m/z$), GSSG ([13]C4[15]N2) (361 $m/z$) or MDA-D2-DNPH (191 $m/z$), respectively. The resulting values were then normalized to the total protein amount.

The chromatographic separation of GSH-NEM generated two adjacent peaks due to the formation of diastereomers. We selected the first peak that had faster elution time to be used for the quantification. The concentrations of GSH, GSSG, and MDA were determined by using standard materials of these compounds and subjecting them to the same preparation procedures described above.

**2-oxoglutarate (2OG) quantification.** The levels of 2OG in the LV myocardium of WT and $Mlp^{-/-}$ male mice were measured by utilizing the colorimetric 2-oxoglutarate Assay Kit MAK054 (Sigma-Aldrich), according to the manufacturer's instructions with minor modifications. In brief, 20 mg of homogenized and grinded frozen heart tissues were weighed in a 2 ml Eppendorf tube (Sarstedt). Then 100 μl of the assay buffer was added to each tube, with a 0.5 mm metallic ball (Qiagen). The tubes were immediately homogenized by the TissueLyser LT (Qiagen, Germany) at 4 °C for 2 min at 50 Hz and then spun quickly before freezing them on dry ice. The tubes were then allowed to thaw partially, and then homogenized again for 2 min at 50 Hz. This cycle of freezing, thawing, homogenization, and brief spinning was repeated for the total of 4 times to break down all cellular and mitochondrial walls. The temperature was always kept below 4 °C during the whole procedure to minimize any enzymatic reaction of the metabolites. Insoluble particles were removed by centrifuging at 4 °C at the maximum speed of 20800 g for 10 min, and then the supernatant was transferred to a new tube, and the centrifugation step was repeated for another 15 min. Finally, 50 μl of the supernatant was transferred to a 0.2 mL PCR strips and stored at −80ºC. Two μl of this tissue's extract was utilized for estimating total protein concentration by BCA assay.

To measure the levels of 2OG, we obtained a standard curve for 2OG with the concentrations of 0, 0.5, 1, 1.5, 2, 2.5 nmol/well. Samples and standards were measured in duplicates, in a total volume of 25 μl per well in a 384-well plate (Corning). The absorption was measured at 570 nm by SpectraMax® i3 (Molecular Devices, US). The levels of 2OG were normalized to the total protein content.

**Cell viability assay.** Cell viability of hFF1 cells treated with increasing concentrations of SF were measured by utilizing the fluorometric CellTiter-Fluor™ Cell Viability Assay Kit G6080 (Promega), according to the manufacturer's instructions with minor modifications. The cells were seeded in 384-well plate and maintained until it reached ~ 80% confluency, and then were treated with increasing concentrations of SF for 6, 24, and 48 h as described above. The final volume of the culture media during treatment was 20 μl. Before measurement, the culture media were replaced with 20 μl of fresh media without SF and then 20 μl of the CellTiter-Fluor™ Cell Viability Assay's reagent was added. Then the plate was incubated at 37 °C for 3 h and the fluorescence intensity was measured by SpectraMax® i3 (Molecular Devices, US). The ratio of cell viability was normalized to untreated cells.

### Imaging
**Immunostaining.** NRCMs cultured in 6-well plate (Corning) with a plastic bottom were fixed with 4% buffered formalin as described above. Cells were washed with PBS and then permeabilized with 0.2% Triton X-100 for 10 min at RT and blocked with a blocking buffer (1% BSA, 22.52 mg/ml glycine and 0.1% Tween 20 in PBS) for 30 min at RT. The primary antibody (mouse anti-α-actinin antibodies A7811, Sigma-Aldrich) were diluted with an incubation buffer (1%BSA and 0.1% Tween 20 in PBS,1:1000) and incubated overnight at 4 °C. Secondary antibodies (goat anti-mouse, Alexa Fluor 568-conjugated antibody ab175473, Abcam) were diluted with incubation buffer (1:1000) and incubated for 1 h at RT. Cells were washed 3 times with PBS after each step. Confocal and widefield images were acquired as described in the following section.

**Microscopy imaging.** Images were acquired with a S Plan Fluor Ph2 ELWD 60x/0.70 objective on a Nikon Ti2 microscope equipped with a CREST Optics V3 spinning disk confocal (50 um pinholes). The emission iris was closed to match the objective, according to the specifications by CREST Optics. The ring of the objective was adjusted to the thickness of the plate bottom. The camera used was a Photometrix BSI Express Back illuminated sCMOS (for widefield images; Sensitivity 11 Bits mode, readout speed 200 MHz) or a Photometrix 95B Back illuminated sCMOS (for confocal images. Sensitivity 12 Bits mode).

The α-actinin channel was acquired using a 546 nm laser and a Pentaband emission filter (441/30; 511/26; 593/37; 684/34; 817/66 nm). The laser power and exposure time were set to get the brightest possible images without any saturation. The same settings were used for all the samples to be compared.

The brightness and contrast of the analyzed images were adjusted to the same settings, and the images were acquired either in wide field as tiles ($9 \times 9$) or with the spinning disk confocal. Only the widefield images were further analyzed. Visualized images in Fig. 5e and supplementary Fig. 5d were prepared using OMERO (v5.5)[60].

**Image analysis**. Image analysis was performed with the help of the SciLifeLab BioImage Informatics Facility. In short, we developed an unbiased approach to profile the shape of neonatal rat cardiomyocytes in Fiji[61]. The α-actinin channel was utilized for segmentation. The workflow consisted of a pre-processing step in which the median filter was applied to the input images in order to reduce noise. Then, the images were segmented using the Li's Minimum Cross Entropy thresholding method[62–64]. To avoid any bias in quantifying segmented cardiomyocytes between different treatments, we excluded all cropped cells that were on the images' edges. We also excluded segmented particles with an area smaller than 50000 pixels.

To quantify the changes in cell shape upon treatments, we utilized the minimum caliper diameter (MinFeret shape descriptor in Fiji), which measures the smallest diameter between any two points at the boundary of a segmented object[61]. We also utilized the area of the segmented objects to visualize changes in cardiomyocytes' size upon different treatments. However, NRCMs in 2D culture acquire a variety of random shapes and sizes, and the diversity in the shapes and sizes were further greatly enhanced by the SF and NAC treatments, which also affected the identifications of contiguous cells' boundaries. This fact made it impossible to segment all individual cardiomyocytes, especially in the NAC treatment, where cells were hypertrophied and more intertwined. This problem was not even possible to resolve by manual curation of cells' borders or by staining the cytoplasmic membrane. As our interest was to quantify the changes in cell area and diameter between different treatments, we segmented connected cells as one object. We segmented 6 images from the control group, and 8 images from each of SF and NAC treatments. The segmentation resulted in 123-196 objects per image. For each object of each image, we computed the area and minimum diameter. To plot them, we pooled the measures of each condition together. Pooling them resulted in groups of values, in which each value originated from one image (6 for the control condition, 8 for NAC and 8 for SF treatments). Then we calculated the average of each group. This resulted in 148 objects in the control condition, 144 objects in the NAC treatment, and 196 objects in the SF treatment as shown in the Source Data File. These averaged values of the segmented objects' area and MinFeret were plotted in GraphPad Prism. The implemented image analysis pipeline is available in Github (https://github.com/BIIFSweden/CellGeometryProfiling)[65]. All quantified raw images are available on BioImage Archive (Accession number S-BIAD1033).

**Transmission electron microscopy (TEM)**. Small pieces (2-3 mm³) of LV myocardium of WT and $Mlp^{-/-}$ mice were dissected and fixed for 30 min at room temperature in fixation buffer (2% glutaraldehyde + 1% paraformaldehyde in 0.1 M phosphate buffer, pH 7.4) and stored at 4 °C. After rinsing them with 0.1 M phosphate buffer, pH 7.4, specimens were postfixed for 2 h in 2% osmium tetroxide 0.1 M phosphate buffer, pH 7.4 at 4 °C, then dehydrated in ethanol followed by acetone and embedded in LX-112 (Ladd, Burlington, Vermont, USA). Leica ultracut UCT (Leica, Wien, Austria) was utilized to prepare ultrathin sections (approximately 50-60 nm). The sections were later contrasted with uranyl acetate followed by lead citrate and examined in a 100 kV Hitachi HT 7700 (Tokyo, Japan) and the digital images were captured with a Veleta camera (Olympus Soft Imaging Solutions, GmbH, Münster, Germany). Mitochondrial volume density (Vv) was calculated by point counting on printed digital images using a 2 cm square lattice. From each animal, 9 randomly taken images were utilized for appropriate sampling. The number was estimated by utilizing the cumulative mean plot[66].

**Biochemical assays**
**Mitochondrial respiration**. Mitochondria were isolated from fresh WT and $Mlp^{-/-}$ adult hearts (12-15 weeks old). Oxygen consumption of isolated mitochondria was measured with the exact same experimental procedures and settings described in a previous work[15]. The set criteria for including measured samples in subsequent analyzes was that they achieve a reading ≥ 1 (nmol O2 / mg protein/min) in all respiration states. In total, 23 out of 26 measured samples passed this threshold.

**Determining enzymatic activity of Idh2**. Idh2 activity was measured in subsarcolemmal mitochondria isolated from frozen LV myocardium of WT and $Mlp^{-/-}$ mice (males, 12 weeks old). The isolation procedure for mitochondria was adapted from Nickel et. al., 2015[15] with some modifications. Grinded frozen cardiac tissues (5 mg) were weighed in a 2 ml Eppendorf tube (Sarstedt) and homogenized with 200 μl of the isotonic isolation buffer (IS; in mM: sucrose 75, mannitol 225, HEPES 2, EGTA 1, pH 7.4, 4 °C). The homogenization was performed in the TissueLyser LT (Qiagen, Germany) for 2 × 2 min at 50 Hz with the presence of one 0.5 mm metallic ball (Qiagen) and two small 0.2 mm metallic beads (Retsch). After a brief spinning, additional 400 μl of the isolation buffer was added and homogenized again for 2 × 2 min at 50 Hz. The homogenate was centrifuged at 480 g at 4 °C for 5 min, and a 400 μL of the upper supernatant was transferred to a new tube which was kept on ice. The homogenization step (4 min at 50 Hz) was repeated with a fresh 400 μl of isolation buffer, and then 400 μl of the supernatant obtained after centrifugation was combined with previous fraction. The remaining cellular pellet was utilized to obtain the nuclear extract. To remove any cellular debris from the supernatant containing the mitochondria, the tube was gently flicked and centrifuged again for 480 g for 5 min, then 700 μL of the supernatant was moved to a new tube without disturbing the pellet. The supernatant was further centrifuged at 7700 $g$ for 10 min at 4 °C to obtain mitochondrial pellet. The pellet was washed twice with 200 μl of isolation buffer without ETGA (mitochondrial suspension solution MMS), and then lysed in 50 μl of MMS by three cycles of freezing and thawing and stored at −80ºC. The temperature was always kept below 4 °C during the whole isolation procedure. 5 μl of mitochondrial lysate was diluted with NP40 buffer and utilized for total protein quantification by BCA assay.

The activity of Idh2 was measured with the same reaction mix described in Nickel et. al., 2015[15] (in mM: Tris−HCl 10 [pH 8.0], NADP⁺ 0.2, MgCl₂ 5, and isocitrate 2) in 384-well plate with a total volume of 80 μl per well. The change in the absorption at 340 nm was continuously recorded for 60 min with the kinetic function of SoftMax Pro 7 in SpectraMax® i3 plate reader (Molecular Devices, US). All samples were measured in duplicates. As a negative control, Idh2 was inactivated in the samples by adding 1 mM of N-ethylmaleimide (NEM) to the reaction mix[67]. A total of 5 μg of mitochondrial proteins were utilized per each reaction. The generated amounts of NADPH were estimated by using Beer−Lambert law ($A = \varepsilon \times b \times C$), where $A$ is the measured absorbance at 340 nm, $\varepsilon$ is the molar attenuation coefficient ($\varepsilon = 6.22$ mM⁻¹ cm⁻¹ for NADPH), $b$ is the length of the light path (0.7 cm) and $C$ is the concentration of generated NADPH.

**Determining enzymatic activity of Tet1-3**. The ex vivo enzymatic activities of Tet1-3 were compared between the WT and the $Mlp^{-/-}$ hearts in male mice (12 weeks old) by utilizing the colorimetric Epigenase 5mC-Hydroxylase TET Activity Assay Kit (P-3086, Epigentek) according to the manufacturer's instructions. The nuclear extract was prepared by utilizing the nuclear extraction kit (OP-0002, Epigentek). The input materials for nuclear protein's extraction were the remaining cellular pellets after isolating subsarcolemmal mitochondria. 15 μg of extracted nuclear proteins were used per well for the measurement of TET activity. Samples were measured in 4 technical replicates.

## Molecular biology

**Western blotting.** 10 mg of grinded LV myocardium were weighed and lysed with 200 μl of modified RIPA buffer (50 mM TRIS, 150 mM NaCl, 1% sodium deoxycholate, 1% Sodium dodecyl sulfate, 1% Tritonx-100) supplemented with protease and phosphatase inhibitor cocktail (Thermo Scientific, USA). Tissues were homogenized for 2×2 min at 50 Hz in TissueLyser LT (Qiagen, Germany) with a 0.5 mm metallic ball (Qiagen). Then the tubes were incubated in orbital shaker for 2 h at 4 °C and centrifuged at 20800 g for 20 min at 4 °C. The lysate was transferred to another tube and quantified.

NRCMs were trypsinized with TripleE, centrifuged at 6000 g for 10 min at 4 °C to remove the supernatant and the pellet of cells resulted was stored at −80. Then 20 μl of NP40 buffer (Invitrogen) supplemented with protease and phosphatase inhibitor cocktail (Thermo Scientific, USA) was added to each pellet of cells. The pellet was thawed on ice and then frozen again for a total of 5 cycles, with vortexing in between, to completely lyse the cells. Then the tubes were incubated in orbital shaker for 2 h at 4 °C and centrifuged at 20800 g for 20 min at 4 °C. The lysate was transferred to another tube and quantified.

HFF1 cells, cultured in 10 cm dishes and treated with SF or NAC for 28 h, were trypsinized (TrypLE™ Express, ThermoFisher Scientific, 12604013) and centrifugated at 650 g. Protein lysates were prepared from the cellular pellet with the same procedure described above for NRCMs.

Protein concentrations were estimated by bicinchoninic acid assay (BCA protein assay kit, Pierce). The quantification was performed in 384-well plate (Corning) in a total volume of 50 μl.

Equal amounts of total protein lysates (10, 20, 30 or 60 μg) were denatured in sample buffer (Invitrogen) supplemented with sample reducing agent (Invitrogen), and electrophoresed on 4-12% gradient Bis-Tris gels (Invitrogen), with MES SDS running buffer (Invitrogen) and then transferred to 0.2 μm PVDF membrane (BioRad) using BioRad's Trans-Blot Turbo transfer system at 1.3 A and 25 V for 7 min. The membrane was then fixed with 0.4% formalin for 15 min, then washed twice with BPS, and blocked with 5% w/v dry milk in 20 mM Tris base, 135 mM NaCl, 0.1% tween 20 (TBST). Membranes were then blotted with primary antibodies indicated in Supplementary Table 4. Blots were later washed and incubated with the respective secondary antibody indicated in Supplementary Table 4. The PageRuler™ Plus protein ladder (Thermo Scientific™) was utilized to estimate the size of the reactive bands.

Multiple proteins with distinct molecular weights were simultaneously blotted. However, when there was a need to re-incubate a membrane with another antibody, previous antibody staining was stripped with a mild stripping buffer (1.5% glycine, 0.1% SDS and 1% Tween20, pH=2.2), or with a harsh stripping buffer (2% SDS, 62.5 mM Tris pH 6.8 and 100 mM β-mercaptoethanol). Reactive bands were visualized with SuperSignal West Pico PLUS chemiluminescent substrate (ThermoFisher Scientific) and detected with ChemiDoc™ gel imaging system (BioRad).

Western blotting of isolated mitochondria (Supplementary Fig. 1f) was performed using standard protocol. In brief, indicated amounts of mitochondria were solubilized in lysis buffer containing Tris-HCl 60 mM, SDS 2%, glycerol 10%, β-mecaptoethanol 1% and bromphenol blue 0.01%. The cleared homogenate was separated on a 12% SDS-PAGE gel and electrophoretically transferred to a PVDF membrane. Membranes were blocked in TBS containing 5% non-fat dry milk for 120 min at room temperature and blotted with anti-SDHA, anti-Vdac, anti-Ndufb8, and anti-Cox4-1 (Supplementary Table 4). Then blotted with secondary antibodies and visualized as described above.

The bands were quantified using Image Lab v6.1 (BioRad). Fold change was calculated by first normalizing the band intensity of the target protein in each sample to the corresponding band intensity of the housekeeping protein or to total protein, to obtain the band's relative intensity. Then, the relative protein intensity in each sample was divided by the average relative intensities of all control samples on each blot to obtain the fold change for the target protein in both the control and the experimental samples.

All raw images, the intensities of the quantified bands, and the calculations for the relative intensities are available in Source Data File and Supplementary Data 2.

**RNA extraction.** RNA from LV myocardial tissues, NRCMs and hFF1 cells was extracted by a high throughput method utilizing TRIzol (Invitrogen) following the manufacturer's instructions with some modifications. In short, cells in each well of the 24-well plate were lysate with 2 × 75 μl of TRIzol and transferred to 250 μl 8-strip tube (Sarstedt). For tissues, 150 μl of TRIzol was added to 2-3 mg of grinded left ventricle cardiac tissues in 250 μl 8-strip tube and pipetted up and down multiple times until the tissues were completely dissolved. Then after 5 min of incubation at RT, 30 μl of chloroform (Sigma-Aldrich) was added and the resulted two liquid phases were separated by centrifugation. The aqueous phase was transferred to a new 250 μl 8-strip tube containing 1 μl of glycogen (Roche) and the RNA was precipitated overnight by 100 μl of isopropanol at −20 °C. RNA pellet resulted after centrifugation, was washed twice with 100 μl of 80% ethanol and dissolved in 20-50 μl of RNase-free water. The RNA was quantified with Qubit RNA HS assay (Invitrogen) and utilized for subsequent experiments.

For subsequent applications that required high amounts of RNA with a gDNA digestion step, RNA was instead extracted from 15 mg grinded heart tissues. TissueLyser LT (Qiagen, Germany) was used to homogenize the tissue in TRIzol for 2 min at 50 Hz with a 0.5 mm metallic ball (Qiagen). After obtaining the RNA pellet, the pellet was washed twice with 80% ethanol and dissolved in 90 μL of RNase free water. Residual gDNA was then digested with the DNase Max® (Qiagen) and the large-RNA fraction (>200nt) was recovered with RNeasy MinElute Cleanup Kit (Qiagen) following the manufacturer's instructions. The extracted RNA was quantified with Qubit RNA BR assay (Invitrogen) and the quality of RNA was assessed by Fragment Analyzer standard sensitivity RNA kit (Agilent Technologies, USA).

**Reverse Transcription and qPCR.** For the first strand synthesis of cDNA, SuperScript IV First-Strand Synthesis System (Invitrogen) was used according to the manufacturer's instructions. By utilizing the estimated RNA concentration, the cDNA amount was adjusted to 5 ng/reaction. QPCR was performed using the SYBR® Green Supermix according to the manufacturer's instructions with the following thermal program (enzyme activation at 95 °C for 3 min, 40 cycles of [95 °C for 10 sec, 60 °C for 30 sec, plate read], and followed by recording the melting curve [65 to 95 °C, with 0.5 °C increment, for 5 sec]). The qPCR was performed in a 384-well plate (BioRad) by utilizing CFX384 Touch™ Real-Time PCR Detection System (BioRad, US) and the software Bio-Rad CFX Manager 3.1 (3.1.1517.0823). All primers are shown in Supplementary Table 5. The housekeeping gene *GAPDH* was used for normalization. All qPCR reactions were run in two technical replicates.

**Deep RNA sequencing.** Prior to RNA-seq, integrity and quantity of the RNA were assessed by Fragment Analyzer Standard Sensitivity RNA kit (Agilent Technologies, USA). All samples had an RNA integrity number >8.7 and were deemed of sufficient quality for mRNA-seq analysis. 1000 ng of total RNA was used as input to each mRNA-seq library. KAPA mRNA HyperPrep kit (Roche, Switzerland) was used for reverse transcription, generation of double-stranded cDNA and subsequent library preparation and indexing according to the manufacturer's instructions. Quality and quantity of libraries was assessed by Fragment Analyzer standard sensitivity NGS kit (Agilent Technologies, USA). Indexed libraries were pooled in equimolar ratios. The sample pool was quantified with a Qubit Fluorometer (ThermoFisher

Scientific, USA) using the dsDNA HS kit (ThermoFisher Scientific, USA), further diluted, and sequenced to >15 M paired reads/sample on NextSeq500/550 (Illumina, USA) with PE 75 base pair (bp) read length setting.

Raw RNA sequencing data on the myocardium of WT and *Mlp*$^{-/-}$ utilized in Fig. 7c was obtained from the public repository SRA (SRA accession ID: PRJNA327790). The library preparation and the sequencing are described in the associated metadata on SRA. In brief, high-quality RNA was extracted from whole heart tissues, and the libraries were prepared with Illumina TruSeq Stranded Total RNA Library Prep Kit with Ribo-Zero Gold (#RS-122-2301) following the manufacturer's recommendations. Prepared libraries were pooled and sequenced on the Illumina NextSeq 550 (Paired-end. 75 bp) using sequencing-by-synthesis (SBS) chemistry v4 according to the manufacturer's protocols. Each TruSeq RNA library produced an average yield of 3.6 Gb of sequencing data, with an average of 80% of the reads passing a quality score equal to or greater than Q30. The RNA was from the same heart samples that were utilized to derive BS and OxBS libraries provided under the same accession ID: PRJNA327790.

**SmartSeq2 library preparation and sequencing.** Transcriptomic profiling for NRCMs transduced with ShRNA targeting L2hgdh, D2hgdh, Tet1-3 or scrambled control was performed by RNA sequencing using Smart Seq2 protocol[68], with the help from the Single cell core Facility (SICOF). Each sample was prepared in 4 technical replicates utilizing 25 ng of purified RNA. The mRNA was reverse transcribed into cDNA using oligo (dT) primer and SuperScript II reverse transcriptase (Invitrogen). A template-switching oligo was used for the second strand cDNA synthesis. The cDNA was later amplified by PCR for 14 cycles. After purification, the quality of the cDNA was assessed by 2100 Bioanalyzer with a DNA High Sensitivity chip (Agilent Biotechnologies). Then the cDNA was tagmented with Tn5 transposase, and each library was uniquely indexed using the Illumina Nextera XT index kits (Set A–D). The uniquely indexed libraries from half of a 384-well plate were thereafter pooled together and sequenced on one lane of a HiSeq3000 sequencer (Illumina), using dual indexing and single-end 50 base pair (bp) read length setting.

**Bioinformatic analysis**
**Reads processing of RNA sequencing.** Sequenced reads in fastq files were quality controlled and read counts were derived with the bcbio pipeline (1.1.7-b)[69]. Reads were aligned to the mouse genome mm10 or to the rat genome Rnor_6.0 using STAR (2.6.1d)[70] with genome annotation from ENSEMBL (version 91). Counts were calculated using featureCounts (1.6.4)[71]. Differential gene expression analysis was performed with DESeq2 (R package v 1.26.0). A full list of differentially expressed genes are available in Supplementary Data 3 (BaseMean >5).

**Epigenetic analysis.** Raw data of cardiac whole genome BS and OxBS sequencing on WT and *Mlp*$^{-/-}$ were obtained from the public repository SRA (SRA accession ID: PRJNA327790). The library preparation and the sequencing are described in the associated metadata. In brief, Qiagen DNeasy® Blood and Tissue kit was utilized for extracting genomic DNA from the myocardium of WT and *Mlp*$^{-/-}$ mice (males, 12 weeks old, 3 animals in each genotype). True-Methyl® Whole Genome library preparation kit (Cambridge Epigenetix Alpha Version 1.1, June 2015) was utilized to prepare BS and OxBS libraries according to the manufacturer's instructions. Prepared libraries with his kit contain spike-in digestion and sequencing controls, which could be utilized to estimate the successful completion of the DNA oxidation and bisulfite conversion. Libraries were sequenced on Illumina HiSeq 2500 sequencer (paired-end, 125 bp) and each library produced an average yield of 91.8 Gb of sequencing data, with approximately 85.4% of the reads achieving a quality score of Q30 or higher.

For each sample, raw reads of BS or OxBS sequencing from multiple lanes were merged into paired fastq files. The downstream processing analysis was performed according to the TrueMethyl® Data Analysis pipeline (Cambridge Epigenetix (CEGX)). In brief, raw reads were trimmed to remove the adapter and low-quality sequence using Cutadapt (v1.11)[72]. Trimmed BS and OxBS reads were then mapped to the mouse genome (GRCm38.p4) using Bismark (v0.19.0)[73]. The resulting 12 bam files (3 WT, 3 *Mlp*$^{-/-}$, BS, and OxBS) were sorted and indexed using SAMtools (v1.8)[74] and were then passed into the deduplication and methylation extraction tools of Bismark. Base methylation counts were computed from the Bismark output using the Bismark methylation extractor tool.

**Estimation of 5mc and 5hmC.** The average percentage of whole genome 5mC across all biological samples was inferred from the OxBS Bismark alignment report. The whole genome percentage of 5hmC was inferred by the direct subtraction of methylated cytosine' percentages between the BS and OxBS samples from the Bismark alignment report. The percentages of 5mC and 5hmC were found to be 58.7% and 4.4%, respectively. The very low percentage of 5hmC would require a minimum of 100X sequencing coverage to be reliably estimated at a single base pair resolution. Alternatively, an increase in the statistical power can be achieved through the pooling of neighboring CpGs with a small compromise in the actual resolution[75].

The depth of averaged coverage of the publicly available BS and OxBS sequencing data for each biological replicate was estimated to be ~8X. Therefore, a CpGs pooling strategy was developed to boost the minimum coverage to over 100X. Using R (v.3.6), the CytosineReports files were normalized by using methylKit library (v1.12.0)[76] according to the median coverage option. Sliding windows of 30 CpGs with a step size of 2 CpGs were generated for all CpGs in the reference genome (every CpG is covered by 15 windows).

All CpGs with an outlier high coverage (more than the 99.9% percentile) were removed. The "unite" function with (min.per.group=1) was utilized to remove all CpGs that had no initial coverage in any of the sequenced libraries. All remaining CpGs (target-CpGs) in every defined sliding window were tiled together and only windows with a minimum of 100X combined coverage were included in subsequent analysis. Then the pooled window coverage was divided by the number of CpGs and the resulting values were assigned to each CpG inside that window. As each individual CpG would be covered by multiple overlapped windows, the resulting multiple assigned coverages for each CpG were averaged again to obtain the final pooled coverage of each individual remaining CpG. Thereafter, MLML2R[77,78] in (R package v0.3.3) was used to estimate 5mC, 5hmC and non-methylated C at every CpG. The codes used for BS and OxBS analysis is available through this link https://doi.org/10.5281/zenodo.10632409 [https://github.com/HSiga/BSseq2]··[79].

**DMR analysis.** Methylation levels and coverage profiles of every CpG site quantified with the bismark_methylation_extractor tool were further used for Differential Methylation Region (DMR) analysis with the "bsseq" R library[80]. The purpose of this analysis is to utilize the natural strong correlation between neighboring CpG sites that typically results in their identical methylation levels. In this way, one can potentially detect long stretches of DNA (beyond individual CpG sites) with differential methylation between two groups of samples. We filtered away CpG sites covered by less than two reads in both WT and *Mlp*$^{-/-}$ samples, as there was not enough statistical evidence for reliable quantification of the methylation levels of those sites. Further, methylation levels of all individual CpG sites were smoothed using the "Bsmooth" algorithm[80] for taking into account the context methylation levels of the neighboring CpG sites. Next, t-statistic was computed in a sliding window across the mouse reference genome with "Bsmooth.tstat", and thresholding on the computed t-statistics was implemented by the

"dmrFinder" function in the "bsseq" R library. For the downstream analysis, we kept only DMRs with at least three CpG sites and a difference of at least 10% in average methylation levels between WT and *Mlp*$^{-/-}$ for 5mC, and 1% for 5hmC data. This resulted in 26 052 DMRs for 5mC and 138 369 DMRs for 5hmC data. Finally, the identified DMRs were ranked by the sum of the *t*-statistics of comprising CpG sites weighted by the number of CpG sites in each DMR, i.e., the "areaStat" metric reported by the "bsseq" R library. A full list of annotated 5mC and 5hmC DMRs is available in Supplementary Data 4.

**CpG and DMR annotation.** Both individual CpG sites and DMRs were annotated by their overlapping functional elements such as exon, intron, CDS, promoter, enhancer, CTCF, TF binding site, and open chromatin regions. For this purpose, we used mm10 mouse reference genome annotation downloaded from the UCSC resource [http://hgdownload.cse.ucsc.edu/goldenpath/mm10/database/], and Ensembl Regulatory Build [http://ftp.ensembl.org/pub/release-99/gtf/mus_musculus/]. For overlapping CpG and DMR coordinates with the functional regions, we used "bedtools closeset" tool and selected only CpG / DMR – functional region pairs that had the 0-distance implying that they overlapped. Further, for each functional group of elements (exon, intron, CDS, promoter, enhancer, CTCF, TF binding site, and open chromatin regions) we counted the number of times they overlapped the DMRs between WT and *Mlp*$^{-/-}$ identified previously (intersection) and normalized this number by the total amount of DMRs and functional elements (union). This allowed us to construct a type of Jaccard enrichment metric (intersection over union) for investigating what group of functional elements the DMRs were predominantly overlapping with.

**Combining methylation data with RNAseq.** RNA-seq data from *Mlp*$^{-/-}$ and WT hearts were aligned with STAR[70], quantified with featureCounts[71] and normalized with TMM normalization[81]. Further, for each functional element (exon, intron, CDS, promoter, enhancer, CTCF, TF binding site, and open chromatin regions). We computed its mean methylation level by averaging individual methylation values across CpG sites within the functional element region. Supplementary fig. 7c demonstrates genome-wide average 5mC and 5hmC levels for each functional element separately for *Mlp*$^{-/-}$ and WT samples. Next, the functional elements were matched to the to their closest genes using a custom R script, and methylation levels of the functional elements were correlated against gene expression levels of the corresponding genes using Spearman rank correlation within each WT and *Mlp*$^{-/-}$ sample. Finally, genome-wide average Spearman correlation coefficients per functional element were visualized as a heatmap, Fig. 7c, by using the "pheatmap" R package [https://cran.r-project.org/web/packages/pheatmap/index.html].

**Pathway analysis.** Differentially expressed or methylated genes were analyzed using Ingenuity pathway analysis ¨IPA: QIAGEN Inc., https://www.qiagenbioinformatics.com/products/ingenuity-pathway-analysis[82]. Only differentially expressed genes (DEG) with *p* values below 0.05 were included in the analysis. When the number of DEG were over the permitted limit by IPA, only DEG with *p* values below 0.01 were included in the analysis. However, in the IPA comparison analysis, when multiple data sets were compared together, the same threshold of *p* value (i.e., 0.01 or 0.05) was utilized for all data sets included in the same comparison analysis. The calculated *p*-value of the IPA core analysis is based on the Right-Tailed Fisher's Exact Test, and it reflects the likelihood that an association between a set of significant molecules in the investigated data set and a given pathway could happen due to a random chance. The smaller the *p*-value is, the less likely that an association is random. The activation z-score reflects the probable activation states and its direction for a pathway, compared to a model that assigns random regulation directions. Plotted pathways in all figures were selected from a list of significantly enriched pathways based on their relevance to the context. All complete lists of pathways for all figures, together with differentially expressed gene lists are provided in Supplementary Data 1 and 3, respectively.

**Upstream regulator analysis.** IPA upstream regulator analysis is a way to predict the activation state of an upstream regulator molecule (such as NRF2), based on observed changes of gene expression in an experimental dataset and expected causal effects derived from the existing literature[82]. It plots the expression of all differentially expressed downstream targets of the upstream regulator (green and red filled circles), and when the expression of most target genes is consistent with the expected activation state derived from the existing literature, the upstream regulator would be predicted to be active in the experimental sample in comparison to the control. The analysis does not consider the expression of the upstream regulator itself, as for example, when NRF2 is activated, it translocates to the nucleus, but without an increase in its mRNA level. Upstream regulator analysis also uses the same activation z-score as the one described above for the IPA Pathway analysis.

**Statistics and reproducibility.** All data are presented as mean ± SEM. Unpaired or paired *t*-test was calculated with GraphPad Prism 8 software. If other tests were used, detailed descriptions were included in the figure legends. When relevant, experiments were replicated a minimum of two or more times, either as technical replicates, by analyzing another set of biological samples, by employing two different techniques, such as qPCR and RNA sequencing, or by deriving data from various cell types or models using the same experimental settings. The resulting data demonstrated a consistent trend similar to the representative results presented. A description of the number of experimental replicates is provided in the Source Data file for certain figures.

Regarding the in vivo trial involving the Nrf2 activator, a preliminary trial was conducted with a limited sample size, followed by a larger-scale trial. Nrf2 activation was notably observed in both studies. However, the impact on cardiac function was solely evaluated in the larger trial, in which male *Mlp*$^{-/-}$ showed a significant improvement, whereas females did not exhibit any tendency of improvement, but rather exhibited the opposite effect. Consequently, due to ethical considerations, this experiment was not repeated.

**Reagents.** Chemical reagents were of the highest grade and obtained from Sigma (Germany) unless indicated otherwise. Supplementary Tables 6 and 7 indicate the characteristics of certain reagents and accessories mentioned in the material and method part.

**Reporting summary**
Further information on research design is available in the Nature Portfolio Reporting Summary linked to this article.

## Data availability
The data supporting conclusions drawn in this study are accessible within the article itself and the Supplementary Information provided. The sequencing data generated in this study have been deposited in the Sequence Read Archive (SRA) under accession numbers PRJNA821374 for the in vivo materials, and PRJNA773866 for the in vitro materials. All quantified raw images are available on BioImage Archive Accession number S-BIAD1033. Processed sequencing data and the densitograms for the quantification of Western blotting images are provided in Supplementary Data 1–4. Further information and reasonable requests for resources and reagents should be directed to the lead contact Zaher ElBeck (zaher.elbeck@ki.se). Source data are provided with this paper.

## Code availability
The codes used for BS and OxBS analysis is available through this link https://doi.org/10.5281/zenodo.10632409 [https://github.com/HSiga/

BSseq2][79]. Whereas the codes used for image analysis is available through this link https://doi.org/10.5281/zenodo.10631363) [https://github.com/BIIFSweden/CellGeometryProfiling][65].

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

## Acknowledgements

We acknowledge all people who have contributed to this study, either by providing funds or technical assistance. We want to particularly acknowledge Mattias Svensson, Department of Medicine Huddinge, for his valuable intellectual input and feedback, Oscar Franzén for his valuable input in the epigenetic bioinformatic analysis, Cristobal Dos Remedios and Amy Li for the provision of the human heart samples from the Sydney biobank and for his scientific input to the manuscript. Byambajav Buyandelger for her valuable input and for helping in preparing samples for TEM. Anne-Laure Lainé, for preparing AZ925, David Brodin at the BEA facility, KI, Huddinge, for his support in analyzing the ShRNA-KD RNA seq data, Kjell Hultenby for performing the TEM and quantifying the images obtained, Sara Fernandez Leon and Charlotte Webster for their technical assistance, and Joseph W DePierre for scientific English editing of the manuscript. We would like to acknowledge

the Single-cell core facility at the Flemingsberg campus (SICOF) at Karolinska Institutet (KI) for their sequencing. This facility is supported by grants from the KI Department of Medicine (MedH) and KI Infrastructure, as well as the infrastructure for the Strategic Research Area (SFO) on Stem Cells and Regenerative Medicine. Fluorescence microscopy was performed at the Live Cell Imaging Core facility/Nikon Center of Excellence at Karolinska Institutet, with support from the Swedish Research Council, KI infrastructure, and the Centre for Innovative Medicine. We also acknowledge Gisele Miranda for her input in the analysis of fluorescence images. The BioImage Informatics Facility for carrying out image analysis is funded by SciLifeLab, the National Microscopy Infrastructure, NMI (VR-RFI 2019-00217), and the Chan-Zuckerberg Initiative. The Proteomics Biomedicum core facility of Karolinska Institutet for performing the mass spectrometric analysis. Antibodies against VDAC, COX4I1, and NDUFB8 were a kind gift from Peter Rehling, Göttingen, and we thank Berkan Arslan for performing the Western blotting. We also acknowledge Hanna Ebrel from the Institute of Pharmacology and Toxicology, University of Würzburg, Germany, for her invaluable assistance in preparing and differentiating hiPSC-CMs. C.M. is supported by the German Research Foundation (DFG; SFB 894, TRR-219; Ma 2528/7-1) and the German Ministry of Education and Research (BMBF, 01EO1504). N.O. is supported financially by the Knut and Alice Wallenberg Foundation as part of the National Bioinformatics Infrastructure Sweden at SciLifeLab.

## Author contributions

Z.E.: Conceived, designed, conducted, and supervised all experiments; led the analysis of the epigenetic data; analyzed and interpreted the data; and wrote the manuscript. M.B.H.: Echocardiography; jointly contributed to the design and animal gavage in connection with the study involving the inhibitor of Keap1; assay of Tets activity; provided study materials, interpreted the data, and edited the manuscript. H.S.: Echocardiography and bioinformatic analysis of BS and OxBS data; differential gene expression analysis of all RNA seq data. N.O.: Development and supervision of the procedures for analysis of epigenetic data. J. L.: Preparation of DNA libraries and sequencing in connection with the in vivo Keap1 inhibitor study. F.K. & A.W.: Alignment and demultiplexing of RNA seq data obtained in the Keap1 inhibitor study. D.K.: IPA analysis of human heart samples. R.J., R.B., T.J., E.F.: Discovery of AZ925 and enabling the pilot study. A.N., M.K., C.M.: Measurement of the rate of respiration by mitochondria isolated from $Mlp^{-/-}$; Western blotting of VDAC, COX4I1, and NDUFB8; interpreted the data, and provided scientific input and editing of the manuscript. M.F, A.D. Cl.B: Scientific contribution to the manuscript. R. M.: Provision of NRCMs for the measurements of GSH and GSSG. K. S.-B.: preparing and differentiating hiPSC-CMs. L.H.L.: Supervised and provided scientific and editing input to the manuscript. A.V.: Design, acquisition, analysis, and interpretation of all mass spectrometric data, edited the manuscript. Ch.B.: Supported the experiments, interpreted the data, and provided scientific and editing input to the manuscript. All authors contributed to the final editing of this manuscript.

## Funding

## Competing interests

F.K., J.L., A.W., R.J., R.B., T.J., E.F., M.F., A.D., Cl.B. and R.M. are current employees or were employees of AstraZeneca and may own stock or stock options. The rest of the authors declare no competing interests.

## Additional information

¹Department of Medicine Huddinge, Karolinska Institutet, Campus Flemingsberg, 141 57 Huddinge, Sweden. ²Departmenty of Immunology, Genetics and Pathology, Rudbeck Laboratory, Uppsala University, Uppsala, Sweden. ³Bioscience Renal, Research and Early Development, Cardiovascular, Renal and Metabolism (CVRM), BioPharmaceuticals R&D, AstraZeneca, Gothenburg, Sweden. ⁴Department of Biology, National Bioinformatics Infrastructure Sweden, Science for Life Laboratory, Lund University, Lund, Sweden. ⁵Data Sciences and Quantitative Biology, Discovery Sciences, R&D, AstraZeneca, Gothenburg, Sweden. ⁶Translational Genomics, Centre for Genomics Research, Discovery Sciences, R&D, AstraZeneca, Gothenburg, Sweden. ⁷Translational Science & Experimental Medicine, Research and Early Development, Cardiovascular, Renal and Metabolism (CVRM), BioPharmaceuticals R&D, AstraZeneca, Gothenburg, Sweden. ⁸Neuroscience, BioPharmaceuticals R&D, AstraZeneca, Cambridge, United Kingdom. ⁹Early Cancer Institute, University of Cambridge, Cambridge, United Kingdom. ¹⁰Institute of Pharmacology and Toxicology, University of Würzburg, Würzburg, Germany. ¹¹Clinic for Cardiology and Pneumology, Georg-August University Göttingen and DZHK (German Center for Cardiovascular Research), Partner Site Göttingen, Göttingen, Germany. ¹²Department of Translational Research, Comprehensive Heart Failure Center (CHFC), University Clinic Würzburg, Würzburg, Germany. ¹³Department of Medicine Karolinska Institutet, and Department of Cardiology, Karolinska University Hospital, Stockholm, Sweden. ¹⁴Division of Chemistry I, Department of Medical Biochemistry & Biophysics, Karolinska Institutet, Stockholm, Sweden. ✉e-mail: zaher.elbeck@ki.se

