## [Peer Review File · Nature Communications]

Epigenetic modulators link mitochondrial redox homeostasis to cardiac function in a sex-dependent mannerREVIEWER COMMENTS

Reviewer #1 (Remarks to the Author):

In this study, Elbeck et al reported altered IDH2 expression level in human and mouse cardiomyocytes with eccentric hypertrophic hearts. The authors tried to link the downregulation of IDH2 with cardiac antioxidant defense and further correlated the change with epigenetic regulation. Although the manuscript presented some interesting data, most conclusions were made from indirect evidence. The authors emphasized on the antioxidant role of various regulators and redox homeostasis, while there is no single direct evidence to demonstrate that the alteration of intracellular redox level is directly related to pathological conditions. Most conclusions were drawn from the correlation between gene expression changes but without experiments to demonstrate the causality. The logic flow of the manuscript is difficult to follow. The authors tried to cover the topic related to metabolic, transcriptional, and epigenetic changes; however, the manuscript lacks a central focus and in-depth insights. The authors are suggested to edit the language and reshape the manuscript.

Figure 1.

1. The rationale of selecting IDH2 is not clearly stated. There are many other genes showing similar expression patterns as IDH2 in Figure 1E. The authors do not provide a strong justification for selecting IDH2 as a downstream target.
2. GAPDH is also involved in mitochondria metabolism. It would be better to use other proteins as a loading control. This also applies to other figure panels.
3. Please specify the gender of human analysis.
4. Please spell out the full name of DCM when it was first mentioned in the text file and figures.
5. It would be better to use a second irrelevant cardiomyopathy model to demonstrate that the downregulation of IDH2 is specific in eccentric hypertrophy.

Figure 2.

1. Western blot to evaluate protein levels of key targets mentioned in this figure is needed since RNA level is not necessarily aligned with protein levels. Furthermore, the authors observed significant reduction of total RNA with increased H₂O₂ treatment. The decreased expression of IDH2 might simply be due to the reduction of total RNA.
2. H₂O₂ treatment might result in cell death, especially at high concentrations of H₂O₂ treatment. The authors need to consider the cell conditions during the experiments. Working with unhealthy cells might result in biased and inaccurate conclusions.

Figure 3.

1. The authors claim that increased IDH2 activity might be due to desuccinylation, however in the study that authors cited in this study (Zhou et al EMBO reports 2016) is compared to the IDH2 activity with and without succinylation at the same amount protein level. In this study, the authors observed significant downregulation of IDH2 at the protein level, but it lacked direct biomechanical evidence demonstrating that desuccinylation of IDH2 could overcome the decreased protein level and is responsible for increased IDH2 activity.
2. The presentation of Figure 3O is confusing. It is unclear what key information the authors would like to convey in this figure.
3. The authors keep emphasizing that cellular redox level was altered. The conclusion is made from the alteration of expression of IDH2, L2HG and D2HG. There is no evidence directly showing the changes of cellular redox level.
4. In Figure 1 and Figure 3, the authors claim correlation between decreased *Idh2* expression and increased L2HG in *Mip*^{-/-} mouse than WT group. However, in the later study (Figure 3I-M), the authors knocked down *L2hgdh* and induced the increased L2HG, which then resulted in increased *Idh2*. These two results seemed to be in conflict. Please clarify this point.
5. The rationale of pointing to NRF2 as the downstream target is not clearly justified. Based on data shown in Figure 3N, other pathways also exhibited similar alterations.

Figure 4.

1. The authors claimed that NAC upregulates *Idh2* and reduces Nrf2 targets. The results do not agree with the conclusion. Based on data shown in Figure 4B, there was almost no changes of *Idh2* expression in NAC treatment. Furthermore, not data was presented to demonstrate deactivation of Nrf2. Barely any changes were noted for three genes listed in Figure 4B under NAC treatment. Furthermore, the expression of these 3 genes was not sufficient to reflect Nrf2 activity. The nuclear translocation is the key to demonstrate the NRF2 activity.
2. The authors emphasized that the changes were due to the antioxidative control mechanism, while there is not direct evidence to demonstrate that NAC or SF treatment indeed alters intracellular ROS.
3. The authors observed reduction of cell numbers of SF and NAC treatment, suggesting toxicity of these drugs. The recovery of the cells might be due to short half-life and the inactivation of these drugs after 48 hr treatment. The authors claimed that "intracellular redox homeostasis is involved in regulating the morphology of CMs". However, it might simply be due to the toxicity of this drugs. The experimental design is questionable.
4. The authors mentioned gender difference in this study might be due to Nrf2 mediated response to oxidative stress. However, the conclusion is solely based on RNA-seq analysis which is not convincing. The authors need strong biochemical evidence to validate differential Nrf2 activity as well as oxidative

stress in female and male. The authors also need strong experimental evidence to demonstrate the causal relationship between gender difference and oxidative stress.

Figure 5.

The entire figure is based on the usage of AZ925. However, there is no citation of this medicine and there is no data demonstrating that AZ925 could disrupt the NRF2 activity. The foundation of this experiment is questionable.

Figure 6.

1. Many other dioxygenases are regulated by 2OG and L2HG. TET proteins are only one of the protein family that can be regulated by 2OG and L2HG. The rationale of focusing on TET enzyme is unclear. The DNA methylation and 5hmC analysis is not clearly stated. The low coverage (58.7% of mC and 4.4% of hmC) of CpG sites of DNA methylation and 5hmC analysis might introduce biased and inaccurate results.
2. Although the authors identified DMRs or DHMRs in the study. It is unclear how these changes were directly correlated to phenotypes observed in an earlier study. Again, the causal relationship of altered epigenetic modifications with transcriptional outputs is unclear. The authors need to perform dCas9 based epigenome editing to dissect this causality.
3. Based on the example of genome-browser view shown in Figure 6F, only subtle changes of DNA methylation and hydroxymethylation were observed. It is unclear how these subtle changes would be responsible for downstream transcriptional changes. The similar concern applies to Figure 6G. Although there is statistical significance among all the analyzed groups, only less than 1.5-fold change of expression was noted, which might not have strong biological consequence.

Reviewer #2 (Remarks to the Author):

Re: "Epigenetic modulators link mitochondrial redox homeostasis to cardiac function"

The Zaher ElBeck et al. integrated several cardiac hypertrophic datasets and showed downregulation of Idh2 but increased activity in cardiac eccentric hypertrophic. Increased oxidative stress by H₂O₂ or SF decreased IDH2, whereas reduced ROS upregulated Idh2. Nrf2 and Idh2 exhibited mutually antagonistic regulation. ROS regulated Idh2 through altered activity of Tet enzymes, which utilize Idh2's product metabolite to demethylate DNA (5mC->5hmC). The authors demonstrate that heart failure altered 5mC and 5hmC distribution, including at Idh2 itself, suggesting a mechanism by which ROS epigenetically regulates Idh2 expression. This study provides some new insight into redox homeostasis during cardiac hypertrophy, especially about the regulation of Idh2, an essential component of the antioxidant system.

This paper was disjointed and hard to follow. The hypothesis and the logic of the supporting evidence should be more clear. Tangential side stories should be minimized or eliminated. Several lines of evidence are correlative and indirect. Overall the main points are not adequately supported by data.

Major:

1. "IDH2 is downregulated in eccentric hypertrophy." This point is made from several datasets. IDH2 protein was downregulated when normalized to GAPDH (Fig. 1C) or b-actin (Fig. 3L-M). IDH2 is a mitochondrial matrix protein, while GAPDH is cytosolic. The mitochondrial matrix protein Cox4 was decreased in Mlp^{-/-} hearts (Fig. S1G) and ETC proteins were also decreased (Fig. S1F-G). Is IDH2 downregulated when normalized to a mitochondrial matrix protein? Why were different housekeeping genes used for normalization? IDH2 protein levels was analyzed in Fig. 1I-J and S1K-L – why are these analyses separated?
2. The author show that IDH2 mRNA and protein level, and IDH2 activity, are dissociated, i.e. expression level can decrease while activity increases. Yet most of the paper focuses on IDH2 level rather than activity. When the topic is upstream regulation of IDH2 expression, measurement of IDH2 level is appropriate. When the topic is IDH2 regulation of downstream pathways, its activity should be measured rather than making inferences based on its expression.
3. The author knocked down the L2hgdh and showed "accumulation of L2-hydroxyglutarate associated with heart failure upregulates Idh2" in NRCMs. But in the Mlp^{-/-} mice heart, L2HG was increased (Fig.3F) but with decreased Idh2 (Fig.1F-1K). These results appear to conflict. There are several pathways which can regulate Idh2 expression, but which are most important to decrease IDH2 in eccentric heart failure, exemplified by Mlp^{-/-} mice?
4. What is the relevance of "intracellular redox homeostasis modulates the morphology of cardiomyocytes" to the focus of the manuscript? Many studies show ROS is elevated in failing hearts, which are often hypertrophied. However, SF (increased ROS) led to cardiomyocyte atrophy and NAC (reduced ROS) promoted cellular hypertrophy. This seems contrary to what would be expected based on elevated ROS in cardiac hypertrophy and failure.
5. In several sections, inferences from transcriptomic analyses are relied on rather than direct measurement. For example, IPA enrichment analyses of gene expression are used to argue that manipulations affected redox homeostasis. However, direct measurements, such as biochemical markers of ROS or direct measurement of ROS levels, are not made. Since IDH2 is mainly responsible for generation of mitochondrial NADPH, a necessary cofactor for antioxidative processes, the NDAP⁺/NADPH ratio could be another readout for redox homeostasis.
6. The causal link between impaired systolic dysfunction in Mlp^{-/-} male mice and insufficient ROS defenses is not well established. Mlp^{-/-} male mice have more severely impaired systolic function than females. This correlated with transcriptomic profiles more enriched for oxidative stress responses in males. Even if elevated ROS levels were measured (point 4), this would remain a correlation rather than demonstration of a less adequate oxidative stress defense in males. The conclusion was also supported by improvement of heart function by activation of NRF2 in males. However, NRF2 has multiple roles, not just regulation of ROS defenses. For example, NRF2 also influences mitochondrial respiration and fatty acid oxidation.

7. The gender dimorphism of *Mlp*^{-/-} mice is interesting but it is a side story. The male mice support the main line of logic of the manuscript. This could be presented first, then the female mice could be presented second, with their preserved heart function, relatively lower ROS levels, and lack of benefit from NRF2 activation providing correlative supportive evidence.

8. The authors should provide biochemical evidence that AZ925 activates NRF2. For instance, immunostaining and subcellular fractionation to show NRF2 nuclear localization, and reporter assays to show NRF2 transcriptional activation.

9. The authors argue that changes in 5mC and 5hmC in *Mlp*^{-/-} mice are responsible for downregulation of *Idh2* in response to oxidative stress. This is not sufficiently supported by data. In 6A-B, small, statistically insignificant changes are described as showing a “trend towards increased levels of 5mC and fewer 5hmC”. The magnitude of change is small and the pvalue is solidly insignificant (p=0.22-0.26). The authors indicate that there is a difference in the distribution of 5mC and 5hmC across the genome, but do not provide a statistical analysis of differential 5mC or 5hmC, or the relationship of differential 5mC or 5hmC to differential gene expression. There is insufficient evidence that differential 5mC or 5hmC are responsible for repression of *Idh2* or other genes in response to oxidative stress or eccentric heart failure. With specific reference to *Idh2*, can the authors target Tet using dCas9 to the relevant DMR intron within *Idh2* and show that this alters *Idh2* expression?

10. 5mC is widely reported as a transcriptional repressor. For the heatmap in Fig.6E, what does “combined” mean? The heatmap appears to show positive correlation between 5mC and gene expression, contrary to its canonical role as a transcriptional repressor. Why are the only regions with negative correlation “enhancer” and “OpenChrom”?

11. For Fig.S6A and S6B, the author showed TET1 expression increased in both male and female mice heart (but not DCM). But overall Tet activity (combined Tet1, Tet2, Tet3) remained unchanged. The conclusion “there was no significant difference in either the level of mRNA encoding Tet1-3 or its enzymatic activity” does not describe these data accurately. Could differences in activities of individual Tet enzymes, but no overall change in Tet1-3, be biologically significant? Is there a way to measure the tet1, 2 and 3 activities individually?

Minor

1. The authors should adhere to standard nomenclature convention for use of capitalization and italics for gene symbols. Instead, there are inconsistent uses of caps and italics for *IDH2* throughout the manuscript (e.g., “*IDH2*”, “*Idh2*” and “*ldh2*”).

2. For Fig.S1H, mitochondrial respiratory control ratio (RCR = state 3/state 4) is a more widely used readout to reflect mitochondrial function than the absolute respiratory rate.

3. Most of the experiments focused on the *IDH2* expression and related mechanism. The title is too general and does not reflect the main experiments and results of this manuscript.

4. Please add genome coordinates for Fig.6F.

Reviewer #3 (Remarks to the Author):

This manuscript from ElBeck et al. examines a role for the epigenetic regulation of IDH2 expression in the regulation of redox homeostasis in the heart. The authors initially note that IDH2 is downregulated in humans and mouse models of eccentric hypertrophy, but also follow an oxidative stress challenge with H₂O₂ in NRVMs. Interestingly, the authors find that IDH2 downregulation is associated with an increase in activity, which appears to serve antioxidant functions. Furthermore, the authors find an association between epigenetic marks and IDH2 expression, suggesting the potential for epigenetic control of redox homeostasis. This study is important and of interest to the field, but a number of issues need to be addressed before this manuscript is suitable for publication.

1. The authors stress a role for IDH2 in antioxidant defense. However, the authors do not actually examine oxidative stress. The authors should consider examining ROS production and/or markers of oxidative stress in further support of an antioxidant role for IDH2.

2. One-carbon metabolism is critical for the generation of methyl donors for epigenetic regulation and for the generation of certain antioxidants. Do the authors examine one-carbon metabolism to see if this pathway is altered and/or impaired?

Response to reviewers' comments regarding manuscript NCOMMS-22-31381/ NCOMMS-22-31381A (Elbeck et al 2023)

Epigenetic modulators link mitochondrial redox homeostasis to cardiac function

Table of content:

	Page
• Highlighted changes in the updated manuscript	1
• Response to a common comment raised by all reviewers regarding measuring redox species	5
• Response to comments from reviewer 1	10
• Response to comments from reviewer 2	29
• Response to comments from reviewer 3	43

Reviewers' comments are highlighted in **black and bold font**, whereas authors' response is colored blue and in normal font.

We thank the reviewers for their careful consideration of our work, and all suggestions to improve it. Accordingly, we have now done a major revision and restructured the manuscript, which substantially strengthened our previous conclusions and improved the readability of the manuscript. We highlight below the changes that we made to the revised version, as well as we provide an auxiliary file, in which all changes were highlighted (Manuscript file with track-changes):

Figure 1:

- *IDH2* expression in human samples that was presented in Figures 1F and 1G were merged under new Fig. 1f.
- New data for cardiac *Idh2* expression in *Pln-R14^{Δ/Δ}* mice, myocardial ischemic model (MI) and diabetic cardiomyopathy (STZ model) were added under new Fig. 1g, together with cardiac *Idh2* expression in *Mlp^{-/-}* mice.
- Western blotting for *Idh2*, *Gapdh* and *Cox4-1* in *Mlp^{-/-}* hearts presented in old figure 1L was removed, whereas the quantification for the western blotting presented in the supplementary figure S1k was kept in Fig. 1h, but the data points corresponding to males and females were color coded.

Figure 2:

- qPCR analysis of *Hmox1*, *Nqo1* and *Osgin1* expression in LV myocardium of *Mlp*^{-/-} in old figures 2A-2C were moved to new Fig 4h.
- qPCR analysis of *Idh2*, *Hmox1*, *Nqo1* & *Osgin1* expression in H₂O₂-treated NRCMs in old figures 2D-2I are now under Fig 2a.
- Old figure 3D is now under Fig. 2b, old figures 3A and 3B are now under Fig. 2c, and old figure 3D is now under Fig. 2d.

Figure 3:

- Old figures 3E-3P are now under Fig. 3a-3j, in the exact same order.

Figure 4:

- Old figures 4A, 4B, 4D are now under Fig. 4a.
- Old figures 4E-4J were moved to Fig. 5.
- Fig. 4c depicts new data for *Idh2* expression in NRCMs treated with increasing doses of SF.
- Fig. 4d-4f depict new data for the measurement of redox species (GSH, GSSG and MDA) in human foreskin fibroblasts treated with increasing doses of SF.
- Fig. 4g depicts new data for western blotting of *Hmox1*, *Nqo1* and *Osgin1* in the LV of *Mlp*^{-/-}.
- Fig. 4h depicts qPCR analysis of *Hmox1*, *Nqo1* and *Osgin1* expression in LV myocardium of *Mlp*^{-/-}, which were previously presented under figures 2A-2C.
- Fig. 4d-4f depict new data for the measurement of redox species (GSH, GSSG and MDA) in the LV of *Mlp*^{-/-}.
- Canonical pathway analysis that was presented in old figure 4K was moved to Supplementary Fig S4j.

Figure 5:

- Old Figure 5 (Activation of Nrf2 improves cardiac function in male, but not female *Mlp*^{-/-} mice) is now Fig. 6.
- New Fig. 5a-5d are the same old figures 4G-4J with same order.
- New Fig. 5e is a merge of old figures 4E and 4F.

Figure 6:

- Old figure 6 is now Fig. 7.

- New Fig. 6a-6g are same as old figures 5A-5G, with same order. Canonical pathway analysis presented in Fig. 6a was shortened to the most relevant 4 pathways, to make space for new figures. The full list of enriched pathways is available in the Supplementary Data file 6.
- Figures 6h-6j depict new data for the measurement of redox species (GSH, GSSG and MDA) in the LV of *Mlp*^{-/-} treated with Nrf2 activator or scrambled control.
- Fig. 6k and 6l are the same as old figures 5H and 5I.

Figure 7:

- New Fig. 7a-7g are the same as old figures 6A-6I with the exact same order, but some panels were merged. Summary Fig. (7g) was simplified.
- The Spearman correlation presented in Fig. 7c was updated with a new analysis on the whole genome level, whereas the previous version included only randomly selected 1000 elements. The method section was also updated accordingly.

Supplementary figures were also updated according to the main figures, as following:

- Respiratory control ration (RCR) of freshly isolated mitochondria from the LV of *Mlp*^{-/-} mice were added under Supplementary Fig 1h. *Idh2* expression in enriched cardiomyocytes from transaortic banding (TAB) model was added under Supplementary Fig. S1I.
 - Supplementary Fig. 2a is a merge of previous figures S3A and S3B.
 - Supplementary Fig. 4a and 4b depict new data from NRCMs treated with increasing doses of SF and NAC. Whereas Supplementary Fig. 4c, 4d and 4d depict new data for Western blotting and qPCR analyses for NRF2, HMOX1, NQO1, OSGIN1 and caspase 3 in hFF1 cells treated with increasing doses of SF and 3mM of NAC. Supplementary Fig. 4F depicts measurement of GSH and GSSG level in hFF1 cells treated with 3mM NAC. Supplementary Fig. 4j was moved here from the old main figure 4K.
 - New Supplementary Fig. 5 is same as old Supplementary figure 4, but the order of the subfigures was restructured.
 - New Supplementary Fig. 6 is same as old Supplementary figure 5, with the exact same order.
 - New Supplementary Fig. 7 is same as old Supplementary figure 6, with the exact same order. Supplementary Fig. 7c was updated with the analysis on the whole genome level, whereas the previous version included only randomly selected 1000 elements. The method section is also updated accordingly.
- Supplementary Fig. 7f is a newly added figure.

- New Supplementary Fig. 8 is the same as old Supplementary figure 7. Supplementary Fig. 8a is updated with qPCR analysis of *Hmox1*, *Nqo1* and *Osgin1* levels in brain, liver and kidney of mice treated with single dose of AZ925, in addition to previous data from the heart. Old Supplementary figures 7b and 7c were moved to Supplementary Fig. 4b.

General changes:

Quantifications of Western blotting were plotted as fold change instead of relative intensities, as it makes it easier to observe changes when the control group is adjusted to 1. All *p*-values are still the same, as it does not get affected with calculating the fold change from the relative intensities. All raw values and calculations are provided in Supplementary Data files 3 and 4.

I- Response to a common comment raised by all reviewers regarding measuring redox species

Before responding to all individual concerns raised, we thought first to address a common concern raised by all three reviewers about including direct measurements of oxidative stress markers. We totally agree with all reviewers that direct measurements of redox species would be essential to support most conclusions inferred from transcriptomic profiling. We have now supported most previous conclusions with data from new measurements of reduced and oxidized glutathione (GSH and GSSG respectively) and malonaldehyde (MDA) from both *in vivo* and *in vitro* studies. However, we want also to highlight below our efforts to measure other redox species and the reliability of available methods.

Quantifying oxidative and antioxidative markers in heart tissue was extremely challenging. Most available methods that we tested did not yield reliable data or were not sensitive enough to determine minute differences between males and females. We needed to develop sensitive mass spectrometry-based methods to enable reliable quantifications of these species.

Initial trials:

- We initially tested Western blotting of protein bound malondialdehyde (MDA), which is a marker for lipid peroxidation and oxidative damage. Western blotting of MDA adducts is widely used in publications, and many companies claim to offer specific antibodies. We did Western blotting for protein bound MDA in protein extracts from male *Mlp*^{-/-} hearts with an antibody from Abcam (ab6463, Abcam), as shown in the figure below.

Figure legend: Western blotting of protein bound malondialdehyde in protein extracts from LV myocardium of male *Mlp*^{-/-} and littermate controls.

The blot showed only very few bands after long exposure to UV light when imaging the chemiluminescence signal (1.5 h) and did not show remarkable changes between WT and *Mlp*^{-/-} male mice. The very low numbers of detected bands and their very low intensities raised a concern about the specificity of the utilized antibody (ab6463, Abcam) and the reliability of this whole approach

with analyzing MDA adducts using antibodies. This antibody was later discontinued by Abcam without indicating the reason.

- We also tested Amplex™ UltraRed Reagent (A36006, Thermo Fisher Scientific) (AUR), which reacts with H_2O_2 to form a fluorescent product called resorufin. AUR is commonly used in biochemical reactions to monitor emission of H_2O_2 . We initially observed that adding AUR with horseradish peroxidase (HRP) to cardiac Idh2 activity kinetic assay resulted in higher resorufin formation in *Mlp*^{-/-} male mice in comparison to WT controls (see the figure below), suggesting initially higher H_2O_2 production rate in *Mlp*^{-/-} male. However, the presence of reductants such as NAD(P)H and GSH in the biological sample can erroneously result in high formation of resorufin, as the formed NAD(P)[•] and GS[•] radicals react with dissolved oxygen to yield superoxide radical ($O_2^{\bullet -}$), which is in turn dismutated to H_2O_2 by endogenous superoxide dismutases (SOD)¹⁻³. These undesired reactions make AUR unsuitable to measure endogenous levels of H_2O_2 in cardiac tissues and indicated that the higher formation of resorufin that we observed in the Idh2 activity assay was to some extent mediated by the NADPH produced by Idh2, and not by indigenously produced H_2O_2 . We also observed that adding NADPH to AUR reaction mix substantially increased the fluorescence signal of resorufin (data is not shown). Subsequent attempts to indirectly measure the emission of H_2O_2 in cardiac tissues by using a combination of catalase and SOD with AUR, as suggested by Votyakova et al 2004² did not work either. Catalase was supposed to prevent resorufin formation because it is faster in decomposing H_2O_2 to water before it reduces AUR, but we observed instead a substantial increase in the fluorescence signal of resorufin, particularly in the blank sample when catalase was added. In contrast, SOD did not appear to increase the fluorescence signal of resorufin in the reactions containing extracts from the tissue sample (see the figure below).

Figure legend: Artifacts induced by Amplex™ UltraRed (AUR) Reagent

A) Measuring resorufin formation coupled to *Idh2* activity assay in cardiac mitochondria isolated from frozen LV myocardium of male *Mlp*^{-/-} and littermate controls with or without the *Idh2*-activity-inhibitor N Ethylmaleimide (NEM). 5 μg of mitochondrial proteins were used in each reaction. 0.1mM AUR and 0.1 U/ml of HRP were added to the reaction mix containing (in mM: Tris-HCl 10 [pH 8.0], NADP⁺ 0.2, MgCl₂ 5, and isocitrate 2). **B)** Measuring resorufin formation in tissue extracts from LV myocardium of male *Mlp*^{-/-} and littermate WT controls in reaction mix containing (in mM: Tris-HCl 10 [pH 8.0], NADP⁺ 0.2, MgCl₂ 5, 0.1mM AUR and 0.1 U/ml of HRP), with/without catalase (10³ U/ml) or SOD (40U/ml). Dashed lines were added to mark the florescent value of 1x10⁷ to enable easier visual comparison.

- We then tried to utilize a spectrophotometric method to quantify free MDA utilizing a commercial kit (ALDetect™ Lipid Peroxidation assay kit, BML-AK170, Enzo). This kit utilizes a chromogenic reagent (N-methyl-2-phenylindole) that reacts with MDA to form a stable chromophore with maximal absorbance at 586 nm. We found that this kit was relatively good, but the indigenous concentration of MDA in heart tissues was on the boarder of the lower quantification limit of this kit. This was due to both the low amounts of MDA in the heart and the presence of relatively huge interfering unspecific complexes with maximal absorbance at 540 nm and at 510 nm (See the figure below). The assay *per se* worked perfectly, as artificial induction of lipid peroxidation in cardiac samples with N-Ethylmaleimide (NEM) resulted in a high absorbance peak at the specific wavelength of 586 nm (See the figure below). Subsequent efforts to quantify total protein-bound MDA instead of the free one, which presents at higher amounts comparing to the free one, as well as utilizing dual wavelengths for the quantification to reduce optical interference, did not yield remarkable improvements neither. In conclusion, the very low amounts of MDA in the cardiac tissues of *Mlp*^{-/-} mice would not allow a reliable comparison between male and female *Mlp*^{-/-} hearts, or before and after the treatment with Keap1 inhibitor using this kit.

Figure legend: Spectrophotometric measurement of malondialdehyde (MDA) in the cardiac tissues of *Mlp*^{-/-} mice, with a commercial kit (ALDetect™ Lipid Peroxidation assay kit, BML-AK170, Enzo)

A) Measuring free MDA according to the manufacturer instructions **B)** Measuring protein bound MDA: Proteins were precipitated with 80% methanol after homogenizing cardiac tissue with PBS containing 5mM BHT, then 40 μl of PBS was added to the resulted pellet, with HCl and other reagents as described in the experimental procedure in the kit manual and heated to 45 °C **C)** Tissue were homogenized in water containing 5mM N-Ethylmaleimide for 20 min at room temperature to induce lipid peroxidation, then proteins were precipitated with 80% methanol, and the resulted pellet were subjected to the same procedure described in figure (B). Sample reactions in figure (C) were diluted 10 times before recording the absorption spectra, whereas the reaction for the standards were not diluted.

- We also tried to use a well-established kinetic assay for GSH and GSSG⁴. The method is based on determining total glutathione in one reaction, and the oxidized form in another reaction. Then the level of GSH is deduced indirectly by subtracting GSSG from total GSH. The assay is very sensitive for determining total glutathione but not for GSSG, due to its very low level in the biological samples, and thus requires large quantity of starting materials.

We instead thought that using a direct assay that can measure both GSH and GSSG from the same prepared sample would reduce the technical variation to detect small differences, and also allow to quantify GSH and GSSG from small amounts of cells. We therefore aimed to use a targeted mass spectrometric approach for quantifying GSH and GSSG. However, most available methods utilize lysis buffers with high salt content, which are incompatible with subsequent mass spectrometric analysis, and would require a desalting step that decreases the sensitivity and increases technical variations, and therefore we focused on developing a custom mass spectrometric approach.

Successful trials:

For GSH/GSSG analysis, we developed a custom method that lysed the tissue by repeated freezing and thawing cycles in the presence of only aquatic N-Ethylmaleimide (NEM) to derivatize GSH. Auto-oxidation of GSH or auto-reduction of GSSG was accounted for by spiking the tissue with two different isotopic internal standards, i.e., deuterated GSH-d5 and GSSG-¹³C₄,¹⁵N₂.

For MDA analysis, we adapted a method that utilizes sodium hydroxide to release protein-bound MDA in the plasma⁵, which we found that it was also capable in dissolving grinded cardiac tissues when heated to 60 °C. Released MDA was later derivatized with 2,4-Dinitrophenylhydrazine

Hydrochloride. Both targeted mass spectrometric methods that we developed for analyzing GSH/GSSG and MDA are described in detail in the extended Materials and methods section.

Results:

We integrated the results into the updated manuscript in Figure 4 and Figure 6, as shown below.

Figure 4

Figure 4

Figure 6

II- Response to individual reviewers' comments

1. Reviewer #1 (Remarks to the Author):

In this study, Elbeck et al reported altered IDH2 expression level in human and mouse cardiomyocytes with eccentric hypertrophic hearts. The authors tried to link the downregulation of IDH2 with cardiac antioxidant defense and further correlated the change with epigenetic regulation. Although the manuscript presented some interesting data, most conclusions were made from indirect evidence. The authors emphasized on the antioxidant role of various regulators and redox homeostasis, while there is no single direct evidence to demonstrate that the alteration of intracellular redox level is directly related to pathological conditions. Most conclusions were drawn from the correlation between gene expression changes but without experiments to demonstrate the causality. The logic flow of the manuscript is difficult to follow. The authors tried to cover the topic related to metabolic, transcriptional, and epigenetic changes; however, the manuscript lacks a central focus and in-depth insights. The authors are suggested to edit the language and reshape the manuscript.

1.1. Figure 1.

1.1.1. The rationale of selecting IDH2 is not clearly stated. There are many other genes showing similar expression patterns as IDH2 in Figure 1E. The authors do not provide a strong justification for selecting IDH2 as a downstream target.

Authors response: We edited the text to emphasize that *IDH2* was selected for downstream focus based on being the most quantitatively downregulated gene among genes encoding mitochondrial metabolic enzymes in failing genetic DCM human hearts, as it can be observed in (Fig. 1e).

"Intriguingly, IDH2 was the most downregulated gene in this intersection (Fig. 1e, f). Moreover, IDH2 was also significantly downregulated in all other transcriptomic data sets from failing genetic DCM, idiopathic DCM and ICM hearts, as well as from hearts of rodent models of genetic DCM, e.g., Mlp^{-/-} and mutated phospholamban Pln-R14^{Δ/Δ}; ischemic DCM, e.g., myocardial infarction⁶; diabetic DCM, e.g., rats treated with streptozotocin (STZ)⁷ (Fig. 1f, g and Supplementary fig. 1j)."

1.1.2. GAPDH is also involved in mitochondria metabolism. It would be better to use other proteins as a loading control. This also applies to other figure panels.

Authors response: We agree with the reviewer that GAPDH is involved in mitochondrial metabolism. However, it is mainly a cytoplasmic protein, and we tested several housekeeping proteins, from which we found that Gapdh was the most suitable housekeeping protein in murine cardiac tissue, but not in human heart tissue, as explained below.

A- Murine samples:

In the Western blotting for Hmxo1, Nqo1, and Osgin1 in male and female *Mlp*^{-/-} hearts, we compared normalizing band intensities of target proteins to Gapdh, total protein or β -actin, as shown in the figure below.

Figure I-1: Comparing normalizing target proteins to Gapdh, β -actin or total protein in Western blotting in murine cardiac samples: The plots and quantified target proteins normalized to Gapdh are also shown in fig. 4g. Detailed procedures are described in the extended method section.

The trend of regulation appeared to be very similar when target protein was normalized to Gapdh or total protein, but it differed substantially when it was normalized to β -actin, as the expression of β -actin *per se* is substantially increased in *Mlp*^{-/-} (β -actin panel is shown in the figure above).

We also tested other commonly used housekeeping proteins, such as α -tubulin, which appeared to be substantially elevated in *Mlp*^{-/-} hearts, and hence does not work as well as Gapdh as a housekeeping protein in murine cardiac tissues.

Figure I-2: Western blotting to validate whether α -tubulin can be used as a housekeeping protein in murine cardiac samples: The blot was simultaneously blotted with anti- α -tubulin (ab7291, Abcam), anti-Gapdh (mA5-15738, Invitrogen), anti-Idh2 (MA5-17271, Invitrogen), and anti-cox4-1 (ab16056, Abcam).

Total protein, obtained through ponceau S staining, appears also to be a suitable alternative for normalizing target proteins in murine cardiac tissue, as shown in figure I-1. However, quantifying total proteins is also prone to some artifacts, such as non-homogeneous excess dye present in certain parts of the membrane and inaccuracies in defining the exact border between two lanes (especially when adjacent lanes are not completely straight). Therefore, we think that both Gapdh and total protein were suitable controls for normalizing target protein from murine cardiac tissues, but we preferred Gapdh.

B- Human samples:

In the Western blotting of proteins from end-stage human failing hearts (**Fig. 1c**), we observed that there was a large variation in the pattern of GAPDH bands among different samples, even within the same genotype (control or HF). Thus, GAPDH does not work as a housekeeping control.

Figure I-3: Gapdh and Ponceau S staining panels extracted from Fig. 1c

Moreover, we could hardly detect any band for β -actin, as the total amount of loaded proteins were only 10 μ g in this experiment, and it showed no bands even after long exposure for UV light when imaging the chemiluminescence signal. In addition, β -tubulin was substantially increased in HF samples in comparison to controls. Therefore, neither GAPDH, nor β -actin, nor β -tubulin can be used for normalization.

Figure I-4: Western blotting to validate whether α -tubulin can be used as a housekeeping protein in human cardiac samples:

The blot was blotted with anti- β -tubulin (T4026, Sigma-Aldrich)

Therefore, in the human samples, total protein was more suitable for normalization than GAPDH; β -actin or β -tubulin.

C- NRCMs samples:

In Western blotting for Idh2 upon *in vitro* induction of L2HG in neonatal rat cardiomyocytes (NRCMs) presented in **Fig. 3g**, we initially blotted for Gapdh as a housekeeping protein, but we observed an artifact from the blotting procedure affecting the first two lanes. Therefore, we reblotted the membrane for β -actin, which exhibited no variation between the samples. The normalization against Gapdh or β -actin showed the exact same trend of regulation for Idh2 (Figure I-5 below). We chose to display the normalization to β -actin in **Fig. 3g** to be more rigorous, as we explained in Supplementary Data file 2.

Figure I-5: Idh2, Gapdh, and β -actin panels of Figure 3g:

We concluded that even the best suitable housekeeping protein could vary from one sample type to another, and one should therefore choose the housekeeping protein that shows lower variation than that of the investigated protein, as all cellular proteins might be affected in heart failure, directly or

indirectly. Even total protein might be affected in heart failure, as heart failure is associated with alterations in cellular composition of the heart, and in the extracellular matrix. Moreover, Western blotting is not a quantitative method, but rather semiquantitative, and therefore it always requires other types of experiments to support conclusions inferred from Western blotting. Therefore, to the best of our efforts, we think that we have used a correct control in each case.

1.1.3. Please specify the gender of human analysis.

Authors response: Sex is now indicated on **Fig.1c** beneath each lane as M for male and F for female. The sex of these samples was also indicated in Supplementary Table 1 in the extended material and method section, together with other characteristic information for these samples.

All other human data included in the manuscript were obtained from published data sources and we properly cited the source wherever these data were used, where the reader can find full information about the sex and other characteristics of these samples. All these data sets were from mixed sexes, but mostly from males, which reflects the prevalence of DCM. The percentage of male patients were 72% in data obtained from Sielemann et al 2020⁸, 81 % for DCM and 77% for ICM in data obtained from Sweet et al 2018⁹, and in 75% in data obtained from Van Heesch et al 2019¹⁰.

1.1.4. Please spell out the full name of DCM when it was first mentioned in the text file and figures.

Authors response: We fixed it as suggested.

1.1.5. It would be better to use a second irrelevant cardiomyopathy model to demonstrate that the downregulation of IDH2 is specific in eccentric hypertrophy.

Authors response: We thank the reviewer for these suggestions, and we completely agree that providing data for *IDH2* expression from other cardiomyopathy model(s) will strengthen our conclusions. In the old version, we provided data from hearts of human patients with idiopathic or genetic DCM, and from patients with ICM, some of whom also have diabetes. However, human patients usually have a mixed cardiac phenotype, and therefore mouse models might provide better insights. We previously only provided such data from a murine model of genetic DCM (*Mlp*^{-/-}), and from *Mybpc3*^{-/-} as genetic model of concentric hypertrophy.

We have now provided additional data from another genetic eccentric DCM model induced by deleting arginine 14 residue in phospholamban (Pln) (*Pln-R14^{ΔΔ}*)⁶, as well as from models of other cardiomyopathies with eccentric hypertrophy, such as models for myocardial infarction (MI)⁶ and diabetic cardiomyopathy induced by treatment with streptozotocin (STZ)⁷, all of which exhibit

downregulation in *Idh2* expression (**Fig. 1g**, also shown below). We also provided new data from a pressure overload model induced by transaortic banding (TAB), a second heart failure model with concentric hypertrophy. TAB model did not exhibit downregulations in *Idh2* expression, similar to *Mybpc3*^{-/-} (**Supplementary fig. 1l**, also shown below).

Figure I-6: Downregulation of *Idh2* expression in murine models of eccentric hypertrophy but not in concentric hypertrophy

Fig.1g: *Idh2* expression in the LV of models displaying eccentric hypertrophy, such as, *Mlp*^{-/-} mice analyzed by qPCR, or inferred from published transcriptomic data from LV of *Pln-R14*^{Δ/Δ} mice⁶; mice underwent myocardial infarction (MI) surgery 8 weeks after the surgery⁶; and rat model of diabetic cardiomyopathy induced by treatment with streptozotocin (STZ)⁷. **Fig. 1g and Supplementary fig. 1l:** *Idh2* expression in the LV of models displaying concentric hypertrophy, such as *Mybpc3*^{-/-} males analyzed by qPCR, or inferred from published transcriptomic data from enriched cardiomyocytes isolated from a pressure overload model induced by transaortic banding (TAB), 1 week or 8 weeks after banding¹¹.

It also can be noted from the pattern of *Idh2* expression with age in the *Pln-R14*^{Δ/Δ} model that homozygote mice do not exhibit any downregulation in *Idh2* when they are in a pre-DCM stage with no clear DCM phenotype at 3 weeks of age. However, once they exhibit a clear DCM phenotype at 5 weeks of age, *Idh2* get downregulated, and remains downregulated throughout the very late stage of heart failure at 8 weeks of age⁶.

These new sets of data strengthen our previous conclusion that “downregulation of *IDH2* appears to occur specifically in cardiomyopathies involving eccentric hypertrophy”.

1.2. Figure 2.

1.2.1. Western blot to evaluate protein levels of key targets mentioned in this figure is needed since RNA level is not necessarily aligned with protein levels. Furthermore, the authors observed significant reduction of total RNA with increased H2O2 treatment. The decreased expression of *IDH2* might simply be due to the reduction of total RNA.

Authors response: We agree with the reviewer that RNA level is not necessarily aligned with protein levels, and therefore we have now provided Western blotting for Hmox1, Nqo1, and Osgin1 from the

exact same animals (Fig. 4g). Western blotting demonstrated that the regulation of these three genes appears to be similar on the RNA and on the protein level.

Regarding the significant reduction of total RNA and its potential effect on decreased expression of IDH2: The reduction of total RNA observed with the increased doses of H₂O₂ treatment exerted no effect on the measurement of *Idh2* expression, because we employed same amounts of total RNA in all qPCR reactions, and we further normalized Cq values of target genes in every reaction to corresponding Cq value of *Gapdh*. We quantified total extracted RNA with Qubit, and then employed 5 ng of total RNA in all qPCR reactions, as we stated both in the manuscript and in the figure legend. Any potentially induced oxidative damage by H₂O₂ on the RNA molecules would affect both target genes and *Gapdh*, and thus it would be subsequently normalized.

Below we also show a screenshot for Cq values of target genes from neonatal cardiomyocytes treated with H₂O₂ for 1 day (Figure I-7). All Cq values of *Gapdh* remain largely unchanged by increasing the doses of H₂O₂, indicating equal amounts of employed total RNA in each reaction.

Sample	Idh2	Hmoa1	Hsp70	Osgin1	Gapdh
	C1	22.36	24.61	24.89	28.06
	22.49	24.62	24.87	28.10	19.13
C2	22.20	24.67	24.77	28.15	18.86
	22.28	24.90	24.75	28.47	19.19
C3	22.10	24.52	24.88	28.14	18.92
	22.09	24.62	25.04	28.29	18.99
C4	22.00	24.47	24.76	28.22	18.92
	22.12	24.88	25.00	28.41	18.86
25 μ M-1	22.07	24.72	24.76	28.31	18.96
	22.31	24.99	25.01	28.46	19.05
25 μ M-2	22.06	24.69	24.71	28.24	18.88
	22.16	24.84	25.19	28.48	19.12
25 μ M-3	22.21	24.86	24.74	28.31	19.09
	22.30	24.82	25.05	28.57	19.17
25 μ M-4	22.50	24.87	24.73	28.40	19.05
	22.49	24.99	24.91	28.30	19.01
50 μ M-1	22.93	24.06	23.99	28.11	18.76
	22.82	24.23	24.18	28.07	18.95
50 μ M-2	22.37	24.45	24.08	28.10	18.92
	22.48	24.30	24.39	28.25	19.13
50 μ M-3	22.28	24.27	24.18	28.28	18.80
	22.29	24.20	24.38	28.07	19.01
50 μ M-4	22.33	24.11	24.06	28.48	19.01
	22.50	24.12	24.25	28.32	19.11
100 μ M-1	23.88	22.58	22.96	27.96	19.12
	24.08	22.87	22.61	27.98	19.30
100 μ M-2	24.14	23.12	22.38	27.76	19.21
	24.16	22.63	22.45	28.04	19.37
100 μ M-3	24.07	22.66	23.04	27.97	19.25
	24.13	22.63	22.63	28.06	19.37
100 μ M-4	24.07	22.90	22.12	27.75	19.25
	24.24	22.58	22.13	27.65	19.09
200 μ M-1	25.00	21.55	21.20	27.22	18.90
	24.93	21.77	21.91	27.36	19.03
200 μ M-2	25.03	21.57	21.23	27.33	18.80
	24.98	21.81	21.43	27.34	19.08
200 μ M-3	24.57	21.42	21.19	27.31	18.84
	24.67	21.69	21.46	27.39	18.90
200 μ M-4	24.95	21.68	21.46	27.47	18.95
	25.33	21.91	21.68	27.42	19.00
300 μ M-1	25.48	21.25	21.16	26.77	18.61
	25.52	21.46	21.20	27.07	18.78
300 μ M-2	26.02	21.66	21.42	26.97	18.85
	26.07	21.78	21.51	27.07	19.07
300 μ M-3	25.84	21.39	21.23	26.96	18.61
	26.16	21.53	21.26	26.93	18.71
300 μ M-4	25.74	21.46	21.14	27.16	18.77
	26.02	21.49	21.24	27.35	18.65

Gapdh Cq \approx 19

Figure 2a

Figure I-7: Decreased expression of *IDH2* is not due to the reduction of total RNA

Screenshot from the raw Cq values of qPCR analysis of NRCMs treated for 24 hours with increasing doses of H₂O₂ presented in figure 2a. The two rows for each sample represent technical duplicates.

1.2.2. H₂O₂ treatment might result in cell death, especially at high concentrations of H₂O₂ treatment. The authors need to consider the cell conditions during the experiments. Working with unhealthy cells might result in biased and inaccurate conclusions.

Authors response: The reviewer is right, both H₂O₂ and SF treatment might result in oxidative stress, oxidative damage and eventually cell death. However, inducing oxidative damage and cell death depend on the potency of the oxidant (i.e., its redox potential and its concentration) and the capacity of the antioxidative system, and this was the actual propose of this experiment, which we have tried now to better clarify in the updated version. Cardiomyocytes exhibiting oxidative stress in heart failure are not healthy cells, and this is what we aimed to mimic and study in these experiments.

1.3. Figure 3.

1.3.1. The authors claim that increased IDH2 activity might be due to desuccinylation, however in the study that authors cited in this study (Zhou et al EMBO reports 2016) is compared to the IDH2 activity with and without succinylation at the same amount protein level. In this study, the authors observed significant downregulation of IDH2 at the protein level, but it lacked direct biomechanical evidence demonstrating that desuccinylation of IDH2 could overcome the decreased protein level and is responsible for increased IDH2 activity.

Authors response: The reviewer is right. In this experiment (Fig. 2c), we measured total subsarcolemmal IDH2 activity, which is the sum of both the activity of individual IDH2 molecules, and the total number of IDH2 molecules. The aim of measuring overall IDH2 activity here was to explain the observed increase in 2OG level, in which we showed that increased IDH2 activity corresponds to increase in 2OG level.

Defining exact contributing factors to IDH2 activity is interesting, but experimentally challenging, and it is not within the scope of our study (as we further explain in our response to comment 1.3.4). Succinylation¹² of IDH2 is not the only posttranslational modification that regulate its activity, but also glutathionylation¹³, and acetylation¹⁴. Characterizing their combined contributions in cellular context would require developing complicated targeted mass spectrometric approach that enable simultaneous quantification of all of them and subsequent *in vitro* studies to determine how their levels collectively affect IDH2 activity, and therefore it cannot be a side story to our work. It is most likely that glutathionylation also play a role here, as it increases IDH2 stability¹³ and it would explain, at least in part, observed discrepancies between *substantial* downregulation of *IDH2* transcript, and the *mild* downregulation in IDH2 protein that we observed in HF.

1.3.2. The presentation of Figure 3O is confusing. It is unclear what key information the authors would like to convey in this figure.

Authors response: The description of this experiment (**Fig. 3i**) was missing from the method part, and we have now added it. Upstream regulator analysis is a way to visualize the activation state of an upstream regulator molecule (here Nrf2). It plots the expression of all differentially expressed downstream targets of Nrf2 (green and red filled circles), and then computes expected activation state of Nrf2. In **Fig. 3i** in the updated manuscript (or figure 3O in previous version), most of the downstream targets of Nrf2, which are expected to be suppressed when Nrf2 is inactive, were downregulated when L2HG was induced by ShRNA targeting L2hgdh (green circles and blue connecting lines). Therefore, it can be concluded from this figure that Nrf2 gets inactivated when L2HG is induced.

1.3.3. The authors keep emphasizing that cellular redox level was altered. The conclusion is made from the alteration of expression of IDH2, L2HG and D2HG. There is no evidence directly showing the changes of cellular redox level.

Authors response: Our conclusion that the increase in IDH2 activity and ensuing elevations in 2OG and L2HG levels *suggested* that it enhances the antioxidative capacity of cardiomyocytes. This conclusion was made from different experiments and published studies. Oldham et al 2015¹⁵ reported that L2HG represent a source for reducing equivalents as it increases NADH/NAD⁺. We showed that L2HG increased IDH2 expression, which is a major source for cardiac mitochondrial NADPH^{16,17}, and it inactivated NRF2 (**fig. 3i**), which is a master regulator for cellular antioxidative response. In subsequent *in vitro* studies, we tried to study these effects in more details by using more potent inducers, i.e., SF and NAC, from which we provided direct measurements for redox species in the updated manuscript, as described in our response to the next comment (1.3.4).

1.3.4. In Figure 1 and Figure 3, the authors claim correlation between decreased Idh2 expression and increased L2HG in *Mlp*^{-/-} mouse than WT group. However, in the later study (Figure 3I-M), the authors knocked down L2hgdh and induced the increased L2HG, which then resulted in increased Idh2. These two results seemed to be in conflict. Please clarify this point.

Authors response: We agree with the reviewer that these results appear to conflict, but in fact they are not, as they describe two opposing mechanisms, which we have tried now to clarify more in the updated version of the manuscript, and now also with the results of direct measurements of oxidative and antioxidative species.

Mainly, we are discussing the effect of two competing mechanisms. The first one is a feedforward cycle involving 2OG and L2HG, which increases the expression and activity of IDH2 in response to oxidative stress. Whereas in the opposing mechanism, the response to oxidative stress induces NRF2 activation, which in turn tries to decrease the expression of IDH2 to limit overwhelming cells with antioxidants.

The feedforward cycle starts with oxidative stress increasing the activity of IDH2 by posttranslational modifications¹²⁻¹⁴, which results in an increase in 2OG that induces L2HG. L2HG in its turn increases the expression of IDH2, which together with the posttranslational modifications enhance the antioxidative contribution of IDH2. As a support for this mechanism, we showed that oxidative stress in *Mlp*^{-/-} hearts (**Fig. 4g, h**) was accompanied by an increase in *Idh2* activity (**Fig. 2c**), associated with an increase in 2OG (**Fig. 2d**), an increase in GSH (**Fig. 4i**), and a decrease in MDA as a marker for oxidative stress (**Fig. 4k**). The rise in cellular levels of 2OG and lactate while the decrease in the level of L-malate causes lactate (LDH) and malate dehydrogenases (MDH) to reduce 2OG to L2-hydroxyglutarate (L2HG)^{18,19}, which we showed to happen in *Mlp*^{-/-} hearts (**Fig. 3a, b and Supplementary fig.3a**). In the *in vitro* knockdown of *L2hgdh* with ShRNA targeting *L2hgdh*, we demonstrated that the induction of L2HG *per se* (i.e., without the presence of oxidative stress) induced the expression of IDH2 (**Fig. 3f, g**), therefore we concluded that observed accumulation of L2HG in heart failure upregulates IDH2.

As a support for the competing mechanism, which decreases *IDH2* expression, we showed that certain downstream targets of NRF2 were induced in *Mlp*^{-/-} hearts (**Fig. 4g, h**), suggesting some activation of this pathway. Therefore, we induced stronger activation of NRF2 *in vitro* in NRCMs with H₂O₂ and SF, which resulted in the downregulation of *IDH2* (**Fig. 2a, 4a, c**), whereas inducing the inactivation of NRF2 by NAC induced slight upregulation in *IDH2* (**Fig. 4a**). We further showed that *IDH2* downregulation induced by low doses of oxidants (**Fig. 4c**) was accompanied by an increase in GSH levels (**Fig. 4d, e**), but no elevation in lipid peroxidation (**Fig. 4f**), similar to the observations in the *Mlp*^{-/-} hearts (**Fig. 4i-k**). Moreover, we demonstrated that increasing Nrf2 activation *in vivo* by AZ925 increased GSH levels in *Mlp*^{-/-} hearts (**Fig. 6h**), and thus increased the antioxidative capacity, which induced further downregulation of *Idh2* (**Fig. 6k**), particularly in females as they originally exhibited higher antioxidative capacity (**Fig. 4i-k**). As there is no direct regulation between NRF2 and IDH2, we concluded from all above data that this mutual regulation between NRF2 and IDH2 is induced through regulating the antioxidative capacity, as IDH2 is the main source for NADPH necessary for GSH/GSSG regeneration^{16,17}, while NRF2 controls the expression of several enzymes that regulate the synthesis and regeneration of GSH²⁰.

1.3.5. The rationale of pointing to NRF2 as the downstream target is not clearly justified. Based on data shown in Figure 3N, other pathways also exhibited similar alterations.

Authors response: The reason of selecting NRF2 was its central role in the antioxidative response, as it regulates the expression of more the 1000 genes containing the antioxidative-response-element (ARE) in their promoters ²¹. Actually, many of the enriched pathways upon knocking down L2hgdh in (Fig. 3h, previously figure 3N) are either regulated by, or regulate, Nrf2 activation, such as glutathione redox reaction ²⁰, oxidative phosphorylation ²² and others. We have now clarified in the introduction that NRF2 represents a master regulator of the antioxidative response.

1.4. Figure 4.

1.4.1. The authors claimed that NAC upregulates Idh2 and reduces Nrf2 targets. The results do not agree with the conclusion. Based on data shown in Figure 4B, there was almost no changes of Idh2 expression in NAC treatment. Furthermore, not data was presented to demonstrate deactivation of Nrf2. Barely any changes were noted for three genes listed in Figure 4B under NAC treatment. Furthermore, the expression of these 3 genes was not sufficient to reflect Nrf2 activity. The nuclear translocation is the key to demonstrate the NRF2 activity.

Authors response: We kindly do not agree with the reviewer opinion. The graphs in this figure (Fig. 4a) show substantial downregulation of all three genes and in all time points of the treatments of NRCMs with NAC. The data presented in this figure represent fold change, and one has to compare the values to 1 to conclude the trend of regulation. We have now added a horizontal dashed line to mark the value of 1 on the Y axes in the updated manuscript, and we plotted below the graphs for the treatment of NRCM with NAC only to enlarge the range of the y axes. The graphs below (figure I-8) show clearly that 3 mM of NAC induces substantial downregulations in all three genes. The downregulation of these genes upon NAC treatment is also evident on the protein level, except for OSGIN1, whose signal was undetectable in basal non-activated state (see figure I-9 below).

Figure I-8: qPCR analysis of Hmox1, Nqo1, Osgin1 in NRCMs treated with 3mM of NAC.

The graphs were extracted from Figure 4a. Horizontal dashed lines mark the mean value 1 of the control samples.

Showing reduced levels of nuclear-translocated NRF2 upon the treatment with NAC by western blotting is experimentally challenging due to its extreme low levels in the basal non-activated state, which would become even lower when it is further inactivated by NAC. Moreover, available antibodies for NRF2 are not good, which makes detecting a specific band for NRF2 only feasible when NRF2 is stabilized with a potent inducer, such as SF, but not when it is inactivated by NAC.

There is a misconception in the field about the migratory and real molecular weight of NRF2. NRF2 has the molecular weight of ~55–65 KDa, but it runs on the SDS gel at ~95–110 KDa, most likely due to its abundant acidic residues. This misconception made most companies focusing on developing antibodies that detect a band at ~55–65 KDa, which resulted in a bias toward selecting antibodies with unspecific binding. A letter published by Lau et al in 2013²³ has nicely addressed this issue, and tested several commercial antibodies, all of them shows several unspecific bands together with a specific band for NRF2 at ~95–110 KDa. We also tested an antibody against NRF2 from Proteintech (16396-1-AP) on Western blotting of hFF1 cell line treated with increasing doses of SF (1,5, 10 μ M), or with 3 mM of NAC, and we could only detect a specific band for NRF2 at ~95–110 KDa, and only in cells treated with high doses of SF, but not in untreated cells, or cells treated with NAC. We loaded huge amount of total protein lysate on this gel (60 μ g), used high concentration of the antibody (1:500), and exposed the membrane for long time for UV light (~250 sec) when imaging the chemiluminescence signal.

Figure I-9: Western blotting of NRF2, HMOX1, NQO1, OSGIN1 in hFF1 cells treated with SF and NAC
 A schematic overview of the experimental design for this Western blotting is explained in Supplementary Fig. 4c.

Therefore, we do not think that Western blotting, immunoprecipitation, or any other antibody-based assay would give reliable data for analyzing nuclear translocation of NRF2 in cells treated with NAC, even with enriched nuclear fraction. Thus, inferring the activation state of NRF2 from the expression of downstream target of NRF2 is more reliable in this case, and all the three genes (*HMOX1*, *NQO1*, *OSGIN1*) have been experimentally well-validated to be directly regulated by NRF2²¹.

1.4.2. The authors emphasized that the changes were due to the antioxidative control mechanism, while there is not direct evidence to demonstrate that NAC or SF treatment indeed alters intracellular ROS.

Authors response: We totally agree with the reviewer that these data should have been presented in the manuscript to support our conclusions (Please see section I on page 5 of this letter). We explained the reason why the measurement for redox species were not included in the previous version and included them in the updated version, where they provided more direct evidence and supported our conclusions.

1.4.3. The authors observed reduction of cell numbers of SF and NAC treatment, suggesting toxicity of these drugs. The recovery of the cells might be due to short half-life and the inactivation of these drugs after 48 hr treatment. The authors claimed that “intracellular redox homeostasis is involved in regulating the morphology of CMs”. However, it might simply be due to the toxicity of this drugs. The experimental design is questionable.

Authors response: We agree with the reviewer that both SF and NAC induce cytotoxicity, but it is actually the aim of this experiment. We used SF and NAC to induce oxidative *stress* and reductive *stress*, respectively, which both are a form of *cytotoxicity*. Cardiomyocytes in heart failure exhibit redox stress, and this is what we aimed to mimic *in vitro* in these experiments to study cellular responses. Both SF and NAC are very commonly used compounds for these purposes.

We have now clarified better in the updated manuscript the role of redox in modulating the morphology of cardiomyocytes and added more data. These findings support previous indirect observations that antioxidants, such as NAC, activates ERK pathway^{24,25}, which in turn promotes concentric hypertrophy²⁶.

SF and NAC were freshly supplemented when changing the culturing media after 24 hours, not 48 hours, as illustrated in the experimental scheme and in the method part. However, these compounds are very likely to gradually degrade over time, or to be consumed by the cells, but after 24 hours, all three downstream genes of NRF2, were still induced with SF or reduced by NAC (**Fig. 4a, b and**

Supplementary fig. 4a), indicating that the cells were still responding to oxidative or reductive stress caused by these compounds.

Therefore, we consider that our experimental design answered adequately the aim of this experiment.

1.4.4. The authors mentioned gender difference in this study might be due to Nrf2 mediated response to oxidative stress. However, the conclusion is solely based on RNA-seq analysis which is not convincing. The authors need strong biochemical evidence to validate differential Nrf2 activity as well as oxidative stress in female and male. The authors also need strong experimental evidence to demonstrate the causal relationship between gender difference and oxidative stress.

Authors response: We thank the reviewer for these suggestions. We have now added more supportive data of western blotting and direct measurements of several redox species. We also restructured the manuscript to better clarify the relationship between sex difference and oxidative stress.

Mainly, sex differences in responses to oxidative stress which we previously observed on the mRNA level from *Mlp*^{-/-} heart (**Fig. 4h**) was also evident on the protein level (**Fig. 4g**), as males also show higher activation of Hmox1 than females, which indicates more acute response to oxidative stress in males. This observed difference was further supported by higher levels of both GSH and GSSG in female *Mlp*^{-/-} hearts (**Fig. 4i, j**). The increase in both GSH and GSSG indicates a reductive milieu in female *Mlp*^{-/-} hearts, and reminiscent of what we observed in cells treated with low doses of SF (**Fig. 4d, 4e**) or NAC (**Supplementary fig. 4f**). This was reflected on having substantially lower levels of MDA in female *Mlp*^{-/-} hearts (**Fig. 4i**).

The higher oxidative stress observed in male *Mlp*^{-/-} hearts was associated with a more severe DCM phenotype of male mice than females (**Fig. 5a-d, and Supplementary fig.5a, b**).

Consistent with the observations above, *in vivo* activation of Nrf2 by AZ925 revealed significant improvement of LVEF in male but not in female *Mlp*^{-/-} mice (**Fig. 6b-e**). The improvement in the cardiac function of males was further attested by a substantial increase in the levels of GSH in treated male *Mlp*^{-/-} hearts, but with no observable changes in GSSG or MDA levels (**Fig. 6h-j**). These results supported our observation that untreated *Mlp*^{-/-} females demonstrate less oxidative stress than males, and an increase in the reductive equivalents (**Fig. 4i-k**), and therefore additional exogenous induction of antioxidative response resulted in pronounced additional downregulation of *Idh2* in treated female mice (**Fig. 6k**).

We think that all above experimental data strongly attest a relationship between sex difference and oxidative stress.

1.5. Figure 5.

1.5.1. The entire figure is based on the usage of AZ925. However, there is no citation of this medicine and there is no data demonstrating that AZ925 could disrupt the NRF2 activity. The foundation of this experiment is questionable.

Authors response: We have now provided the reference for AZ925, which was under a pending patent application²⁷.

AZ925 was originally developed and validated for treating Alzheimer disease. Therefore, when we adapted it to induce NRF2 activation in the heart, we validated that it induced similar activation of NRF2 in the heart in a pilot study as described in the extended Materials and methods section. We measured the induction of selected downstream targets of NRF2 (*Hmox1*, *Nqo1*, *Osgin1*) in the heart, brain, liver and kidney after 9 hours of administration. The results showed that 10 mg/kg of AZ925 induced substantial elevation in the expression of these genes in all investigated organs. We provided previously the data for the heart in (**Supplementary fig. 8a**), but we have now provided data for all investigated organs in the updated version.

Figure I-10 (Supplementary Fig. 8): qPCR analysis for some downstream targets of Nrf2 (*Hmox1*, *Nqo1*, *Osgin1*) in the heart, brain, liver, and kidney from mice treated with one dose of 10 mg/kg of AZ925, investigated organs were dissected 9 hours after administration.

Moreover, it also appears clearly from the IPA Upstream analysis of the downstream targets of Nrf2 (**Fig. 6g**) that many downstream targets of Nrf2, which are predicted to be induced when Nrf2 is activated, were upregulated in Nrf2 activator treated mice (pink arrows with red-filled circles). In contrast, those genes that are predicted to be inactivated by Nrf2 were downregulated in Nrf2 activator treated mice (blue lines with green-filled circles).

Therefore, we think that these data provided direct evidence for the activation of NRF2 by AZ925, as it is claimed in the patent application, and there was no need to acquire more functional data for further validations.

1.6. Figure 6.

1.6.1. Many other dioxygenases are regulated by 2OG and L2HG. TET proteins are only one of the protein family that can be regulated by 2OG and L2HG. The rationale of focusing on TET enzyme is unclear. The DNA methylation and 5hmC analysis is not clearly stated. The low coverage (58.7% of mC and 4.4% of hmC) of CpG sites of DNA methylation and 5hmC analysis might introduce biased and inaccurate results.

Authors response: The numbers 58.7% of mC and 4.4% of hmC do not represent the coverage, but the actual percentages of these DNA modifications in the murine genome, i.e., among each 1000 CpGs, 587 CpGs are 5mC-methylated, 44 CpGs are 5hmC-methylated, and the rest 36.9% (1000 – [587 + 44] = 369) is unmethylated CpGs. The coverage was 38.7 million CpGs out of total of 43.8 million CpG sites in the murine genome, which is among the highest coverage one could get due to the presence of repetitive elements and other difficult-to-align regions in the genome.

The reviewer is correct about the regulatory role for 2OG/ L2HG on many other dioxygenases, and not only TET1-3 enzymes, which we also explained in the introduction. The main reason for focusing on 5hmC is the relative simplicity of measuring DNA methylations *in comparison* to histone modifications to address the concept of redox mediating mitochondrial-nuclear crosstalk. Histone modifications are much more divergent than DNA methylations, require laborious chromatin immunoprecipitation sequencing (ChIP-seq), and need a dedicated and separate work to address their regulation by 2OG/ L2HG. The second reason is the publicly available whole-genome oxidative bisulfite (OxBS) and bisulfite (BS) raw sequencing data with single-base pair resolution on *Mlp^{-/-}* hearts (SRA accession ID: PRJNA327790), which we clarified in the updated manuscript.

1.6.2. Although the authors identified DMRs or DHMRs in the study. It is unclear how these changes were directly correlated to phenotypes observed in an earlier study. Again, the causal relationship of altered epigenetic modifications with transcriptional outputs is unclear. The authors need to perform dCas9 based epigenome editing to dissect this causality.

Authors response: From the analysis of whole genome 5-mC and 5-hmC, we demonstrated that genes containing DMRs in their introns encoded a particularly high number of proteins involved in hypertrophic remodeling and regulation of redox homeostasis in *Mlp*^{-/-} (**Supplementary fig. 7e**). We also showed that an intronic region of *Idh2* genomic locus in *Mlp*^{-/-} harbored a DMR for 5hmC (**Fig. 7d**), indicating that 5hmC may play a role in regulating the expression of *Idh2* in heart failure. Moreover, in the updated manuscript, we showed that the genomic locus of *Idh2* is embedded in a 5hmC rich region (**Supplementary fig. 7f**), further attesting the regulatory role of 5hmC on *Idh2* expression. In addition, *in vitro* knocking down of Tet1/3 enzymes attenuated the induction of *Idh2* caused by elevated levels of L2HG (**Fig. 7e**), a mitochondrial metabolite which we observed to be induced due to metabolic remodeling associated with heart failure.

All these data provide strong evidence, but they remain correlative, as the reviewer indicated. Nevertheless, this is what is possible to obtain with the current technical advances in methylome sequencing. Using dCas9 to obtain more direct evidence in this case is not feasible for multiple reasons:

- Using dCas9 requires directing it to the exact genomic location of the methylated CpG. DNA methylation data presented in this work was computed in sliding windows of 15 adjacent CpGs, which makes it difficult to know the exact CpGs position affected on *Idh2* locus. This is because 5hmC is not directly measured and it is present at the genome at a very low level (only 4.4% of the CpGs is 5hmC in murine heart). 5hmC is indirectly inferred from subtracting the methylation levels in BS from OxBS libraries, and therefore it requires extremely high coverage to be concluded accurately. Obtaining such extremely high coverage at single base resolution for a proper sample size is technically difficult. Alternatively, an increase in the statistical power can be achieved through pooling of neighboring CpGs with a small compromise in the actual resolution²⁸, which what we did. To the best of our knowledge, the publicly available raw sequencing data of BS and OxBS that we analyzed represents the highest depth of coverage available for whole genome at single base pair resolution, with a sample size of 3 per genotype. Other direct sequencing technologies, such as 3rd generation sequencing are not in the stage to provide such data yet.
- Methylation is cell-type dependent, and we cannot use a cell line or cardiac primary cells with dCas9 for targeting *Idh2* locus *in vitro* instead of *in vivo* hearts. Moreover, TET1-3 enzymes are huge proteins, and the size of their mRNA range between ~ 7-11 Kb, which makes them difficult to pack in a viral vector together with a construct expressing dCas9 and guided RNA. This is in addition to the low transduction efficiency for the delivery viruses *in vivo* in the heart. Therefore,

using dCas9 *in vivo* requires a double transgenic mouse model, with both *Mlp* knockout and dCas9 knock-in.

- The heart contains several cell types, not only cardiomyocytes, where each cell type has its unique methylation profile.

Moreover, DNA methylation is not the only regulator for gene expression, as one would need also to look for histone modifications and for transcriptional factors. The presence of other cell types, other factors controlling gene expression, and the low efficacy of the *in vivo* delivery methods would dilute any effect induced by direct manipulation of 5hmC in *Idh2* locus and makes it difficult to measure.

1.6.3. Based on the example of genome-browser view shown in Figure 6F, only subtle changes of DNA methylation and hydroxymethylation were observed. It is unclear how these subtle changes would be responsible for downstream transcriptional changes. The similar concern applies to Figure 6G. Although there is statistical significance among all the analyzed groups, only less than 1.5-fold change of expression was noted, which might not have strong biological consequence.

Authors response: The 5hmC DMR that we highlighted in the genome-browser view shown in (Fig. 7d, previously Figure 6F) exhibits a huge decrease in 5hmC level (~3-4 fold) in *Mlp*^{-/-} hearts, hence it would be expected to have a marked effect on *Idh2* expression. There were even additional few more DMRs identified in *Idh2* locus, which were listed in Supplementary Data file 6. Moreover, the genetic locus of *Idh2* is embedded in a 5hmC-rich region (Supplementary fig. 7f, also shown in figure I-11 below), with 5hmC reaching ~15-20% in *Idh2* locus, in comparison to very low percentages of 5hmC on the genome-wide level (4.4%), indicating the importance of 5hmC in regulating *Idh2* expression.

In general, the low number of identified DMRs was due to the high threshold of difference between *Mlp*^{-/-} and WT that we set to define a DMR, by which we aimed to account for the very low percentage of 5hmC in the genome. Therefore, decreasing this threshold by increasing the sample size would certainly designate most small differences in the level of 5mC or 5hmC between *Mlp*^{-/-} and WT to be defined as DMRs. These differences could be observed in Fig. 7d, especially when we enlarged the X and Y axis of the genome-browser view in Supplementary fig. 7f (Figure I-11 below).

It is important to note that the differences that we report in Fig. 7a-c and Supplementary Fig. 7c, d are based on genome-wide averages as explained in the method part. Therefore, there will be plenty

of individual CpGs with profound differences between *Mlp*^{-/-} and WT that cannot be seen in these figures.

ne should also note here that 5hmC exists in the heart at intermediate levels comparing to other tissues, with only brain having slightly higher levels²⁹. Even with these low percentages, 5hmC has been widely reported to play important biological functions in almost all tissues. One has also to consider that observed effects of 5hmC are often masked, as methylation represents only one of several other factors that regulate gene expression, such as histone modifications and transcription factors.

In (fig. 7e, previously Figure 6F), the reported 1.5-fold change of expression is also not a small difference, many biological processes and diseases are associated with such, or even smaller, changes in gene expression.

Figure I-11 (Supplementary Fig. 7f): Screenshots from the Integrative Genomic Viewer for genomic 5hmC rich region harboring *ldh2* locus in chromosome 7 of the murine genome.

2. Reviewer 2

Re: “Epigenetic modulators link mitochondrial redox homeostasis to cardiac function”
The Zaher ElBeck et al. integrated several cardiac hypertrophic datasets and showed downregulation of *Idh2* but increased activity in cardiac eccentric hypertrophic. Increased oxidative stress by H₂O₂ or SF decreased IDH2, whereas reduced ROS upregulated *Idh2*. Nrf2 and *Idh2* exhibited mutually antagonistic regulation. ROS regulated *Idh2* through altered activity of Tet enzymes, which utilize *Idh2*'s product metabolite to demethylate DNA (5mC->5hmC). The authors demonstrate that heart failure altered 5mC and 5hmC distribution, including at *Idh2* itself, suggesting a mechanism by which ROS epigenetically regulates *Idh2* expression. This study provides some new insight into redox homeostasis during cardiac hypertrophy, especially about the regulation of *Idh2*, an essential component of the antioxidant system.

This paper was disjointed and hard to follow. The hypothesis and the logic of the supporting evidence should be more clear. Tangential side stories should be minimized or eliminated. Several lines of evidence are correlative and indirect. Overall the main points are not adequately supported by data.

2.1. Major:

2.1.1. “IDH2 is downregulated in eccentric hypertrophy.” This point is made from several datasets. IDH2 protein was downregulated when normalized to GAPDH (Fig. 1C) or β -actin (Fig. 3L-M). IDH2 is a mitochondrial matrix protein, while GAPDH is cytosolic. The mitochondrial matrix protein Cox4 was decreased in *Mlp*^{-/-} hearts (Fig. S1G) and ETC proteins were also decreased (Fig. S1F-G). Is IDH2 downregulated when normalized to a mitochondrial matrix protein? Why were different housekeeping genes used for normalization? IDH2 protein levels was analyzed in Fig. 1I-J and S1K-L – why are these analyses separated?

Authors response: The question regarding the loading control is a very interesting question, as employing a proper loading control for mitochondrial proteins is highly debated in the literature. IDH2 is a component of the TCA cycle in the mitochondrial matrix, whereas cytochrome c oxidase 4 (COX4) is component of the electron transfer chain (ETC), localized to mitochondrial inner membrane. The uncoupling between contractile function and TCA cycle activity, or between cardiac work and ETC activity, or both occur differently in different types of cardiomyopathies³⁰. Therefore, we do not think that COX4, as a component of ETC, represents an ideal loading control for a

mitochondrial matrix protein (IDH2), nor does GAPDH. GAPDH is mainly a cytoplasmic protein as the reviewer indicated, but also exists in mitochondria.

However, the main criterium of choosing a proper loading control is to be minimally affected by the condition being investigated. Therefore, we utilized different loading controls in different species based on experimental validations, as we explained with proper figures in responding to a comment raised by reviewer 1 (1.1.2) in this letter.

The downregulation of IDH2 on the protein levels does not appear to be as pronounced as it is on the RNA level, as we also indicated in the manuscript. This difference between the RNA and the protein levels *per se* indicate that IDH2 is differently regulated on the RNA and on the protein levels, i.e., there are two types of regulations (i) the mechanism that downregulate the expression of IDH2, and (ii) the compensatory mechanism that increase the level of the subsequent translated protein. However, in this work we focused mostly on the underlying reason for the downregulation of IDH2. We restructured the manuscript to make this point clearer.

The difference between the RNA and the protein may result from posttranslational modifications (PTMs) of IDH2 such as glutathionylation¹³, succinylation¹² and acetylation¹⁴, which all heavily regulate IDH2 stability and enzymatic activity in response to oxidative stress. We tried to investigate PTMs, and we found that there was more glutathionylation and less succinylation in *Mlp*^{-/-} (data for glutathionylation is not shown). However, understanding the impact of these modifications on the stability and activity of IDH2 in an *in vivo* setting is challenging, as it requires simultaneous profiling of all these modifications with a targeted mass spectrometric approach, and subsequent *in vitro* studies to interpret their effects. Therefore, we realized that it would make a major deviation from our main story, and it would dissipate the attention from our main story. However, the reader would immediately question the huge discrepancy observed between *IDH2* expression, its protein level, and its enzymatic activity, and therefore we provided few data to explain that posttranslational modifications may underly this observed discrepancy, and it is not due to artifacts in normalization of Western blotting or in the enzymatic activity assay.

In the updated manuscript, we fixed the issue highlighted by the reviewer regarding figures depicting the quantification of Western blotting band's intensities of *Idh2* in *Mlp*^{-/-} hearts (**Fig. 1i, and Supplementary Fig. 1k**, previously figures 1I-J and S1K-L). We separated them before to save some space in the main figure. Now, we used different color-codes to refer to data points corresponding to males and females, instead of plotting them separately. We thank the reviewer for pointing it out.

2.1.2. The author show that IDH2 mRNA and protein level, and IDH2 activity, are dissociated, i.e. expression level can decrease while activity increases. Yet most of the paper focuses on IDH2 level rather than activity. When the topic is upstream regulation of IDH2 expression, measurement of IDH2 level is appropriate. When the topic is IDH2 regulation of downstream pathways, its activity should be measured rather than making inferences based on its expression.

Authors response: We agree with the reviewer that it was not clear why we measured the activity of IDH2 in some experiments, and its expression level in other. We tried to clarify it in the updated manuscript.

Mainly, we examined either the activity of IDH2 or the expression level based on the precise effect that we were reporting. Kinetic IDH2 enzymatic activity assay reports alterations in IDH2 activity resulting from both posttranslational modifications of IDH2 molecules and changes in its protein quantity, without providing information about the contribution of each of them. Whereas expression analysis (qPCR and western blotting) would indicate how the activity could be affected based on altering the amount of the protein only, not through posttranslational modifications:

- In updated (**Fig. 2c**), we aimed to explore the overall function of *Idh2* as an antioxidative enzyme in *Mlp^{-/-}* hearts, therefore we performed *Idh2* activity assay and we subsequently measured its product level, i.e., 2-oxoglutarate.
- Conversely, in updated (**Fig. 3f, g**), we reported a novel redox mechanism that induced IDH2 by increasing its protein amounts, which was mediated by elevated L2HG, and therefore we used qPCR and Western blotting for IDH2.
- In updated (**fig. 2a, 4a**), we reported a mechanism that induced downregulation of IDH2 on the expression level upon oxidative stress, and therefore we reported the alteration on the RNA level. We did not report corresponding alterations on the protein levels in this case due to ethical considerations, as the observed reduction on *Idh2* levels was huge, and it was both time- and concentration-dependent, which would definitely be reflected on the protein level. These experiments were performed on primary neonatal rat cardiomyocytes (NRCMs), as IDH2 is highly expressed only in cardiomyocytes, and there is no left ventricular cell line available. RNA can be accurately and easily quantified using small amounts of starting materials than proteins. Preparing enough protein quantity from 64 samples for western blot analysis would require a high number of rat pups, but without providing substantial new information.
- Increased activity of IDH2 by posttranslational modifications upon oxidative stress was not something novel that we reported, but rather we cited previous works¹²⁻¹⁴.

- In (fig. 6k), we reported that the impact of increasing the antioxidative capacity of the heart induced further *IDH2* downregulation in *Mlp*^{-/-} hearts on the expression level, and therefore we provided expression data for *Idh2*.

2.1.3. The author knocked down the L2hgdh and showed “accumulation of L2-hydroxyglutarate associated with heart failure upregulates Idh2” in NRCMs. But in the *Mlp*^{-/-} mice heart, L2HG was increased (Fig.3F) but with decreased *Idh2* (Fig.1F-1K). These results appear to conflict. There are several pathways which can regulate *Idh2* expression, but which are most important to decrease IDH2 in eccentric heart failure, exemplified by *Mlp*^{-/-} mice?

Authors response: We agree with the reviewer that these results appear to conflict, but in fact they are not, as they describe two opposing mechanisms, which we have tried now to clarify more in the updated version of the manuscript, and now also with the results of direct measurements of oxidative and antioxidative species.

Mainly, we are discussing the effect of two competing mechanisms. The first one is a feedforward cycle involving 2OG and L2HG, which increases the expression and activity of IDH2 in response to oxidative stress. Whereas, in the opposing mechanism, the response to oxidative stress induces NRF2 activation, which in turn decreases the expression of IDH2 to limit overwhelming cells with antioxidants.

The feedforward cycle starts with oxidative stress increasing the activity of IDH2 by posttranslational modifications¹²⁻¹⁴, which results in an increase in 2OG that induces L2HG. L2HG in its turn increases the expression of IDH2, which together with the posttranslational modifications enhance the antioxidative contribution of IDH2. As a support for this mechanism, we showed that oxidative stress in *Mlp*^{-/-} hearts (Fig. 4g, h) was accompanied by an increase in *Idh2* activity (Fig. 2c), associated with an increase in 2OG (Fig. 2d), an increase in GSH (Fig. 4i), and a decrease in MDA levels as a marker for oxidative stress (Fig. 4f). The rise in cellular levels of 2OG and lactate while the decrease in the level of L-malate causes lactate (LDH) and malate dehydrogenases (MDH) to reduce 2OG to L2-hydroxyglutarate (L2HG)^{18,19}, which we showed to happen in *Mlp*^{-/-} hearts (Fig. 3a, b, and Supplementary fig. 3a). In the *in vitro* knockdown of L2hgdh with ShRNA targeting L2hgdh, we demonstrated that this induction of L2HG *per se* (i.e., without the presence of oxidative stress) induced the expression of IDH2 (Fig. 3f, 3g), therefore we concluded that the accumulation of L2HG in heart failure upregulates IDH2.

As a support for the competing mechanism, which decreases *IDH2* expression, we showed that certain downstream targets of NRF2 were induced in *Mlp*^{-/-} hearts (Fig. 4g, h), suggesting some activation of this pathway. Therefore, we induced stronger activation of NRF2 *in vitro* in NRCMs with H₂O₂ and SF, which resulted in the downregulation of *IDH2* (Fig. 2a, 4a, c), whereas inducing the inactivation of NRF2 by NAC induced slight upregulation in *IDH2* (Fig. 4a). We showed that *IDH2* downregulation induced by low doses of oxidants (Fig. 4c) was accompanied by an increase in GSH levels (Fig. 4d, e), but no elevation in lipid peroxidation (Fig. 4f), similar to the observations in *Mlp*^{-/-} hearts (Fig. 4i-k). Moreover, we demonstrated that increasing Nrf2 activation *in vivo* by AZ925 increased GSH levels in *Mlp*^{-/-} hearts (Fig. 6h), and thus increased the antioxidative capacity, which induced further downregulation of *Idh2* (Fig. 6k), particularly in females as they originally exhibited higher antioxidative capacity (Fig. 4i-k). As there is no direct regulation between NRF2 and IDH2, we concluded from all above data that this mutual regulation between NRF2 and IDH2 is induced through regulating the antioxidative capacity, as IDH2 is the main source for NADPH necessary for GSH/GSSG regeneration^{16,17}, while NRF2 controls the expression of several enzymes that regulate the synthesis and regeneration of GSH²⁰.

2.1.4. What is the relevance of “intracellular redox homeostasis modulates the morphology of cardiomyocytes” to the focus of the manuscript? Many studies show ROS is elevated in failing hearts, which are often hypertrophied. However, SF (increased ROS) led to cardiomyocyte atrophy and NAC (reduced ROS) promoted cellular hypertrophy. This seems contrary to what would be expected based on elevated ROS in cardiac hypertrophy and failure.

Authors response: Our observations actually support previous indirect studies indicating that ROS promote cellular hypertrophy. Activation of MEK1-ERK1/2 signaling mediates concentric growth (width increase) of cardiomyocytes, whereas inhibiting ERK1/2 induces cardiomyocytes lengthening²⁶. Moreover, in DCM or HCM, MEK1-ERK1/2 signaling also affects the tension generated by the sarcomeres through calcium dependent excitation-contraction coupling³¹.

NAC activates MEK1-ERK1/2 signaling^{24,25}, and thus it would be expected to promote the addition of sarcomeres in parallel to thicken the cardiomyocytes, similar to what was observed in our data (Fig. 5e). Although it would be expected that ROS would have the opposite effect, the actual mediator of this effect is most likely not ROS *per se*, but rather the capacity of the cellular antioxidative system. Chronic and low concentrations of ROS induced oxidative stress but without overriding the capacity of the antioxidative system (Fig. 4d, e, and Supplementary fig. 4a, c, d), or inducing observable oxidative damage (Fig. 4f). Even with a relatively high concentration of SF (i.e., 5 μm), which

induced substantial cell death (**Fig. 4a, and Supplementary fig. 4b, c, d**), surviving cells maintained substantial levels of GSH and normal levels of MDA (**Fig. 4d, f**). This has also been observed in *ex vivo* ischemic/reperfused hearts, which maintain substantial levels of GSH and capable of scavenging free radicals formed without substantial induction of MDA. Formation of MDA would rather require extreme oxidative condition to be induced, such as perfusing the heart with cumene hydroperoxide³². Therefore, the activation of the antioxidative response by the relatively low concentrations of ROS, such as 1 μ M of SF, or 25 μ M of H₂O₂, which leads to a substantial increase in reductants in the cells (**Fig. 4d**), would promote a hypertrophic phenotype similar to the effect of NAC. Such hypertrophic effect of 1 μ M of SF on NRCM can be observed in our data in the slight increase of recovered RNA amounts (**Supplementary fig. 4b**).

The redox state of cells treated with 1 μ M of SF recapitulated the state observed in the left ventricles of *Mlp*^{-/-} animals, and particularly in females, whereas male hearts showed slightly higher oxidative stress, but without overriding the antioxidative capacity (**Fig. 4g-k**). Female *Mlp*^{-/-} hearts showed a substantial induction in GSH, reflected on a substantial reduction in MDA (**Fig. 4k**) and a higher activation of ERK/MAPK pathway (Figure II-1). These observations may link the observed cardiac hypertrophy in *Mlp*^{-/-} females to the redox state, as the activation of MEK1-ERK1/2 signaling is expected to mediate concentric growth²⁶.

Figure II-1: Ingenuity Pathways comparison analysis on transcriptomic data sets from the LV of male and female *Mlp*^{-/-} in comparison to WT, or from the LV of male and female *Mlp*^{-/-} treated with Nrf2 activator AZ925 in comparison to scrambled treated *Mlp*^{-/-} controls.

Furthermore, when *Mlp*^{-/-} mice were treated with Nrf2 activator, male *Mlp*^{-/-} hearts, which had less basal activation of GSH (**Fig. 4i, k**), exhibited substantial improvement in their function (**Fig. 6b, c**). This improvement was associated with an increase in the GSH level (**Fig. 6**), increased activation of ERK/MAPK pathway (Figure II-1) and thickening in the left ventricle walls (**Supplementary fig. 6a**). Whereas female *Mlp*^{-/-}, which had already elevated basal levels of GSH, did not benefit from such treatment, but rather exhibited adverse effects preventing further thickening of the left ventricle walls (**Supplementary fig. 6a**).

Therefore, our data may not contradict previous reports, but rather support and clarify them. Moreover, the correlation between redox homeostasis and the morphology of cardiomyocytes

provides functional and important insights that link redox homeostasis to sex dimorphism, as explained above. However, this important concept requires a dedicated study to unravel the precise underlying mechanisms.

2.1.5. In several sections, inferences from transcriptomic analyses are relied on rather than direct measurement. For example, IPA enrichment analyses of gene expression are used to argue that manipulations affected redox homeostasis. However, direct measurements, such as biochemical markers of ROS or direct measurement of ROS levels, are not made. Since IDH2 is mainly responsible for generation of mitochondrial NADPH, a necessary cofactor for antioxidative processes, the NADP⁺/NADPH ratio could be another readout for redox homeostasis.

Authors response: We thank the reviewer for these suggestions. We provided direct measurement of redox species levels in the updated manuscript (Please see section I on page 5 of this letter). We think that these new data helped to explain many of the concerns raised.

We chose to quantify GSH instead of NADPH as GSH is a highly abundant reductant in all cells, including cardiomyocytes and prevalent at high intracellular concentrations (0.1-10mM) ³³. Moreover, GSH links the antioxidative role of IDH2 and NRF2, as many of the genes involved in antioxidative defenses utilizing glutathione are induced by NRF2 ³⁴, and in cardiomyocytes, NADPH that is required to regenerate the reduced form of glutathione (GSH) is provided primarily by IDH2 and downstream OGDH coupling to NNT in mitochondria ^{16,17}. In addition, direct measurements of NADPH are more challenging than GSH.

2.1.6. The causal link between impaired systolic dysfunction in *Mlp*^{-/-} male mice and insufficient ROS defenses is not well established. *Mlp*^{-/-} male mice have more severely impaired systolic function than females. This correlated with transcriptomic profiles more enriched for oxidative stress responses in males. Even if elevated ROS levels were measured (point 4), this would remain a correlation rather than demonstration of a less adequate oxidative stress defense in males. The conclusion was also supported by improvement of heart function by activation of NRF2 in males. However, NRF2 has multiple roles, not just regulation of ROS defenses. For example, NRF2 also influences mitochondrial respiration and fatty acid oxidation.

Author response: With the new direct measurements of the antioxidative and oxidative species and the *in vitro* functional studies on NRCMs treated with SF and NAC, we clarified in comment 2.1.4 how redox homeostasis is linked to impaired systolic dysfunction in *Mlp*^{-/-} male and female mice. The

results of direct measurements of antioxidative and oxidative species supported our previous conclusions drawn from transcriptomic profiling.

NRF2 is a key transcription factor that regulates the transcription of more than 1000 genes containing antioxidative-response-element (ARE). Therefore, the reviewer is correct that NRF2 has multiple roles, but actually all of which are redox related. ROS, and other reactive species, induce posttranslational modifications on NRF2 and KEAP1, which lead to dissociation of NRF2 from KEAP1, and subsequent nuclear translocation of NRF2 to activate the transcription of NRF2 downstream target genes³⁵. Reactive species are not a homogenous group of metabolites, neither they are unwanted side products of biological processes, but they are instead important metabolites that regulate numerous cellular signaling, posttranslational modifications, and enzymatic activities.

2.1.7. The gender dimorphism of *Mip*^{-/-} mice is interesting but it is a side story. The male mice support the main line of logic of the manuscript. This could be presented first, then the female mice could be presented second, with their preserved heart function, relatively lower ROS levels, and lack of benefit from NRF2 activation providing correlative supportive evidence.

Authors response: With providing new data for the directed measurements of redox metabolites, which supported observed differences in the cardiac phenotype between males and females, we demonstrate that sex dimorphism is not a side story, as explained in a previous comment (comment 2.1.4). These data also provided direct support to our main hypothesis that elevated antioxidative capacity underlies observed downregulation of IDH2 in eccentric DCM (**Fig. 6k**). Moreover, sex dimorphism explains our main clinical implication for this work, on why heart failure patients may or may not benefit from an antioxidative treatment, based on the capacity of their endogenous antioxidative system.

We agree with the reviewer that this part was confusing and needed restructuring, which we tried to improve in the updated manuscript.

2.1.8. The authors should provide biochemical evidence that AZ925 activates NRF2. For instance, immunostaining and subcellular fractionation to show NRF2 nuclear localization, and reporter assays to show NRF2 transcriptional activation.

Authors response: We have now provided the reference for AZ925, which was under a pending patent application²⁷.

AZ925 was originally developed and validated for treating Alzheimer disease. Therefore, when we adapted it to induce NRF2 activation in the heart, we validated that it induced similar activation of

NRF2 in the heart in a pilot study as described in the extended material and method section. We measured the induction of selected downstream targets of NRF2 (*Hmox1*, *Nqo1*, *Osgin1*) in the heart, brain, liver, and kidney after 9 hours of administration. The results showed that 10 mg/kg of AZ925 induced substantial elevation in the expression of these genes in all investigated organs. We previously provided the data for the heart in (Supplementary fig. 8a), but we have now provided data for all investigated organs in the updated version.

Figure II-2 (Supplementary Fig. 8a): qPCR analysis for certain downstream targets of Nrf2, (*Hmox1*, *Nqo1*, *Osgin1*) in the heart, brain, liver, and kidney from mice treated with one dose of 10 mg/kg of AZ925, investigated organs were dissected 9 hours after administration.

It also appears clearly from the IPA Upstream analysis of the downstream targets of Nrf2 (Fig. 6g) that many of downstream targets of Nrf2, which are predicted to be induced when Nrf2 is activated, were upregulated in Nrf2 activator treated mice (pink arrows with red-filled circles). Whereas those genes that are predicted to be inactivated by Nrf2 were downregulated in Nrf2 activator treated mice (blue lines with green-filled circles).

Therefore, we think that these data provided direct biochemical evidence for the activation of NRF2 by AZ925, as it is claimed in the patent application, and there was no need to acquire more functional data for further validations.

2.1.9. The authors argue that changes in 5mC and 5hmC in *Mlp*^{-/-} mice are responsible for downregulation of *Idh2* in response to oxidative stress. This is not sufficiently supported by data. In 6A-B, small, statistically insignificant changes are described as showing a “trend towards increased levels of 5mC and fewer 5hmC”. The magnitude of change is small and the pvalue is solidly insignificant (p=0.22-0.26). The authors indicate that there is a difference in the distribution of 5mC and 5hmC across the genome, but do not provide a statistical analysis of differential 5mC or 5hmC, or the relationship of differential 5mC or 5hmC to differential gene expression. There is insufficient evidence that differential 5mC or 5hmC are responsible for repression of *Idh2* or other genes in response to oxidative stress or eccentric heart failure. With specific reference to *Idh2*, can the authors

target Tet using dCas9 to the relevant DMR intron within *Idh2* and show that this alters *Idh2* expression?

Authors response: The non-significant effect in (**Fig. 7a**, previously figures 6A-6B) can be explained by testing only 6 samples (3 vs. 3) which is very hard to expect to be statistically significant with data inferred from whole genome bisulfide/oxidized bisulfide sequencing (BS and OxBS). However, the non-overlapping standard error of mean (SEM) in both figures indicate that increasing the sample size would result in significant *p*-values. However, we only described the trend of regulation of both 5-mC and 5-hmC, and we did not claim that the differences are statically significant.

In (**Supplementary fig. 7c**, previously S6C), we do not report any *p*-values or error bars because in this particular case they may be misleading in our opinion. In fact, t-test and Mann-Whitney U test result in significant difference with *p*-value < 2.2e-16 between WT and *Mlp*^{-/-}, but we believe that this should be treated with caution. This is because the number of statistical observations (N, the number of CpG sites) used to calculate SEM (t-value is computed as mean / SEM) is extremely large, in fact, in our case on the order of hundreds of thousands. The large number of observations used for statistical test results in very small SEM values and in turn significant *p*-values. This is a general property of *p*-value that can become significant almost certainly no matter how subtle the difference between two groups is being tested once the sample size becomes large enough. For this reason, *p*-values as a tool for making decisions have been extensively criticized³⁴. Here, we only describe the trend of regulation of both 5-mC and 5-hmC in this figure, and we do not claim that the differences are statically significant.

Regarding the comments that there is insufficient evidence that differential 5mC or 5hmC are responsible for repression of *Idh2* or other genes in response to oxidative stress or eccentric heart failure:

From the analysis of whole genome 5-mC and 5-hmC, we demonstrated that genes containing DMRs in their introns encoded a particularly high number of proteins involved in hypertrophic remodeling and regulation of redox homeostasis in *Mlp*^{-/-} (**Supplementary fig. 7e**). We also showed that an intronic region of *Idh2* genomic locus in *Mlp*^{-/-} harbored a DMR for 5hmC, which exhibited a huge decrease in 5hmC level (~3-4 fold) (**Fig. 7d**), indicating that 5hmC may play a role in regulating the expression of *Idh2* in heart failure. Moreover, in the updated manuscript, we showed that the genomic locus of *Idh2* was embedded in a 5hmC rich region (**Supplementary fig. 7f**), further attesting the regulatory role of 5hmC on *Idh2* expression. In addition, *in vitro* knocking down of Tet1/3 enzymes attenuated the induction of *Idh2* caused by elevated levels of L2HG (**Fig. 7e**), a mitochondrial

metabolite which we observed to be induced due to the metabolic remodeling associated with heart failure.

Figure II-3 (Supplementary Fig 7f): Screenshots from the Integrative Genomic Viewer for genomic 5hmC rich region harboring *Idh2* locus in chromosome 7 of the murine genome.

All these data provide strong evidence, but they remain correlative, as the reviewer indicated. Nevertheless, this is what is possible to obtain with the current technical advances in methylome sequencing. Using dCas9 to obtain more direct evidence in this case is not feasible for multiple reasons:

- Using dCas9 requires directing it to the exact genomic location of the methylated CpG. DNA methylation data presented in this work was computed in sliding windows of 15 adjacent CpGs, which makes it difficult to know the exact CpGs position affected on *Idh2* locus. This is because 5hmC level is not directly measured and it is present at the genome at a very low level (only 4.4% of the CpGs is 5hmC in murine heart). 5hmC is indirectly inferred from subtracting the methylation levels in BS from OxBS libraries, and therefore it requires extremely high coverage

to be concluded accurately. Obtaining such extremely high coverage at single base resolution for a proper sample size is technically challenging. Alternatively, an increase in the statistical power can be achieved through pooling of neighboring CpGs with a small compromise in the actual resolution²⁸, which what we did. To the best of our knowledge, the publicly available raw sequencing data of BS and OxBS that we analyzed represents the highest depth of coverage available for whole genome at single base pair resolution, with a sample size of 3 per genotype. Other direct sequencing technologies, such as 3rd generation sequencing are not in the stage to provide such data yet.

- Methylation is cell-type dependent, and we cannot use a cell line or cardiac primary cells with dCas9 for targeting *Idh2* locus *in vitro* instead of *in vivo* hearts. Moreover, TET1-3 enzymes are large proteins, and the size of their mRNA range between ~ 7-11 Kb, which makes them difficult to pack in a viral vector together with a construct expressing dCas9 and guided RNA. This is in addition to the low transduction efficiency for the delivery viruses *in vivo* in the heart. Therefore, using dCas9 *in vivo* requires a double transgenic mouse model, with both *Mip* knocked-out and dCas9 knocked-in.
- The heart contains several cell types, not only cardiomyocytes, where each cell type has its unique methylation profile.

Moreover, DNA methylation is not the only regulator for gene expression, as one would need also to look for histone modifications and for transcriptional factors. The presence of other cell types, other factors controlling gene expression, and the low efficacy of the *in vivo* delivery methods would dilute any effect induced by direct manipulation of 5hmC in *Idh2* locus and makes it difficult to measure.

2.1.10. 5mC is widely reported as a transcriptional repressor. For the heatmap in Fig.6E, what does “combined” mean? The heatmap appears to show positive correlation between 5mC and gene expression, contrary to its canonical role as a transcriptional repressor. Why are the only regions with negative correlation “enhancer” and “OpenChrom”?

Authors response: We updated (fig. 7c, previously figure 6E) to include whole genome, as previous version included only randomly selected 1000 elements, due to the extensive computing power that was needed to achieve the calculation on the whole genome level.

5mC is reported as transcription repressor when it presents in the CpG islands of the promoter region, but it has different functions when it presents in other genetic regions. The function of DNA

methylation is highly depended on the genetic region and the cell type³⁶ as well as the methylation type, as they may attract or repel different expression factors, repressors or other functional proteins³⁷. For example, methylation of the promoter is associated with repressed gene expression, but with increased gene expression when present in the gene body³⁸.

The Spearman correlation in (**fig. 7c**) was calculated between the expression of individual genes and the proportion of methylation in each adjacent functional region of individual samples. The values are the average of three biological replicates in WT and *Mlp*^{-/-}, or 6 replicates in *combined* WT and *Mlp*^{-/-}. We have now updated the figure legend with this missing information.

As we explained in the previous comment, the presence of several cell types in the heart, and several regulators of gene expression would make the general impact of methylation alone on gene expression small. This can be observed from the relatively small Spearman's rank correlation coefficient (ρ) in most regions (as shown on the color scale on **fig. 7c**), and therefore we only focused on the intronic region which appears to have the highest ρ value. Thus, results of Spearman's rank correlation for other regions require additional analysis and experiments to confirm their regulation trend. However, they may still be informative, as for example previous reports suggested that 5hmC might promote enhancer activity³⁹, which agrees with the different trend of correlations observed for 5mC and 5hmC in the enhancer region in (**fig. 7c**).

2.1.11. For Fig.S6A and S6B, the author showed TET1 expression increased in both male and female mice heart (but not DCM). But overall Tet activity (combined Tet1, Tet2, Tet3) remained unchanged. The conclusion “there was no significant difference in either the level of mRNA encoding Tet1-3 or its enzymatic activity” does not describe these data accurately. Could differences in activities of individual Tet enzymes, but no overall change in Tet1-3, be biologically significant? Is there a way to measure the tet1, 2 and 3 activities individually?

Authors response: Increased *Tet1* expression in *Mlp*^{-/-} observed in the heatmap in (**Supplementary fig. 7a**, old Figure S6A) is statistically insignificant, and the observed differences between *Mlp*^{-/-} and WT are very small (figure II-4). We have now added *p*-values on the heatmap and added another data set from *Mlp*^{-/-} male hearts (data set 2).

Figure II-4 (Supplementary Fig. 7a): Expression of TET1,2,3 genes in the LV of patients with DCM¹⁰ and male and female *Mlp*^{-/-} mice analyzed by RNA sequencing. Data set 1 of *Mlp*^{-/-} mice were produced for this work, whereas data set 2 of *Mlp*^{-/-} mice was analyzed from the publicly available raw sequencing on *Mlp*^{-/-} hearts (SRA accession ID: PRJNA327790). These data are from same animals of which whole-genome oxidative bisulfite (OxBS) and bisulfite (BS) raw sequencing data was generated, which is publicly available under same accession ID on SRA.

We have two different expression data sets from the left myocardial tissues of *Mlp*^{-/-} mice and their littermate controls. The first data set (set 1) is from 7 male *Mlp*^{-/-} vs 6 WT, and from 6 female *Mlp*^{-/-} vs 6 WT females. The second data set (set 2) is from a public source (SRA accession ID: PRJNA327790), and it is from 3 *Mlp*^{-/-} vs 3 WT males. Both data sets are from 12 weeks old animals, and from the same murine strain. *Tet1* was upregulated in data set 1 in male *Mlp*^{-/-} (statistically insignificant), but it was downregulated in data set 2 (statistically significant). Opposite contradiction is also observed for *Tet2* expression. These differences are most likely originating from technical artifacts, due to the long length of these transcripts and the relatively small fold change difference in their expression between *Mlp*^{-/-} and WT, as it can be inferred from the color scale of (Supplementary fig. 7a). Therefore, all these data can collectively be best described as “there was **no significant difference** in either the level of mRNA encoding *Tet1-3* or its enzymatic activity”.

There could be a difference in the activities of individual Tet enzymes, and it could be biologically significant. Although it is an interesting concept, it is not within the focus of this work. Our interest is in the final product of the activity of all TET1-3, i.e., the levels of genomic 5hmC.

Measuring individual activity of Tet1,2 or 3 in the heart is technically very challenging. The problem is mainly the low sensitivity and accuracy of available methods. Current methods rely on using antibodies to recognize hydroxymethylated substrates. Using an antibody would subject these methods to many artifacts, especially when the biological samples have very low expression level of TET1-3 enzymes, such as in heart. We used a commercially available kit (P-308, Epigentek), with which we found that the observed activity level of all TET1-3 in the heart was close to the kit’s lower quantification limit, and therefore we used 4 technical replicates, and large sample size to obtain more

reliable data. However, we do not think that this kit would be sensitive enough if two of the three Tet enzymes were blocked with blocking proteins. We tried previously to develop more sensitive methods to measure TET1-3 activity, by using synthetic 5mC- and 5hmC-methylated fragments of DNA, combined with qPCR or mass spectrometry, instead of the antibodies, but without any success.

2.2. Minor

2.2.1. The authors should adhere to standard nomenclature convention for use of capitalization and italics for gene symbols. Instead, there are inconsistent uses of caps and italics for IDH2 throughout the manuscript (e.g., “IDH2”, “Idh2” and “Idh2”).

Authors response: Now we revised the manuscript and we tried to correct previously utilized nomenclature. We used *IDH2* or IDH2 to refer to RNA and protein respectively when the data was obtained from human samples, or when we discussed a general conclusion. Whereas we used *Idh2* or Idh2 to refer to RNA and protein when the data was obtained from murine or rat samples.

2.2.2. For Fig.S1H, mitochondrial respiratory control ratio (RCR = state 3/state 4) is a more widely used readout to reflect mitochondrial function than the absolute respiratory rate.

Authors response: We added the graphs for RCR in the updated manuscript (**Supplementary fig. 1h**).

2.2.3. Most of the experiments focused on the IDH2 expression and related mechanism. The title is too general and does not reflect the main experiments and results of this manuscript.

Authors response: The manuscript does not focus on IDH2 only, but also on redox associated sex differences and antioxidative capacity in general. Reflecting all these concepts in the title will make it too long, and therefore we preferred a shorter title and explaining the different concepts in the abstract.

2.2.4. Please add genome coordinates for Fig.6F.

Authors response: We added the genomic coordinates to updated (**fig. 7d**).

3. Reviewer #3 (Remarks to the Author):

This manuscript from ElBeck et al. examines a role for the epigenetic regulation of IDH2 expression in the regulation of redox homeostasis in the heart. The authors initially note that IDH2 is downregulated in humans and mouse models of eccentric hypertrophy, but also follow an oxidative stress challenge with H₂O₂ in NRVMs. Interestingly, the authors find that IDH2 downregulation is associated with an increase in activity, which appears to serve antioxidant functions. Furthermore, the authors find an association between epigenetic marks and IDH2 expression, suggestion the potential for epigenetic control of redox homeostasis. This study is important and of interest to the field, but a number of issues need to be addressed before this manuscript is suitable for publication.

3.1.1. The authors stress a role for IDH2 in antioxidant defense. However, the authors do not actually examine oxidative stress. The authors should consider examining ROS production and/or markers of oxidative stress in further support of an antioxidant role for IDH2.

Authors response: We thank the reviewer for the careful consideration of our work and their suggestions. Indeed, direct measurements of redox species would be essential to support most conclusions inferred from transcriptomic profiling. We have now provided direct measurement for the levels of redox species in the updated manuscript (Please see section I on page 5 of this letter). We also restructured the manuscript to integrate these new data in the updated manuscript. We think that these new data helped to explain and strengthened many of our previous conclusions.

3.1.2. One-carbon metabolism is critical for the generation of methyl donors for epigenetic regulation and for the generation of certain antioxidants. Do the authors examine one-carbon metabolism to see if this pathway is altered and/or impaired?

Authors response: Indeed, one-carbon metabolism represents an essential pathway in regulating epigenetics as a donor for methyl residues (–CH₃), and in redox homeostasis as many intermediates in the folate metabolism are involved in redox reactions, and in NADH and GSH syntheses. In the heart, we examined the expression of key enzymes that regulate one carbon metabolism in both human cardiac DCM samples and in *Mip*^{-/-} hearts (see table below). By comparing the average number of normalized reads of target transcript with the that of *IDH2*, which is abundant in cardiomyocytes, or *GAPDH* that is abundant in all cellular types in the heart, we found that the expression of these enzymes was either low or show small differences in the expression between DCM and control samples. This would imply that either one-carbon metabolism has low basal

expression in all cell types in the heart, or it is only expressed in a subset of relatively sparse cell type, such as immune cells or vascular cells. However, the expression on the mRNA level is not necessarily reflected on the protein level, and therefore additional studies are required to determine corresponding protein levels, enzymes' activities, and even metabolites and intermediate metabolites involved in folate metabolism. Moreover, most of these key genes appeared to be dysregulated and highly statistically significant in both data sets (see table below), which indicates that one-carbon metabolism might be altered in cardiovascular diseases associated with DCM. However, although it is very interesting, it is not within the focus of the current manuscript.

Gene	Mlp ^{-/-}			Human DCM		
	Average reads	Log2 Fold Change	p value	Average reads	Log2 Fold Change	p value
SHMT1	36.07	-0.32	0.011	214.07	-0.17	0.219
PSPH	26.12	0.33	0.018	142.96	0.30	0.010
PSAT1	20.95	0.60	0.001	20.39	-2.25	<0.001
PHGDH	4.43	0.77	0.041	290.22	-0.99	<0.001
MTHFD1	185.43	-0.13	0.052	2751.32	-0.002	0.976
MTHFD2	23.88	1.34	<0.001	485.12	-0.70	<0.001
MTHFD2L	38.38	-0.27	0.017	210.48	0.05	0.600
SHMT2	156.79	-0.26	<0.001	1264.19	-0.53	<0.001
GLDC	1.91	-1.01	0.082	1.01	0.02	0.973
AMT	20.22	0.44	0.007	457.83	0.28	0.038
GCSH	276.35	0.06	0.605	15.70	-0.56	0.017
MTHFR	74.47	0.03	0.748	1260.52	-0.06	0.585
CBS	1.42	-1.21	0.057	10.77	0.43	0.282
IDH2	6987.14	-0.62	<0.001	28020.71	-0.44	0.001
GAPDH	8596.13	0.07	0.068	41457.54	-0.24	0.021

Table legend: Expression of key enzymes regulating one carbon metabolism in cytoplasm and nucleus⁴⁰, estimated by RNA sequencing on LV myocardium of male and female *Mlp*^{-/-} mice (n=12 for WT and 13 for *Mlp*^{-/-}), and from publicly available transcriptomic data from 65 cardiac tissue from patients with idiopathic dilated cardiomyopathy (DCM)¹⁰. *IDH2* and *GAPDH* were included to compare expression level as illustrated in the text above.

References:

- 1 Zhao, B., Summers, F. A. & Mason, R. P. Photooxidation of Amplex Red to resorufin: implications of exposing the Amplex Red assay to light. *Free radical biology & medicine* **53**, 1080-1087, (2012).
- 2 Votyakova, T. V. & Reynolds, I. J. Detection of hydrogen peroxide with Amplex Red: interference by NADH and reduced glutathione auto-oxidation. *Arch Biochem Biophys* **431**, 138-144, (2004).
- 3 De Sandro, V., Dupuy, C., Kaniewski, J., Ohayon, R., Deme, D., Virion, A. & Pommier, J. Mechanism of NADPH oxidation catalyzed by horse-radish peroxidase and 2,4-diacetyl-[2H]heme-substituted horse-radish peroxidase. *Eur J Biochem* **201**, 507-513, (1991).
- 4 Rahman, I., Kode, A. & Biswas, S. K. Assay for quantitative determination of glutathione and glutathione disulfide levels using enzymatic recycling method. *Nature Protocols* **1**, 3159-3165, (2006).
- 5 Mendonca, R., Gning, O., Di Cesare, C., Lachat, L., Bennett, N. C., Helfenstein, F. & Glauser, G. Sensitive and selective quantification of free and total malondialdehyde in plasma using UHPLC-HRMS. *J Lipid Res* **58**, 1924-1931, (2017).
- 6 Eijgenraam, T. R., Boogerd, C. J., Stege, N. M., Oliveira Nunes Teixeira, V., Dokter, M. M., Schmidt, L. E., Yin, X., Theofilatos, K., Mayr, M., van der Meer, P., van Rooij, E., van der Velden, J., Sillje, H. H. W. & de Boer, R. A. Protein Aggregation Is an Early Manifestation of Phospholamban p.(Arg14del)-Related Cardiomyopathy: Development of PLN-R14del-Related Cardiomyopathy. *Circulation Heart Failure* **14**, e008532, (2021).
- 7 Xi, Y., Chen, D., Dong, Z., Lam, H., He, J., Du, K., Chen, C., Guo, J. & Xiao, J. RNA Sequencing of Cardiac in a Rat Model Uncovers Potential Target LncRNA of Diabetic Cardiomyopathy. *Frontiers in genetics* **13**, 848364, (2022).
- 8 Sielemann, K., Elbeck, Z., Gartner, A., Brodehl, A., Stanasiuk, C., Fox, H., Paluszkiwicz, L., Tiesmeier, J., Wlost, S., Gummert, J., Albaum, S. P., Sielemann, J., Knoll, R. & Milting, H. Distinct Myocardial Transcriptomic Profiles of Cardiomyopathies Stratified by the Mutant Genes. *Genes (Basel)* **11**, (2020).
- 9 Sweet, M. E., Cocciolo, A., Slavov, D., Jones, K. L., Sweet, J. R., Graw, S. L., Reece, T. B., Ambardekar, A. V., Bristow, M. R., Mestroni, L. & Taylor, M. R. G. Transcriptome analysis of human heart failure reveals dysregulated cell adhesion in dilated cardiomyopathy and activated immune pathways in ischemic heart failure. *BMC genomics* **19**, 812, (2018).
- 10 van Heesch, S., Witte, F., Schneider-Lunitz, V., Schulz, J. F., Adami, E., Faber, A. B., Kirchner, M., Maatz, H., Blachut, S., Sandmann, C. L., Kanda, M., Worth, C. L., Schafer, S., Calviello, L., Merriott, R., Patone, G., Hummel, O., Wyler, E., Obermayer, B., Mucke, M. B., Lindberg, E. L., Trnka, F., Memczak, S., Schilling, M., Felkin, L. E., Barton, P. J. R., Quaipe, N. M., Vanezis, K., Diecke, S., Mukai, M., Mah, N., Oh, S. J., Kurtz, A., Schramm, C., Schwinge, D., Sebode, M., Harakalova, M., Asselbergs, F. W., Vink, A., de Weger, R. A., Viswanathan, S., Widjaja, A. A., Gartner-Rommel, A., Milting, H., Dos Remedios, C., Knosalla, C., Mertins, P., Landthaler, M., Vingron, M., Linke, W. A., Seidman, J. G., Seidman, C. E., Rajewsky, N., Ohler, U., Cook, S. A. & Hubner, N. The Translational Landscape of the Human Heart. *Cell* **178**, 242-260 e229, (2019).
- 11 Vigil-Garcia, M., Demkes, C. J., Eding, J. E. C., Versteeg, D., de Ruyter, H., Perini, I., Kooijman, L., Gladka, M. M., Asselbergs, F. W., Vink, A., Harakalova, M., Bossu, A., van Veen, T. A. B., Boogerd, C. J. & van Rooij, E. Gene expression profiling of hypertrophic cardiomyocytes identifies new players in pathological remodelling. *Cardiovascular research* **117**, 1532-1545, (2021).
- 12 Zhou, L., Wang, F., Sun, R., Chen, X., Zhang, M., Xu, Q., Wang, Y., Wang, S., Xiong, Y., Guan, K. L., Yang, P., Yu, H. & Ye, D. SIRT5 promotes IDH2 desuccinylation and G6PD deglutarylation to enhance cellular antioxidant defense. *EMBO reports* **17**, 811-822, (2016).
- 13 Kil, I. S. & Park, J. W. Regulation of mitochondrial NADP⁺-dependent isocitrate dehydrogenase activity by glutathionylation. *The Journal of biological chemistry* **280**, 10846-10854, (2005).
- 14 Yu, W., Dittenhafer-Reed, K. E. & Denu, J. M. SIRT3 protein deacetylates isocitrate dehydrogenase 2 (IDH2) and regulates mitochondrial redox status. *Journal of Biological Chemistry* **287**, 14078-14086, (2012).

- 15 Oldham, W. M., Clish, C. B., Yang, Y. & Loscalzo, J. Hypoxia-Mediated Increases in L-2-hydroxyglutarate Coordinate the Metabolic Response to Reductive Stress. *Cell metabolism* **22**, 291-303, (2015).
- 16 Wagner, M., Bertero, E., Nickel, A., Kohlhaas, M., Gibson, G. E., Heggermont, W., Heymans, S. & Maack, C. Selective NADH communication from alpha-ketoglutarate dehydrogenase to mitochondrial transhydrogenase prevents reactive oxygen species formation under reducing conditions in the heart. *Basic Research in Cardiology* **115**, 53, (2020).
- 17 Nickel, A. G., von Hardenberg, A., Hohl, M., Loffler, J. R., Kohlhaas, M., Becker, J., Reil, J. C., Kazakov, A., Bonnekoh, J., Stadelmaier, M., Puhl, S. L., Wagner, M., Bogeski, I., Cortassa, S., Kappl, R., Pasiaka, B., Lafontaine, M., Lancaster, C. R., Blacker, T. S., Hall, A. R., Duchen, M. R., Kastner, L., Lipp, P., Zeller, T., Muller, C., Knopp, A., Laufs, U., Bohm, M., Hoth, M. & Maack, C. Reversal of Mitochondrial Transhydrogenase Causes Oxidative Stress in Heart Failure. *Cell metabolism* **22**, 472-484, (2015).
- 18 Wise, D. R., Ward, P. S., Shay, J. E., Cross, J. R., Gruber, J. J., Sachdeva, U. M., Platt, J. M., DeMatteo, R. G., Simon, M. C. & Thompson, C. B. Hypoxia promotes isocitrate dehydrogenase-dependent carboxylation of α -ketoglutarate to citrate to support cell growth and viability. *Proceedings of the National Academy of Sciences* **108**, 19611-19616, (2011).
- 19 Intlekofer, A. M., Wang, B., Liu, H., Shah, H., Carmona-Fontaine, C., Rustenburg, A. S., Salah, S., Gunner, M. R., Chodera, J. D., Cross, J. R. & Thompson, C. B. L-2-Hydroxyglutarate production arises from noncanonical enzyme function at acidic pH. *Nature chemical biology* **13**, 494-500, (2017).
- 20 Harvey, C. J., Thimmulappa, R. K., Singh, A., Blake, D. J., Ling, G., Wakabayashi, N., Fujii, J., Myers, A. & Biswal, S. Nrf2-regulated glutathione recycling independent of biosynthesis is critical for cell survival during oxidative stress. *Free radical biology & medicine* **46**, 443-453, (2009).
- 21 Thimmulappa, R. K., Mai, K. H., Srisuma, S., Kensler, T. W., Yamamoto, M. & Biswal, S. Identification of Nrf2-regulated Genes Induced by the Chemopreventive Agent Sulforaphane by Oligonucleotide Microarray. **62**, 5196-5203, (2002).
- 22 Dinkova-Kostova, A. T. & Abramov, A. Y. The emerging role of Nrf2 in mitochondrial function. *Free radical biology & medicine* **88**, 179-188, (2015).
- 23 Lau, A., Tian, W., Whitman, S. A. & Zhang, D. D. The predicted molecular weight of Nrf2: it is what it is not. *Antioxidants & redox signaling* **18**, 91-93, (2013).
- 24 Li, W. Q., Dehnade, F. & Zafarullah, M. Thiol antioxidant, N-acetylcysteine, activates extracellular signal-regulated kinase signaling pathway in articular chondrocytes. *Biochemical and Biophysical Research Communications* **275**, 789-794, (2000).
- 25 Yan, C. Y. & Greene, L. A. Prevention of PC12 cell death by N-acetylcysteine requires activation of the Ras pathway. *Journal of Neuroscience* **18**, 4042-4049, (1998).
- 26 Kehat, I., Davis, J., Tiburcy, M., Accornero, F., Saba-El-Leil, M. K., Maillet, M., York, A. J., Lorenz, J. N., Zimmermann, W. H., Meloche, S. & Molkenstin, J. D. Extracellular signal-regulated kinases 1 and 2 regulate the balance between eccentric and concentric cardiac growth. *Circulation research* **108**, 176-183, (2011).
- 27 Bartholomeus, J., Bürli, R., Jarvis, R., Johnstone, S., Ostefeld, T., Terstiege, I., Travagli, M. & Turcotte, S. Small molecule modulators of the BTB domain of Keap1. *U.S. Patent and Trademark Office. U.S. Patent No. 11,479,539*, (2022).
- 28 Booth, M. J., Ost, T. W., Beraldi, D., Bell, N. M., Branco, M. R., Reik, W. & Balasubramanian, S. Oxidative bisulfite sequencing of 5-methylcytosine and 5-hydroxymethylcytosine. *Nature Protocols* **8**, 1841-1851, (2013).
- 29 Globisch, D., Munzel, M., Muller, M., Michalakakis, S., Wagner, M., Koch, S., Bruckl, T., Biel, M. & Carell, T. Tissue distribution of 5-hydroxymethylcytosine and search for active demethylation intermediates. *PloS one* **5**, e15367, (2010).
- 30 Liu, B., el Alaoui-Talibi, Z., Clanachan, A. S., Schulz, R. & Lopaschuk, G. D. Uncoupling of contractile function from mitochondrial TCA cycle activity and MVO₂ during reperfusion of ischemic hearts. *Am J Physiol* **270**, H72-80, (1996).
- 31 Davis, J., Davis, L. C., Correll, R. N., Makarewich, C. A., Schwanekamp, J. A., Moussavi-Harami, F., Wang, D., York, A. J., Wu, H., Houser, S. R., Seidman, C. E., Seidman, J. G., Regnier, M., Metzger, J. M.,

- Wu, J. C. & Molkenin, J. D. A Tension-Based Model Distinguishes Hypertrophic versus Dilated Cardiomyopathy. *Cell* **165**, 1147-1159, (2016).
- 32 Janssen, M., Koster, J. F., Bos, E. & de Jong, J. W. Malondialdehyde and glutathione production in isolated perfused human and rat hearts. *Circulation research* **73**, 681-688, (1993).
- 33 Meister, A. Glutathione metabolism and its selective modification. *Journal of Biological Chemistry* **263**, 17205-17208, (1988).
- 34 Thimmulappa, R. K., Mai, K. H., Srisuma, S., Kensler, T. W., Yamamoto, M. & Biswal, S. Identification of Nrf2-regulated Genes Induced by the Chemopreventive Agent Sulforaphane by Oligonucleotide Microarray. *Cancer Research* **62**, 5196-5203, (2002).
- 35 Dinkova-Kostova, A. T., Holtzclaw, W. D., Cole, R. N., Itoh, K., Wakabayashi, N., Katoh, Y., Yamamoto, M. & Talalay, P. Direct evidence that sulfhydryl groups of Keap1 are the sensors regulating induction of phase 2 enzymes that protect against carcinogens and oxidants. *Proceedings of the National Academy of Sciences of the United States of America* **99**, 11908-11913, (2002).
- 36 Ponnaluri, V. K., Ehrlich, K. C., Zhang, G., Lacey, M., Johnston, D., Pradhan, S. & Ehrlich, M. Association of 5-hydroxymethylation and 5-methylation of DNA cytosine with tissue-specific gene expression. *Epigenetics* **12**, 123-138, (2017).
- 37 Spruijt, C. G., Gnerlich, F., Smits, A. H., Pfaffeneder, T., Jansen, P. W., Bauer, C., Munzel, M., Wagner, M., Muller, M., Khan, F., Eberl, H. C., Mensinga, A., Brinkman, A. B., Lephikov, K., Muller, U., Walter, J., Boelens, R., van Ingen, H., Leonhardt, H., Carell, T. & Vermeulen, M. Dynamic readers for 5-(hydroxy)methylcytosine and its oxidized derivatives. *Cell* **152**, 1146-1159, (2013).
- 38 Yang, X., Han, H., De Carvalho, D. D., Lay, F. D., Jones, P. A. & Liang, G. Gene body methylation can alter gene expression and is a therapeutic target in cancer. *Cancer cell* **26**, 577-590, (2014).
- 39 Angeloni, A. & Bogdanovic, O. Enhancer DNA methylation: implications for gene regulation. *Essays Biochem* **63**, 707-715, (2019).
- 40 Shuvalov, O., Petukhov, A., Daks, A., Fedorova, O., Vasileva, E. & Barlev, N. A. One-carbon metabolism and nucleotide biosynthesis as attractive targets for anticancer therapy. *Oncotarget* **8**, 23955-23977, (2017).

REVIEWER COMMENTS

Reviewer #1 (Remarks to the Author):

The authors have addressed most of the concerns from the first submission.

Reviewer #2 (Remarks to the Author):

Re: Revision of “Epigenetic modulators link mitochondrial redox homeostasis to cardiac function”.

The authors have revised the manuscript on redox homeostasis and cardiac hypertrophy and failure. Although the revised manuscript was accompanied by an extensive rebuttal letter, in many cases I did not find the revision responsive to the initial comments. The manuscript continued to be disjointed, hard to follow, and not adequately supported by data. Small changes, in some cases changes that are not statistically significant, are combined to support the proposed model.

1. The authors support their claims using differences that are not statistically significant. They indicate in the rebuttal that statistics cannot be relied on due to small sample sizes. While small sample sizes are problematic, ignoring statistics comes at the cost of rigor and reproducibility.
2. The claim that male *Mlp*^{-/-} hearts have higher oxidative stress than females remains poorly supported.
 - Female *Mlp*^{-/-} mice had higher levels of GSH and GSSG than male counterparts (although not statistically significant). However, male *Mlp*^{-/-} and WT mice had no difference in GSH or GSSG. This suggests there is no significant elevation of oxidative stress in the *Mlp*^{-/-} genotype.
 - Often GSH/GSSG ratio is used as a marker of oxidative stress levels. What happens to this ratio in male vs female mice?
3. Oxidative stress levels in NRCMs was inferred by measurement of GSH/GSSG in human fibroblasts. This is a big assumption that oxidative stress levels in fibroblasts model those in NRCMs.
4. The authors should provide data on NRF2 activity in mice treated with AZ925. While they authors cite a manuscript supporting the activity of this small molecule, it is necessary to show in that in this system it is activating NRF2.
5. In Fig 7 it seems better use can be made of the NRCMs and the groundwork laid in Fig 3 to establish the cause-effect regulatory relationships between IDH2, 2OG, L2HG, and the 5hmC DMR at *Idh2*. For

example, the manipulations in panel e could be coupled to analysis of 5hmC at the Idh2 DMR, and the transcriptional enhancer activity of this DMR could be probed.

Minor comments

1. Line 186: ...TAB mice ... which are models of hypertrophic cardiomyopathy. TAB mice are models of concentric hypertrophy but not hypertrophic cardiomyopathy.
2. Line 189: IDH2 is primarily responsible for the regeneration of the mitochondrial NADPH...” This appears to discount IDH2’s role in the TCA cycle?
3. Line 289 typo: cardiomyocyte’s death

Reviewer #3 (Remarks to the Author):

The authors have sufficiently addressed my concerns.

Response to reviewers' comments regarding the revised manuscript NCOMMS-22-31381/ NCOMMS-22-31381A/ NCOMMS-22-31381B (Elbeck et al 2023)

Epigenetic modulators link mitochondrial redox homeostasis to cardiac function

Reviewers' comments are highlighted in **black and bold font**, whereas authors' response is colored blue and in normal font.

We thank the reviewers for their careful consideration of our work, and all suggestions to improve it. We have now revised the manuscript again, which substantially improved it from the previous version. We highlight below the changes that we made to the revised figures, as well as we provide an auxiliary file, in which all changes were tracked:

Figure 4:

- New data for the intracellular concentrations of GSH and GSSG in NRCMs and hiPS-CMs were added in figure 4d, e.
- New data for the intracellular concentrations of MDA in NRCMs exposed to increasing concentrations of SF were added in figure 4f, and previous data depicting MDA level in hFF1 cells were moved to supplementary fig. 4i.
- We removed Western blotting data for Osgin1 in the LV of *Mlp*^{-/-} and their subsequent bands quantifications from figure 4g, as we became less confident that the observed bands were specific for Osgin1 (please see the appendix at the end of this letter for more information and explanation).
- We added the GSH/GSSG ratio in the LV of male and female *Mlp*^{-/-} in figure 4j.

Supplementary Fig 4:

- Under supplementary figures 4c, d, h, we added bar plots for the intracellular concentrations of GSH and GSSG in NRCMs, hiPS-CMs and hFF1 cells, respectively. This data is also presented in main fig. 4d, e, and the idea of showing the bar plots in the supplementary materials was to indicate actual *p*-values for all statistical analyses between different groups, as we refrained from using star-based indicators. We also aimed to visualize sample replicates.

Supplementary Fig 8

Supplementary Fig 8b: We added new data for Western blotting of Nrf2, Hmox1 and Gapdh in LV from two mice treated with AZ925 or scramble control and then dissected 9 hours post administration.

Response to individual reviewers' comments

Reviewer #1 (Remarks to the Author):

The authors have addressed most of the concerns from the first submission.

Authors' response: We thank the reviewer for all the feedback and all constructive suggestions, which helped us to substantially improve the quality of our work.

Reviewer #2 (Remarks to the Author):

Re: Revision of "Epigenetic modulators link mitochondrial redox homeostasis to cardiac function".

The authors have revised the manuscript on redox homeostasis and cardiac hypertrophy and failure. Although the revised manuscript was accompanied by an extensive rebuttal letter, in many cases I did not find the revision responsive to the initial comments. The manuscript continued to be disjointed, hard to follow, and not adequately supported by data. Small changes, in some cases changes that are not statistically significant, are combined to support the proposed model.

Authors' response: We thank the reviewer for further comments and suggestions. Here, we wish to state that we did our very best to respond comprehensively to the previous round of comments and provided extensive additional new data.

We provide below detailed responses to the new raised concerns, supporting them with new data.

1. The authors support their claims using differences that are not statistically significant. They indicate in the rebuttal that statistics cannot be relied on due to small sample sizes. While small sample sizes are problematic, ignoring statistics comes at the cost of rigor and reproducibility.

Author response: We completely agree with the Reviewer about the importance of statistics to support rigor and reproducibility of the data. As far as we understand, the Reviewer is referring only to figure 7a, which deals with epigenetics data, because all other conclusions in the manuscript are based on the result from statistical analysis. However, epigenetic sequencing is currently technically very challenging and costly, and this has consequences for the possibility

to perform appropriate statistical analysis on such data. To the best of our knowledge, the hydroxy methylation data presented in our manuscript represents the highest resolution currently available from cardiac tissues, or from any other tissues. For these reasons, we think that the observed trend is both interesting and important to report to the scientific community. However, we agree with the reviewer that the observed differences in 5mC or 5hmC levels between *Mlp*^{-/-} and WT data should be presented as such (a trend, albeit statistically not significant). We did this in the previous version, but have clarified the message in the revised manuscript Results as follows:

Lines 463-466: *“Our analysis indicated that the average methylation percentages of the total CpGs in the murine genome were 58.7% for 5mC and 4.4% for 5hmC. In Mlp^{-/-} hearts, there were trends, albeit statistically insignificant, toward increased levels of 5mC and fewer 5hmC, (Fig. 7a).*

We also explained this limitation in the Discussion.

Lines 582-488: *“Our data regarding 5hmC remain correlative due to current technical challenges in available technologies; this represents a limitation to our approach. We demonstrate here that during cardiac remodeling, 5hmC in specific regions of genes is a potent regulatory of gene expression. Unfortunately, because of its extremely low levels in the genome, elucidating this regulatory role of 5hmC in greater detail would require high sequencing coverage (> 200X) with relatively large sample size, which is technically challenging at present. Hopefully, emerging third-generation sequencing and other future advances will facilitate such investigations in the future.”*

2. The claim that male *Mlp*^{-/-} hearts have higher oxidative stress than females remains poorly supported. Female *Mlp*^{-/-} mice had higher levels of GSH and GSSG than male counterparts (although not statistically significant). However, male *Mlp*^{-/-} and WT mice had no difference in GSH or GSSG. This suggests there is no significant elevation of oxidative stress in the *Mlp*^{-/-} genotype. Often GSH/GSSG ratio is used as a marker of oxidative stress levels. What happens to this ratio in male vs female mice?

Authors' response: We thank the Reviewer for their comment, but we kindly disagree with their opinion. We have provided several direct lines of evidence for oxidative stress in both male and female *Mlp*^{-/-} hearts, as well as for the redox associated sex dimorphism. This evidence

is not only inferred from measuring GSH/GSSG levels, but also from the expression of antioxidative genes, echocardiography and treatment with AZ925. Below we summarize the main evidence:

- Both male and female *Mlp*^{-/-} hearts show an increase in *Hmox1* and *Nqo1* expression, which are two marker-genes for acute and chronic exposure to oxidative stress, respectively (**Fig. 4g**). Moreover, the increase in *Hmox1* was more noticeable in male mice ($p=0.026$) comparing to females, a difference that was even more pronounced in the mRNA levels ($p=0.017$), where *Hmox1* mRNA was upregulated only in males (**Fig. 4h**). We also show that the expression level of *Hmox1* is negatively correlated with cardiac ejection fraction ($p=0.001$), with males having lower ejection fraction (**Fig. 5a**).
- Both male and female *Mlp*^{-/-} hearts show a trend (albeit statistically not significant) elevation in GSH levels, which however is reflected in a highly statistically significant reduction of MDA levels in females ($p<0.001$) (**Fig. 4k**).
- GSH/GSSG ratio, which is considered a marker for oxidative stress, is highly elevated in males vs. females *Mlp*^{-/-} ($p=0.001$) (**Fig. 4j**, also shown below).

Figure 4

- The higher oxidative stress observed in male *Mlp*^{-/-} mice made their response to AZ925 treatment more effective, resulting in a significant increase in GSH levels ($p=0.003$) (**Fig. 6h**). In contrast, female *Mlp*^{-/-} mice, exhibiting comparatively lower basal oxidative stress, demonstrated a milder elevation in GSH levels upon the treatment with AZ925 (**Fig. 6h**). This discrepancy could be attributed to the reduced demand for oxidant neutralization compared to males. In this regard, we found that treating NRCMs with the antioxidant NAC led to elevated GSH levels, but raising the concentration of NAC did not further enhance GSH levels (Please see the figure below, these data are not included in the manuscript).

In summary, we have provided multiple evidence for the presence of oxidative stress in both male and female $Mip^{-/-}$, as well as for redox associated sex dimorphism. The majority of the evidence was based on statistically significant data. We also wish to emphasize that statistical tests are not a filter to eliminate data, but rather a tool employed to aid interpretation of the data (e.g., it is not appropriate to consider a dataset with $p=0.05$ while discarding one with $p=0.058$). To facilitate transparent assessment by readers, we have presented actual p -values in all our figures and refrained from using star-based indicators.

3. Oxidative stress levels in NRCMs was inferred by measurement of GSH/GSSG in human fibroblasts. This is a big assumption that oxidative stress levels in fibroblasts model those in NRCMs.

Authors' response: We thank the reviewer for highlighting this issue. We agree with the Reviewer that it is problematic to infer oxidative stress in cardiomyocytes from data in fibroblasts. Therefore, we now provide such data from NRCMs, as well as from human cardiomyocytes derived from induced pluripotent stem cells (hiPSC-CMs). We found that both NRCMs and iPSC-CMs show similar trends of regulation for GSH and GSSG to those previously observed in hFF1C line (**Fig. 4d, e**).

Figure 4

Notably, NRCMs display a comparable trend to hiPSC-CMs but with lower GSH and higher GSSG basal levels. These differences could be attributed to the metabolic nature of cardiomyocytes, their relative differentiation status, and purity in both cultures, as well as to variations in their derivation methods, species, and culturing conditions.

Furthermore, both hiPSC-CMs and NRCMs exhibit a biphasic response to SF, with an initial increase up to 2-5 μ M SF and a subsequent reduction in GSSG levels at higher SF concentrations. The turning point of this biphasic behaviour occurs at lower SF concentrations in NRCMs than in hiPSC-CMs, which may be also attributed to the factors described above. This observed decline in GSSG levels is probably due to the increased formation of GSH adducts with other molecules at high concentrations of SF. Most available direct methods for metabolomic quantification, including our developed method, assess only free GSH and GSSG, as proteins carrying GSH or GSSG conjugations precipitate in the 80% aqueous methanol solvent, necessitating a targeted proteomic approach for their quantification.

Nonetheless, our focus in this study centres on the substantial increase in GSH level with low concentrations of SF, mirroring the observed elevation in *Mlp*^{-/-} hearts and indicating a reductive microenvironment. This increase appears consistent across all three cell types.

4. The authors should provide data on NRF2 activity in mice treated with AZ925. While they authors cite a manuscript supporting the activity of this small molecule, it is necessary to show in that in this system it is activating NRF2.

Authors' response: We have provided multiple biochemical evidence that AZ925 induces the expression of NRF2 target genes. These data are presented in main figure 6 and in supplementary figure 8, as well as in our first Response to Referees letter (Section 2.1.8)

Here below we summarize mentioned evidence, and we also add additional biochemical data supporting the activation of NRF2 by AZ925:

- According to unpublished pharmacokinetic observations during development of AZ925, a peak in the expression of the downstream targets of NRF2 is observed in the brain 9 hours post oral administration. Therefore, we validated that AZ925 activates NRF2 in the heart in a pilot study, in which we treated two mice with AZ925 or scramble control and then dissected their hearts, brains, kidneys and livers 9 hours post administration for biochemical assays. We showed by qPCR that AZ925 induced the expression of

selected experimentally validated downstream targets of NRF2 (*Hmox1*, *Nqo1* and *Osgin1*) in all investigated organs, including the heart, which represents an immediate molecular effect of activating NRF2 (Supplementary figure 8a, also shown below). We now also show an increase in the protein level of NRF2 in the myocardium of mice treated with AZ925 and the corresponding increase in Hmox1 protein level in these hearts (Supplementary figure 8b, also shown below).

Supplementary fig. 8

The increase in the protein level of NRF2 upon AZ925 treatment represents another immediate effect of activating NRF2, as under basal conditions, NRF2 is kept inactive with a very short turnover half-life (few mins) due to its continuous proteasomal degradation by Keap1- Cul3 complex¹. The dissociation of NRF2 from KEAP1 stabilizes it and thus increases its protein level and leads to its subsequent nuclear translocation to activate the transcription of downstream antioxidative genes.

We also wish to stress here that in our previous Response to Referees letter (Section 1.4.1), we comprehensively discussed the problem related to the non-specificity of commercially available antibodies for NRF2. We first observed this issue when blotting for activated NRF2 in hFF1 cells upon treatment with sulforaphane. The exact same issue can be also clearly observed in the Western blotting shown above (Supplementary figure 8b). This issue, which was nicely highlighted in a letter published by Lau et al in

2013², makes utilizing further reporter antibody-based assays to show NRF2 transcriptional activation highly inaccurate.

- **Fig. 6G** displays the induction/repression of more than 90 experimentally validated direct downstream targets of Nrf2 in *Mlp^{-/-}* treated with AZ925 with a daily gavage for a period of 30 days. The differential expression of these genes, plotted by the filled green and red circles, is not a bioinformatic prediction, but rather a direct measurement of differential expression in treated hearts by RNA sequencing. Indicated *p*-value for Nrf2 activation is based on the Right-Tailed Fisher's Exact Test, and it reflects the likelihood that the association between the collective expression of all NRF2 downstream target genes in the investigated samples, and the inferred expression state of these genes in published literature when NRF2 is activated could happen due to a random chance. The *p*-value in this case is extremely small (1.73×10^{-27}), indicating that the common upstream regulator of all these genes, i.e., NRF2 is activated upon the treatment with AZ925.

Figure 6

In summary, we provided multiple lines of evidence for AZ925 induced expression of NRF2 downstream of antioxidant genes, which represents our main interest for this work. The exact mechanism of action of AZ925 is not within the scope of this study, and we instead reference the patent application of AZ925³.

5. In Fig 7 it seems better use can be made of the NRCMs and the groundwork laid in Fig 3 to establish the cause-effect regulatory relationships between IDH2, 2OG, L2HG, and the 5hmC DMR at Idh2. For example, the manipulations in panel e could be coupled to analysis of 5hmC at the Idh2 DMR, and the transcriptional enhancer activity of this DMR could be probed.

Authors' response: We do agree with the reviewer that further investigation is needed to unravel detailed mechanism of the regulatory relationships between IDH2, 2OG, L2HG, and the 5hmC DMR at Idh2 genomic locus. However, we explained in our previous Response to Referees letter (Section 2.1.9) current limitations for achieving such study. We also explained that our data regarding 5hmC remains correlative due to the current technical limitations in available technologies. Furthermore, identifying transcriptional enhancer activity is not within the main scope of our current study, and it would require a major study just to plot all different histone modifications, the conclusions of which would be limited by the (limited) resolution of available 5hmC data, as explained in section (2.1.9) in our previous Response to Referees letter. However, we agree with reviewer that it is an important issue, and we therefore highlight it as a limitation to our study, as we also indicated above in our response to comment 1.

Minor comments

1. Line 186: ...TAB mice ... which are models of hypertrophic cardiomyopathy. TAB mice are models of concentric hypertrophy but not hypertrophic cardiomyopathy.

Authors' response: We corrected it in the text.

2. Line 189: IDH2 is primarily responsible for the regeneration of the mitochondrial NADPH..." This appears to discount IDH2's role in the TCA cycle?

Authors' response: We reformulated it and corrected the phrasing. We meant that IDH2 represent the primary source of mitochondrial NADPH in cardiomyocytes. This doesn't discount IDH2's role in the TCA cycle. We thank the review for highlighting it.

3. Line 289 typo: cardiomyocyte's death

Authors' response: We corrected it in the text.

Reviewer #3 (Remarks to the Author):

The authors have sufficiently addressed my concerns.

Authors' response: We thank the Reviewer for all the feedback and all constructive suggestions, which helped us to substantially improve the quality of our work.

Appendix:

We decided to exclude the Western blotting data for Osgin1 in *Mlp*^{-/-} hearts from Figure 4. We made this decision after becoming less confident in the specificity of observed bands for Osgin1, even though observed molecular weight matched the one indicated in the specification sheet of the Osgin1 antibody (15248-1-AP, Proteintech). However, according to Proteintech, this antibody recognizes three bands for Osgin1 at 38, 52, 61 kDa. but neither 38 nor 61 kDa are listed as potential isoforms for murine Osgin1 in the UniProt protein database.

Furthermore, when blotting for Osgin1 in hearts of mice treated with a single dose of AZ925, the intensity of the 52 kDa band did not increase, despite a twofold increase at its mRNA level (**Supplementary figure 8a**). Moreover, when blotting for OSGIN1 in hFF1 cells treated with increasing concentrations of SF, a 52 KDa band appeared for Osgin1 only at high doses of SF and after long exposure of the blotted membrane for UV light when imaging the chemiluminescence signal, but no bands appeared in the basal untreated controls (**Supplementary figure 4e**).

In light of these uncertainties regarding the specificity of the observed bands for Osgin1 in murine heart samples, we judiciously opted to remove these data from figure 4. Importantly, these data are not pivotal to the drawn conclusions. It's noteworthy that the qPCR data for Osgin1 were retained, illustrating a lack of changes in gene expression between *Mlp*^{-/-} and WT, both in male and female hearts.

References

- 1 Suzuki, T. & Yamamoto, M. Molecular basis of the Keap1-Nrf2 system. *Free radical biology & medicine* **88**, 93-100, (2015).
- 2 Lau, A., Tian, W., Whitman, S. A. & Zhang, D. D. The predicted molecular weight of Nrf2: it is what it is not. *Antioxidants & redox signaling* **18**, 91-93, (2013).

- 3 Bartholomeus, J., Bürli, R., Jarvis, R., Johnstone, S., Ostenfeld, T., Terstiege, I., Travagli, M. & Turcotte, S. Small molecule modulators of the BTB domain of Keap1. *U.S. Patent and Trademark Office*. **U.S. Patent No. 11,479,539**, (2022).

REVIEWER COMMENTS

Reviewer #4 (Remarks to the Author):

In the current article, investigators test the hypothesis that IDH2 is a major regulator of antioxidant defences in the heart. To do so, they used a model of eccentric hypertrophy by knocking-out Mlp in mouse hearts which, induces eccentric hypertrophy comparably to human DCM. This model is poorly described in the text, even though it is extensively used in the study.

The article is not very easy to follow as the information discussed later becomes new information, which is not detailed in the introduction. The primary concern seem to be the marginal effect of the data presented, which may not necessarily achieve statistical significance.. Nonetheless, it is possible that there may be biological significance and correlative trends. In some cases, however, the number of replicates seems to differ from one graph to another even though the replicates are from the same experiment. This suggests that the number of replicates were “adjusted” to bias for statistical significance, as the effect size of the data may be marginal. Additionally, the authors have seemed to make prior substantive changes and additional experiments for the first revision.

Below are comments on the respective figures and text:

Comments

Figure 1g: Authors report Idh2 expression in Sham vs MI hearts. Using 2 animals for sham and 4 for MI is a bit too low to calculate statistical significance, and to draw conclusions.

Figure 1h: The reduction of IDH2 protein level is approximately only 10%. Is this enough to have a physiological effect? Also, it would be probably better to include the Western Blot in the main figure, together with the quantification for better clarity.

Figure 2a: The authors use neonatal rat cardiomyocytes to test how oxidative stress modulates IDH2.

The authors assess that Idh2 expression is modulated according to higher concentration of H₂O₂. Here, they do not consider that higher concentration of H₂O₂ could kill the cells. Idh2 reduction of expression could be simply associated to the reduction of cell number. The authors should perform a cell viability

assay to show the number of viable cells. Quantifying the amount of RNA is not enough as, CMs can divide the nuclear content without undergoing cytokinesis.

Line 197 to 201:the authors state:” as the level of H₂O₂ gradually declined, this expression returned almost to normal, particularly in the case of cells exposed to low concentrations”. At the same time, further incubation of these cells with fresh medium free from H₂O₂ for an additional 24 hours did not raise the level of Idh2 expression any further, indicating the potential presence of long-lasting effects”. Additionally, this data is not present in Figure 2A, and may be mislabelled.

Figure 2c and d: I presume that the animals used to measure IDH2 activity are the same. But in figure 2c they report n=7 WT and n=11 Mlp^{-/-}, while in figure 2d the numbers are n=7 WT and n=14 Mlp^{-/-}. Are those 2 independent experiments, with different animals used, or the number of Mlp^{-/-} animals has been modified between figure 2c and d? Were outliers removed for Figure 2C? In contrary,for figure 3a, the authorsmeasure L2HG and D2HG from the same number of WT and Mlp^{-/-} animals.

Figure 3f and g: The authors state:” Although elevated levels of D2HG have been reported to downregulate IDH2 in a variety of cell types, we observed that elevated levels of L2HG in NRCMs, but not D2HG, were associated with upregulation of both Idh2 mRNA and protein”. Here, the level of Idh2 upregulation is minimal and the increase of IDH2 protein is not significant so the above statement is not completely supported by the data they show.

Figure 4a: Idh2 is downregulated in neonatal rat cardiomyocytes accordingly to administration oxidative stress by Sulforaphane. This could be again caused by cell death as also stated by the authors in line 300.

Figure 4a: Total RNA is reduced at 24h in cells treated with NAC, which is not supposed to cause cell death. Even if there is no proliferation, one would expect the number of cells and RNA to be at least the same as at 6h and not reduced at 24 and 48h in control condition (NAC). Please clarify.

Figure 4c: The authors show that Idh2 expression is reduced according to increased concentration of Sulforaphane. The authors also declare ”Sulforaphane induced substantial cell death”. This assumption comes from the quantification of total RNA as shown in supplementary figure 4b. Here, also normal incubation with NAC induced a reduction of total RNA after 24 or 48h. If the increase in Sulforaphane concentration cause cell death, this could also imply that lower Idh2 expression could be due to a lower number of cells. This could also be the case at 5uM and 7.5uM Sulforaphane. A cells viability assay, like a TUNEL assay, could clarify this.

In supplementary figure 4e, the authors test the Caspase 3 in protein extracts at different concentration of Sulforaphane. Here, they don't observe any difference in Caspase 3, also at 10uM of Sulforaphane. This is in contrast with what stated and measured at total RNA level.

Figure 4g: The expression of HMOX1 and NQO1 are very similar in Mlp^{-/-} Male and female mice. Even though, Hmox1 is more expressed at mRNA level, HMOX1 protein is expressed in the similar way in both Mlp^{-/-} Male and female mice so the assumption that "the increase in Hmox1 was more noticeable in male mice compared to female ones, a difference that was even more pronounced on the mRNA levels, where Hmox1 mRNA was upregulated only in males". The quantification in Figure 4G suggest otherwise, how many replicates was this performed?

Figure 4j: The authors state that "male Mlp^{-/-} hearts had substantially higher ratio in comparison to female ones ". GSH/GSSG ratio seem to be at base line higher in WT mice in comparison to females.

Figure 5c and d: "at the younger age females exhibited a greater fraction of LVEF, a difference that became more pronounced with age as LVEF improved in female mice, but not in males". The authors should explain how Mlp^{-/-} female improve LVEF from 10 to 14 weeks without any treatment. In addition, some discrepancies are noticed in between panel c and d. In panel c, n=17 mice are used for male and females, while in panel d, n=11 mice are used. Are those 11 mice selected as a subset from the 17 of panel c? Using which criteria?

Figure 6b: "Consistent with our hypothesis, daily gavage with AZ925 for 30 days and monitoring with echocardiography on days 0, 14 and 30 revealed significant improvement of LVEF in male but not female Mlp^{-/-} mice". The significant improvement stated by the authors is very minimal and it is not clear why the number of mice at day 0 is n=6 for both Scr. And Nrf2 act., while it becomes n=5 for Nrf2 act. at Day 14 and Day 30. Additionally, the spread of data for females is about 40%, while for males about 20-25%. Have the authors checked for equal variance when relevant statistical tests are applied.

Figure 6k: "additional exogenous induction of antioxidative response resulted in further pronounced downregulation of Idh2 in treated female mice". This is another statement supported by a borderline statistically significant difference.

Minor comments:

1. Authors may want to consider moving line 191-193 to the next section "oxidative stress downregulates IDH2" as the background and topic sentence.

2. Figure 2 Legends: the word mean is missing in mean+- SEM.

3. Line 235-237: Please reword and reformat the sentence structure – “ Under hypoxic conditions, cellular levels of 2OF and lactate rise while the level of L0malate falls, causing lactate (LDH) and malate dehydrogenase (MDH) to reduce 2OG to L-2-hydroxyglutarate (L2HG).

Response to reviewers' comments regarding the revised manuscript NCOMMS-22-31381/ NCOMMS-22-31381A/ NCOMMS-22-31381B/ / NCOMMS-22-31381C (Elbeck et al 2023)

Epigenetic modulators link mitochondrial redox homeostasis to cardiac function

Reviewers' comments are highlighted in **black and bold font**, whereas authors' response is colored blue and in normal font.

We thank the reviewer for the careful consideration of our work, and all suggestions to improve it. We have now revised the manuscript again and clarified all concerns raised by the reviewer 4 below. We also provide an auxiliary manuscript file, in which all changes were tracked:

Reviewer #4 (Remarks to the Author):

In the current article, investigators test the hypothesis that IDH2 is a major regulator of antioxidant defences in the heart. To do so, they used a model of eccentric hypertrophy by knocking-out Mlp in mouse hearts which, induces eccentric hypertrophy comparably to human DCM. This model is poorly described in the text, even though it is extensively used in the study.

Authors' response: *Mlp*^{-/-} has been used as a mouse model for dilated cardiomyopathy in several studies before where it was extensively characterized, and we referred to these studies in the previous version of the manuscript. Nevertheless, we added information to give at least a brief description of the model in the text here as well (additions in bold font):

*"We subsequently analysed left ventricle (LV) samples from patients with dilated cardiomyopathy (DCM), as well as from **muscle LIM protein (Mlp)**-deficient (*Mlp*^{-/-} or *Csrp3*^{-/-}) mice, which develop an eccentric hypertrophy comparable to human DCM **due to alterations in the cytoarchitecture of cardiomyocytes**"*

The article is not very easy to follow as the information discussed later becomes new information, which is not detailed in the introduction. The primary concern seem to be the marginal effect of the data presented, which may not necessarily achieve statistical significance. Nonetheless, it is possible that there may be biological significance and correlative trends. In some cases, however, the number of replicates seems to differ from

one graph to another even though the replicates are from the same experiment. This suggests that the number of replicates were “adjusted” to bias for statistical significance, as the effect size of the data may be marginal. Additionally, the authors have seemed to make prior substantive changes and additional experiments for the first revision.

Below are comments on the respective figures and text:

Authors’ response: We thank the reviewer for the comment and suggestions. In the Introduction, we aimed to provide a concise overview of the study's essential aspects to help readers grasp the subject easily. We then elaborated more background details in the Results section, preceding each experiment, to clarify the experimental logic. The manuscript represents a comprehensive study involving a combination of metabolic, transcriptional and epigenetic aspects, which is challenging to present in an effective yet simple way, something that we have done our best to accomplish.

Our conclusions are based on multiple diverse experiments and complementary analysis. We are committed to reporting informative data, even where a significance threshold has not been reached, for complete transparency. We do not draw stand-alone conclusions from such data, instead we support them with other experiments and analysis. Heart failure is a chronic disease that develops over decades, and in which patients exhibit variable severity of the phenotype, therefore even small changes can have significant physiological impact.

Regarding the reviewer's concern about sample size, we wish to emphasize that we neither excluded any data points (not even outliers) nor did we adjust any sample sizes with the intention to achieve statistical significance. The data that reviewer is referring to are for two different experiments, measuring two different biological effects and performed on two different cohorts of animals (as clarified in comment 5). To facilitate assessment by readers, we presented actual p-values in all figures transparently and refrained from using star-based indicators.

Comments

1- Figure 1g: Authors report Idh2 expression in Sham vs MI hearts. Using 2 animals for sham and 4 for MI is a bit too low to calculate statistical significance, and to draw conclusions.

Authors' response: This panel in Figure 1g (highlighted in a red square in the figure below) was generated by re-analyzing previously published RNA-seq data². In this data, Eijgenraam T. *et. al.*, had only two animals in the sham group and 4 animals in the myocardial infarction (MI) group. We consulted a statistician when incorporating these data into our manuscript. His opinion was that it is not meaningless to use a control group with only two samples, and suggested that we could still run a *t*-test on it. The *p*-value itself has limited interpretation, while the non-overlapping SEM intervals are more indicative in this case. In any case we do not state that the difference is statistically significant (because it is not, given the low sample size). It is worth emphasizing in this context that in all other murine models and in human patients with DCM, *Idh2/IDH2* downregulation is indeed statistically significant (Fig. 1f,g).

2- Figure 1h: The reduction of IDH2 protein level is approximately only 10%. Is this enough to have a physiological effect? Also, it would be probably better to include the Western Blot in the main figure, together with the quantification for better clarity.

Authors' response: The reviewer is right, there is an observable discrepancy between the *substantial* downregulation of *IDH2* transcript, and the comparably *lower* downregulation in *IDH2* protein that we observed in HF. However, this discrepancy is attributed to mechanisms that regulate the stability and activity of *IDH2* protein in response to oxidative stress.

Defining exact contributing factors to *IDH2* activity and stability would be interesting, but experimentally challenging, and it is not within the scope of our study. Succinylation³, glutathionylation⁴, and acetylation⁵ of *IDH2* are all posttranslational modifications that regulate its stability and activity. Characterizing their combined contributions in cellular context would require developing complicated targeted mass spectrometric approaches that enable

simultaneous quantification of these posttranslational modifications and subsequent *in vitro* studies to determine how their levels collectively affect IDH2 activity. We do not have such techniques at hand and therefore cannot offer them as part of the current paper. However, we discuss in the manuscript the effect of two competing mechanisms. The first one is a feedforward cycle involving 2OG and L2HG, which increases the expression and activity of IDH2 in response to oxidative stress. The other (opposing) mechanism is that oxidative stress induces NRF2 activation, which in turn decreases expression of IDH2 to limit overwhelming cells with antioxidants (as we further explained in comment 1.3.4 in our first Response to Referees' comments).

The reviewer is right that including the Western Blot in the main figure together with the quantification would provide better clarity, but as we are limited by available space, we chose to present it as Supplementary fig. 1k. However, we should be able to move to a main Figure during final editing if requested.

3- Figure 2a: The authors use neonatal rat cardiomyocytes to test how oxidative stress modulates IDH2.

The authors assess that *Idh2* expression is modulated according to higher concentration of H₂O₂. Here, they do not consider that higher concentration of H₂O₂ could kill the cells. *Idh2* reduction of expression could be simply associated to the reduction of cell number. The authors should perform a cell viability assay to show the number of viable cells. Quantifying the amount of RNA is not enough as, CMs can divide the nuclear content without undergoing cytokinesis.

Authors' response: We respectfully hold a differing viewpoint in this matter. Nuclear division in cardiomyocytes without undergoing cellular cytokinesis would be expected to increase the amount of RNA, not decrease it. In this case, quantifying the amount of RNA serves as an indicator of surviving cells. For instance, when treating the cardiomyocytes with 300 μM of H₂O₂, we observed a substantial reduction in the amount of recovered total RNA, which dropped by more than 90% (to 9.31%) of its levels in the control group (100%) after 48 hours of treatment. This drastic decrease in RNA quantity can be attributed to a significant loss of cardiomyocytes due to the treatment, and such a pronounced effect would not necessitate a cell viability assay for confirmation. Cell viability assay in hFCC1 cells exposed to 300 μM of H₂O₂ showed a very similar trend of cell loss to the trend concluded from recovered amounts of total RNA (see the figure below).

Cell viability estimated with CellTiter-Fluor™ Cell Viability Assay (Promega, G6080) in hFCC1 cells exposed to 300 H₂O₂

However, the reduction of total RNA observed with the increased doses of H₂O₂ treatment exerted no effect on the measurement of *Idh2* expression, because we employed same amounts of total RNA in all qPCR reactions, and we further normalized C_q values of target genes in every reaction to corresponding C_q value of *Gapdh*. We quantified total extracted RNA with Qubit, and then employed 5 ng of total RNA in all qPCR reactions, as we stated both in the text and figure legend. Any potentially induced oxidative damage by H₂O₂ on

the RNA molecules would affect both target genes and *Gapdh*, and thus it would be subsequently normalized.

Below we also show a screenshot for Cq values of target genes from neonatal cardiomyocytes treated with H₂O₂ for 1 day. All Cq values of *Gapdh* remain largely unchanged by increasing the doses of H₂O₂, indicating equal amounts of employed total RNA in each reaction.

Sample		Idh2	Hmoz1	Nqo1	Osgin1	Gapdh
		C1	22.36	24.61	24.88	28.06
		22.49	24.62	24.67	28.10	19.13
C2		22.20	24.67	24.77	28.15	18.86
		22.28	24.90	24.75	28.47	19.19
C3		22.10	24.52	24.58	28.14	18.92
		22.09	24.62	25.04	28.29	18.99
C4		22.00	24.47	24.75	28.22	18.92
		22.12	24.88	25.00	28.41	18.86
25 μ M-1		22.07	24.72	24.75	28.31	18.96
		22.31	24.99	25.01	28.46	19.05
25 μ M-2		22.06	24.69	24.71	28.24	18.88
		22.16	24.84	25.19	28.48	19.12
25 μ M-3		22.21	24.86	24.74	28.31	19.09
		22.30	24.82	25.05	28.57	19.17
25 μ M-4		22.50	24.87	24.73	28.40	19.05
		22.49	24.99	24.91	28.30	19.01
50 μ M-1		22.93	24.06	23.90	28.11	18.76
		22.82	24.23	24.18	28.07	18.95
50 μ M-2		22.37	24.45	24.08	28.10	18.92
		22.48	24.30	24.39	28.25	19.13
50 μ M-3		22.28	24.27	24.18	28.28	18.80
		22.29	24.20	24.38	28.07	19.01
50 μ M-4		22.33	24.11	24.06	28.48	19.01
		22.50	24.12	24.25	28.32	19.11
100 μ M-1		23.88	22.58	22.36	27.96	19.12
		24.08	22.87	22.61	27.98	19.30
100 μ M-2		24.14	23.12	22.36	27.76	19.21
		24.16	22.63	22.45	28.04	19.37
100 μ M-3		24.07	22.66	23.04	27.97	19.25
		24.13	22.63	22.63	28.06	19.37
100 μ M-4		24.07	22.50	22.12	27.75	19.25
		24.24	22.58	22.13	27.65	19.09
200 μ M-1		25.00	21.55	21.20	27.22	18.90
		24.93	21.77	21.31	27.36	19.03
200 μ M-2		25.03	21.57	21.23	27.33	18.80
		24.98	21.81	21.43	27.34	19.08
200 μ M-3		24.57	21.42	21.19	27.31	18.84
		24.67	21.69	21.46	27.39	18.90
200 μ M-4		24.95	21.68	21.46	27.47	18.95
		25.33	21.91	21.68	27.42	19.00
300 μ M-1		25.48	21.25	21.16	26.77	18.61
		25.52	21.46	21.20	27.07	18.78
300 μ M-2		26.02	21.66	21.42	26.97	18.85
		26.07	21.70	21.51	27.07	19.07
300 μ M-3		25.84	21.39	21.23	26.96	18.61
		26.16	21.53	21.26	26.93	18.71
300 μ M-4		25.74	21.46	21.14	27.16	18.77
		26.02	21.49	21.24	27.35	18.65

Gapdh Cq ≈ 19

Figure 2a

Screenshot from the raw Cq values of qPCR analysis of NRCMs treated for 24 hours with increasing doses of H₂O₂ presented in figure 2a. The two rows for each sample represent technical duplicates.

4- Line 197 to 201: the authors state:” as the level of H₂O₂ gradually declined, this expression returned almost to normal, particularly in the case of cells exposed to low concentrations”. At the same time, further incubation of these cells with fresh medium free from H₂O₂ for an additional 24 hours did not raise the level of Idh2 expression any further, indicating the potential presence of long-lasting effects”. Additionally, this data is not present in Figure 2A, and may be mislabelled.

Authors’ response: We checked this, and while the text mentioned in the manuscript accurately describes the modulation of *Idh2* levels at different time points depicted in figure 2a, we made small changes for clarity. The figure is not mislabeled and remains unchanged.

We quote below the complete revised paragraph from the manuscript (the first part is missing in the reviewer’s quoted text) with the changes made in bold:

*“When neonatal rat cardiomyocytes (NRCMs) were subjected to oxidative stress through exposure to increasing concentrations (0-300 μM) of H₂O₂ for 6 hours (Fig. 2a), concentration-dependent downregulation of Idh2 occurred (Fig. 2a). Moreover, as the level of H₂O₂ gradually declined, **Idh2** returned almost to normal, particularly in the case of cells exposed to low **H₂O₂** concentrations. At the same time, further incubation of these cells with fresh medium free from H₂O₂ for an additional 24 hours did not raise the level of Idh2 expression any further, indicating the potential presence of long-lasting effects (Fig. 2a).”*

Regarding the figure: At low concentrations of H₂O₂ (i.e., 25 and 50 μM), *Idh2* expression decreases after 6 hours of treatment (green curve). However, it almost returns to its normal level in the cells exposed to 25 μM of H₂O₂ after 24 hours of treatment (red curve) and comes close to its normal level in cells exposed to 50 μM. Nevertheless, further incubation of the cells with fresh media for an additional 24 hours does not induce a further increase in *Idh2* levels in surviving cells treated with 50 μM of H₂O₂ (blue curve). It is well established that H₂O₂ is not stable in the absence of a stabilizer, and its concentration gradually decreases due to processes such as oxidation, reduction, and disproportionation. These processes are especially pronounced in the presence of organic or reactive compounds found in the media or produced by the cultured cells. We believe, therefore, that our description is accurate.

Figure 2a

5- Figure 2c and d: I presume that the animals used to measure IDH2 activity are the same. But in figure 2c they report n=7 WT and n=11 *Mlp*^{-/-}, while in figure 2d the numbers are n=7 WT and n=14 *Mlp*^{-/-}. Are those 2 independent experiments, with different animals used, or the number of *Mlp*^{-/-} animals has been modified between figure 2c and d? Were outliers removed for Figure 2C? In contrary, for figure 3a, the authors measure L2HG and D2HG from the same number of WT and *Mlp*^{-/-} animals.

Authors' response: The experiments described in figures 2c and in figure 2d are two different experiments involving different cohorts of animals. Figure 2c depicts *Idh2* activity, which is measured through a kinetic assay, whereas figure 2d depicts a completely different assay (colorimetric) that measures 2OG level. Both assays are comprehensively described in the materials and method file.

The experiments above had to be done on different animals. The left ventricle of the mouse heart is very small, particularly in WT female animals (~ 30-40 mg). We had two large cohorts of animals, the first consisting of 7 WT and 11 *Mlp*^{-/-} males and 6 WT and 10 *Mlp*^{-/-} females. The majority of the studies described in this work that required small quantity of tissues were performed using this cohort, including Western blotting, qPCR, RNA seq, *Idh2* activity assay, mass spectrometry measurements (L2HG, D2HG, lactate, malate, succinate, GSH, GSSG and MDA). The second cohort comprised 7 WT and 14 *Mlp*^{-/-} males and was specifically utilized for 2OG measurements, which required 20 mg of tissue. Additionally, this cohort was employed to test various assays for quantifying GSH, GSSG, and MDA, as detailed in Section I of our first Response to Referees letter.

Furthermore, we utilized several smaller cohorts of WT and *Mlp*^{-/-} mice and of both sexes to optimize various assays, and some of these samples were combined with samples from the larger cohort of animals. It's important to note that for some assays, the sample size was inherently limited by factors such as the number of wells on a gel or the available indexes for

RNA sequencing. In these instances, animals were chosen based on a random numeric order to ensure unbiased selection.

Regarding the reviewer comment “ *In contrary, for figure 3a, the authors measure L2HG and D2HG from the same number of WT and Mlp-/- animals*”: The sample size is not the same in figure 3a and in figure 2c or 2d.

All IDs of animals utilized in this study will be traceable in the published raw data.

6- Figure 3f and g: The authors state:” Although elevated levels of D2HG have been reported to downregulate IDH2 in a variety of cell types, we observed that elevated levels of L2HG in NRCMs, but not D2HG, were associated with upregulation of both Idh2 mRNA and protein”. Here, the level of Idh2 upregulation is minimal and the increase of IDH2 protein is not significant so the above statement is not completely supported by the data they show.

Authors’ response: Figure 3g shows a trend of increased Idh2 protein expression, which aligns with the statistically significant elevation in Idh2 mRNA levels depicted in Figure 3f.

It's important to note that Western blotting is a not a quantitative method, but rather semiquantitative. Consequently, it typically necessitates additional experiments to support conclusions drawn from Western blot data. Therefore, increasing the sample size or using different loading controls could potentially yield statistically significant results.

Figure 3

7- Figure 4a: Idh2 is downregulated in neonatal rat cardiomyocytes accordingly to administration oxidative stress by Sulforaphane. This could be again caused by cell death as also stated by the authors in line 300.

Authors' response: We appreciate the reviewer's input but maintain our view that the reduction of total RNA observed due to cell death with the increased doses of SF treatment had no effect on *Idh2* expression, because we employed same amounts of total RNA in all qPCR reactions, and we further normalized the Cq values of target genes in every reaction to the corresponding Cq value of *Gapdh*, as explained above in our response to comment 3.

8- Figure 4a: Total RNA is reduced at 24h in cells treated with NAC, which is not supposed to cause cell death. Even if there is no proliferation, one would expect the number of cells and RNA to be at least the same as at 6h and not reduced at 24 and 48h in control condition (NAC). Please clarify.

Authors' response: For clarity: The amount of RNA recovered from cells treated with NAC is presented as a percentage (%) relative to the control (untreated) condition at each time point. Consequently, the value for the control condition remains at 100% across all time points.

According to the best of our knowledge, there are no reports stating that NAC and the same treatment duration as we used does not cause any cell death. In all our experiments, we observed a noticeable yet slight reduction in the recovered amount of total RNA, particularly after 24 hours of NAC treatment, in both cardiomyocytes and hFFC1 cells. Some of this data are presented in Fig. 4a and Supplementary Fig. 4b,f, shown below. Moreover, a higher concentration of NAC (i.e., 6 μ M) resulted in greater reduction in the recovered amount of total RNA (Supplementary Fig. 4b) suggesting a dose-dependent effect of NAC on cell death.

Fig.4

(a)

Supplementary fig.4

(b)

(f)

Cell viability assay in hFCC1 cells exposed to NAC also showed a very similar trend of cell loss to the trend concluded from recovered amounts of total RNA (please see the figure below).

Cell viability estimated with CellTiter-Fluor™ Cell Viability Assay (Promega, G6080) in hFCC1 cells exposed to increasing concentrations of NAC.

On a general note, NAC is extensively employed as a pharmaceutical and antioxidant supplement, and numerous investigations have reported conflicting mechanisms for NAC inducing or inhibiting apoptosis using different cell types and doses. However, cardiomyocytes death associated with NAC is not within the scope of our current study, but it would certainly be interesting to investigate in future studies. While our experiment was designed to answer a specific scientific question regarding Idh2 modulation when exposed to oxidants and antioxidants, it raised new questions for further investigation, as often happens.

9- Figure 4c: The authors show that Idh2 expression is reduced according to increased concentration of Sulforaphane. The authors also declare "Sulforaphane induced substantial cell death". This assumption comes from the quantification of total RNA as shown in supplementary figure 4b. Here, also normal incubation with NAC induced a reduction of total RNA after 24 or 48h. If the increase in Sulforaphane concentration cause cell death, this could also imply that lower Idh2 expression could be due to a lower number of cells. This could also be the case at 5uM and 7.5uM Sulforaphane. A cells viability assay, like a TUNEL assay, could clarify this.

Authors' response: This is the same concern as above regarding using RNA as a proxy for cell viability, so please see our response to comments 4 and 8.

10- In supplementary figure 4e, the authors test the Caspase 3 in protein extracts at different concentration of Sulforaphane. Here, they don't observe any difference in Caspase 3, also at 10uM of Sulforaphane. This is in contrast with what stated and measured at total RNA level.

Authors' response: While the difference in caspase 3 at 10 μM is not statistically significant, the trend is a proportional decrease in caspase 3 levels with increasing concentrations of SF, (Supplementary Fig. 4e, shown below). This limitation is attributed to the relatively small sample size and higher variability within the control group. As previously emphasized in our response to comment 6, Western blotting is a semi-quantitative method. Consequently, it requires complementary experiments to substantiate conclusions drawn from Western blot results.

We do not expect that caspase 3 levels in these extracts would match apoptosis in a linear fashion. This is because caspase 3 is cleaved in the execution-phase of cell apoptosis, meaning that apoptotic cells will be fragmented and detached from the surface. Apoptotic cells are in all likelihood lost during the multiple washing steps carried out before trypsinization and cell harvesting for protein extraction. Accordingly, cleaved caspase 3 will be lost along with these lost cells, thereby reducing the observable effect on caspase 3 levels. In contrast, the RNA content in dying cells is similarly lost, but has the opposite effect of amplifying the difference compared to the control group (Supplementary Fig. 4f, shown below).

Furthermore, to support the interpretation of the observed caspase 3 trend, we also noted a statistically significant upregulation of Osgin1, which its increased expression serves as an upstream element in the apoptosis cascade ⁶, at both the RNA and protein levels (Supplementary Fig. 4e,f, shown below). In addition, cell viability assay in these cells exposed to SF also showed a very similar trend of cell loss to the trend concluded from recovered amounts of total RNA (Supplementary Fig. 4f, shown below).

Supplementary Fig. 4

(e)

(f)

11- Figure 4g: The expression of HMOX1 and NQO1 are very similar in *Mlp*^{-/-} Male and female mice. Even though, *Hmox1* is more expressed at mRNA level, HMOX1 protein is expressed in the similar way in both *Mlp*^{-/-} Male and female mice so the assumption that “the increase in *Hmox1* was more noticeable in male mice compared to female ones, a difference that was even more pronounced on the mRNA levels, where *Hmox1* mRNA was upregulated only in males”. The quantification in Figure 4G suggest otherwise, how many replicates was this performed?

Authors’ response: Contrary to the reviewer, we argue male *Mlp*^{-/-} mice shows a statistically significant ($p = 0.026$) increase in *Hmox1* compared to WT on the protein level whereas in females the increase in *Hmox1* level in *Mlp*^{-/-} vs. WT is not statistically significant ($p = 0.111$) (Fig. 4g, shown below). This difference is even clearer on the RNA level (Fig. 4h, shown below). Both the protein data and qPCR data are presented as a fold-change, meaning that each was normalized to their WT controls, adjusted to the value of 1. We also provide a second technical replicate in which *Hmox1* expression was measured by RNA sequencing, showing the same result: *Hmox1* was significantly upregulated (Log2 Fold change = - 0.636, $p = 0.013$) in male *Mlp*^{-/-} hearts vs. WT, whereas it was similar in female hearts (Log2 Fold change = -0.007, $p = 0.982$). RNA seq data is provided in supplementary data file 5. Therefore, we believe that the presentation of the data in the text accurately describes the results.

Fig. 4g

Fig. 4h

12- Figure 4j: The authors state that “male *Mlp*^{-/-} hearts had substantially higher ratio in comparison to female ones “. GSH/GSSG ratio seem to be at base line higher in WT mice in comparison to females.

Authors' response: We are somewhat confused by this comment. Our statement refers to the difference in GSH/GSSG ratio between male and female $Mlp^{-/-}$ hearts, not WT hearts. The observed difference in GSH/GSSG ratio between male and female $Mlp^{-/-}$ hearts is statistically significant ($p= 0.001$), whereas the difference between male and female WT hearts is not ($p = 0.634$). We have now added also this number to new **Fig. 4j**, shown below.

13- Figure 5c and d: “at the younger age females exhibited a greater fraction of LVEF, a difference that became more pronounced with age as LVEF improved in female mice, but not in males”. The authors should explain how $Mlp^{-/-}$ female improve LVEF from 10 to 14 weeks without any treatment. In addition, some discrepancies are noticed in between panel c and d. In panel c, n=17 mice are used for male and females, while in panel d, n=11 mice are used. Are those 11 mice selected as a subset from the 17 of panel c? Using which criteria?

Authors' response: We believe that we provided both *in vitro* and *in vivo* data to explain the reason of why $Mlp^{-/-}$ female improve LVEF from 10 to 14 weeks without treatment, opposite to males, and completely reverted when treating male and female $Mlp^{-/-}$ with Nrf2 activator AZ925.

In brief, we showed that NAC induces NRCMs hypertrophy, whereas NAC activates MEK1-ERK1/2 signaling^{7,8}, and thus it would be expected to promote the addition of sarcomeres in parallel to thicken the cardiomyocytes, similar to what was observed in our data (**Fig. 5e**). Although it would be expected that ROS would have the opposite effect, the actual mediator of this effect is most likely not ROS *per se*, but rather the capacity of the cellular antioxidative system. Chronic and low concentrations of ROS induced oxidative stress but without overriding the capacity of the antioxidative system (**Fig. 4d, e**), or inducing observable oxidative damage (**Fig. 4f and Supplementary fig. 4i**). Even with a relatively high concentration of SF (i.e., 5

μm), which induced substantial cell death (**Fig. 4a, and Supplementary fig. 4b, e, f**), surviving cells maintained substantial levels of GSH and normal levels of MDA (**Fig. 4d, f and Supplementary fig. 4i**). This has also been observed in *ex vivo* ischemic/reperfused hearts, which maintain substantial levels of GSH and capable of scavenging free radicals formed without substantial induction of MDA. Formation of MDA would rather require extreme oxidative condition to be induced, such as perfusing the heart with cumene hydroperoxide⁹. Therefore, the activation of the antioxidative response by the relatively low concentrations of ROS, such as 1 μm of SF, or 25 μM of H_2O_2 , which leads to a substantial increase in reductants in the cells (Fig. 4d), would promote a hypertrophic phenotype similar to the effect of NAC. Such hypertrophic effect of 1 μm of SF on NRCM can be observed in our data in the slight increase of recovered RNA amounts (**Supplementary fig. 4b**).

The redox state of cells treated with 1 μM of SF recapitulated the state observed in the left ventricles of *Mlp*^{-/-} animals, and particularly in females, whereas male hearts showed slightly higher oxidative stress, but without overriding the antioxidative capacity (**Fig. 4g-k**). Female *Mlp*^{-/-} hearts showed a substantial induction in GSH, reflected on a substantial reduction in MDA (**Fig. 4k**) and a higher activation of ERK/MAPK pathway (The figure is shown below). These observations may link the observed cardiac hypertrophy in *Mlp*^{-/-} females to the redox state, as the activation of MEK1-ERK1/2 signaling is expected to mediate concentric growth¹⁰.

Ingenuity Pathways comparison analysis on transcriptomic data sets from the LV of male and female *Mlp*^{-/-} in comparison to WT, or from the LV of male and female *Mlp*^{-/-} treated with Nrf2 activator AZ925 in comparison to scrambled treated *Mlp*^{-/-} controls.

Furthermore, when *Mlp*^{-/-} mice were treated with Nrf2 activator, male *Mlp*^{-/-} hearts, which had less basal activation of GSH (**Fig. 4i, k**), exhibited substantial improvement in their function (**Fig. 6b, c**). This improvement was associated with an increase in the GSH level (**Fig. 6**), increased activation of ERK/MAPK pathway (The figure is shown above) and thickening in the left ventricle walls (**Supplementary fig. 6a**). Whereas female *Mlp*^{-/-}, which had already elevated basal levels of GSH, did not benefit from such treatment, but rather exhibited adverse effects preventing further thickening of the left ventricle walls (**Supplementary fig. 6a**).

There are certainly more factors contributing to the hypertrophy observed in female *Mlp*^{-/-} mice over time. Heart failure-associated sex dimorphism is a hot research topic.

Authors' response: Regarding the reviewer comment **"In addition, some discrepancies are noticed in between panel c and d. In panel c, n=17 mice are used for male and females, while in panel d, n=11 mice are used. Are those 11 mice selected as a subset from the 17 of panel c? Using which criteria?"**

There is no discrepancy; panels 5c and 5d depict two different types of studies. In panel 5c, we show the ejection fraction of 17 males and 17 females *Mlp*^{-/-} at 10 weeks of age, whereas panel 5d depicts a time-lapse study. Eleven animals were examined with echocardiography at age 10 weeks and were followed for 4 more weeks (Panel d), whereas the remaining six animals were used for a different study immediately after the echocardiography was performed (treated with the scrambled control in the Nrf2 study). Panel 5c comes from a later *in-silico* study in which we studied the effect of the scrambled treatment in the Nrf2 study on the ejection fraction in comparison to untreated animals under different time points, in which all animals were untreated under the first echocardiographic examination. Therefore, it was possible to combine echocardiographic data together. As we highlighted earlier, all IDs of animals utilized in this study will be traceable in the published raw data.

14- Figure 6b: "Consistent with our hypothesis, daily gavage with AZ925 for 30 days and monitoring with echocardiography on days 0, 14 and 30 revealed significant improvement of LVEF in male but not female *Mlp*^{-/-} mice". The significant improvement stated by the authors is very minimal and it is not clear why the number of mice at day 0 is n=6 for both Scr. And Nrf2 act., while it becomes n=5 for Nrf2 act. at Day 14 and Day 30. Additionally, the spread of data for females is about 40%, while for males about 20-25%. Have the authors checked for equal variance when relevant statistical tests are applied.

Authors' response: The significant improvement observed in male *Mlp*^{-/-} upon treatment with the Nrf2 act. AZ925 is not minimal, it is huge in cardiology terms. The ejection fraction increased from 29.7% before treatment to 47.4% after treatment in AZ295 treated group, in comparison to no observable changes in the scrambled treated group (i.e., 34.7% vs. 34.8%). Moreover, this difference is statistically significant ($p = 0.026$ for before vs. after treatment, and $p = 0.0373$ for Nrf2 activator vs. scrambled treated group). Patients with an ejection fraction of 29.7% are considered to have severe DCM, whereas an ejection fraction of 47% is almost close to the normal value (50%).

The lower sample size between day 0 and day 30 in the Nrf2 activator treated males is due to the unfortunate death of one male in the Nrf2 activator group immediately after the first gavage. This was most likely a result of injuries associated with the gavage procedure, as explained in the Materials and Methods section.

We do not compare male vs. female in this experiment. We compare male *Mlp*^{-/-} to each other, (i.e., before vs. after treatment and Nrf2 activator vs. scrambled treated mice, at different time points), and we compare females to each other.

The reviewer is right that *Mlp*^{-/-} male hearts have a lower ejection fraction than females, and this is what we reported and discussed in figure 5 under the section (Redox-associated sex dimorphism in cardiac phenotype), and in other subsequent sections.

We checked for equal variance using a variety of statistical tests, and the data is normally distributed, please see table I below. We additionally performed an unpaired t-test using Welch's correction assuming non equal variance, and the difference is still statistically significant ($p = 0.043$).

Normality of Residuals				
Test name	Statistics	P value	Passed normality test (alpha=0,05)?	P value summary
Anderson-Darling (A2*)	0,2082	0,8181	Yes	ns
D'Agostino-Pearson omnibus (K2)	0,6980	0,7054	Yes	ns
Shapiro-Wilk (W)	0,9612	0,7863	Yes	ns
Kolmogorov-Smirnov (distance)	0,1409	0,1000	Yes	ns

Table I: Normality test of data depicted in fig. 6b on Day 30. The analysis was performed with GraphPad prism.

15- Figure 6k: “additional exogenous induction of antioxidative response resulted in further pronounced downregulation of *Idh2* in treated female mice”. This is another statement supported by a borderline statistically significant difference.

Authors' response: Observed downregulation of *Idh2* in female *Mlp*^{-/-} treated with Nrf2 activator is statistically significant ($p = 0.048$). The statistically significant difference was also reproducible when same samples were analyzed by RNA sequencing, in which the downregulation of *Idh2* transcripts appeared to be statistically significant in female *Mlp*^{-/-} treated with nrf2 activator vs scrambled control (Log2 Fold change = -0.232, $p = 0.037$), but not in males (Log2 Fold change = 0.015, $p = 0.913$). RNA seq data is provided in supplementary data file 5.

16- Minor comments:

1. Authors may want to consider moving line 191-193 to the next section “oxidative stress downregulates IDH2” as the background and topic sentence.

Authors’ response: we edited as suggested.

2. Figure 2 Legends: the word mean is missing in mean \pm SEM.

Authors’ response: We can not find what the reviewer is referring to. The figure legend is already stating “Bars represent mean \pm SEM”.

3. Line 235-237: Please reword and reformat the sentence structure – “ Under hypoxic conditions, cellular levels of 2OG and lactate rise while the level of L-malate falls, causing lactate (LDH) and malate dehydrogenase (MDH) to reduce 2OG to L-2-hydroxyglutarate (L2HG).

Authors’ response: We do not fully understand which part of the sentence should be reworded and reformatted. The sentence is grammatically correct and accurately describes cited literature. It was phrased by an English native scientist. We however changed the word “causing” to “promoting” if this what the review was referring to.

References:

- 1 Arber, S., Hunter, J. J., Ross, J., Hongo, M., Sansig, G., Borg, J., Perriard, J.-C., Chien, K. R. & Caroni, P. MLP-Deficient Mice Exhibit a Disruption of Cardiac Cytoarchitectural Organization, Dilated Cardiomyopathy, and Heart Failure. *Cell* **88**, 393-403, (1997).
- 2 Eijgenraam, T. R., Boogerd, C. J., Stege, N. M., Oliveira Nunes Teixeira, V., Dokter, M. M., Schmidt, L. E., Yin, X., Theofilatos, K., Mayr, M., van der Meer, P., van Rooij, E., van der Velden, J., Sillje, H. H. W. & de Boer, R. A. Protein Aggregation Is an Early Manifestation of Phospholamban p.(Arg14del)-Related Cardiomyopathy: Development of PLN-R14del-Related Cardiomyopathy. *Circulation Heart Failure* **14**, e008532, (2021).
- 3 Zhou, L., Wang, F., Sun, R., Chen, X., Zhang, M., Xu, Q., Wang, Y., Wang, S., Xiong, Y., Guan, K. L., Yang, P., Yu, H. & Ye, D. SIRT5 promotes IDH2 desuccinylation and G6PD deglutarylation to enhance cellular antioxidant defense. *EMBO reports* **17**, 811-822, (2016).
- 4 Kil, I. S. & Park, J. W. Regulation of mitochondrial NADP $^{+}$ -dependent isocitrate dehydrogenase activity by glutathionylation. *The Journal of biological chemistry* **280**, 10846-10854, (2005).
- 5 Yu, W., Dittenhafer-Reed, K. E. & Denu, J. M. SIRT3 protein deacetylates isocitrate dehydrogenase 2 (IDH2) and regulates mitochondrial redox status. *Journal of Biological Chemistry* **287**, 14078-14086, (2012).
- 6 Yao, H., Li, P., Venters, B. J., Zheng, S., Thompson, P. R., Pugh, B. F. & Wang, Y. Histone Arg modifications and p53 regulate the expression of OKL38, a mediator of apoptosis. *The Journal of biological chemistry* **283**, 20060-20068, (2008).

- 7 Li, W. Q., Dehnade, F. & Zafarullah, M. Thiol antioxidant, N-acetylcysteine, activates extracellular signal-regulated kinase signaling pathway in articular chondrocytes. *Biochemical and Biophysical Research Communications* **275**, 789-794, (2000).
- 8 Yan, C. Y. & Greene, L. A. Prevention of PC12 cell death by N-acetylcysteine requires activation of the Ras pathway. *Journal of Neuroscience* **18**, 4042-4049, (1998).
- 9 Janssen, M., Koster, J. F., Bos, E. & de Jong, J. W. Malondialdehyde and glutathione production in isolated perfused human and rat hearts. *Circulation research* **73**, 681-688, (1993).
- 10 Kehat, I., Davis, J., Tiburcy, M., Accornero, F., Saba-El-Leil, M. K., Maillet, M., York, A. J., Lorenz, J. N., Zimmermann, W. H., Meloche, S. & Molkentin, J. D. Extracellular signal-regulated kinases 1 and 2 regulate the balance between eccentric and concentric cardiac growth. *Circulation research* **108**, 176-183, (2011).

REVIEWERS' COMMENTS

Reviewer #4 (Remarks to the Author):

The authors have substantially revised the manuscript with additional experiments using TGFB3 KO and miR-129-5p overexpressing endothelial cells, to support the role of HDAC3 in regulating myocyte proliferation via miR-129-5p. Additionally, using E11.5 Cardiac explants and FACS-sorting of endocardial cells, the authors have clarified the previous difference in effect sizes, and majority of my comments.

While the authors have shown evidence regulated by the HDAC3-miR-129-5p axis, at least other potential signaling pathways, its linkages and future research gap should be discussed in the discussion section.

Response to reviewers' comments regarding the revised manuscript NCOMMS-22-31381D (Elbeck et al 2024)

Epigenetic modulators link mitochondrial redox homeostasis to cardiac function in a sex dependent manner

Reviewer #4

Authors' response: We thank reviewer #4 for all the feedback and all suggestions that improved the quality of our work.